# CONTROLLING REPETITION IN PROTEIN LANGUAGE MODELS

**Jiahao Zhang**[1,*]**, Zeqing Zhang**[1,*]**, Di Wang**[2]**, and Lijie Hu**[1,†]

[1]Mohamed bin Zayed University of Artificial Intelligence (MBZUAI)
[2]King Abdullah University of Science and Technology (KAUST)

## ABSTRACT

Protein language models (PLMs) have enabled advances in structure prediction and de novo protein design, yet they frequently collapse into pathological repetition during generation. Unlike in text, where repetition merely reduces readability, in proteins it undermines structural confidence and functional viability. To unify this problem, we present the first systematic study of repetition in PLMs. We first propose quantitative metrics to characterize motif-level and homopolymer repetition and then demonstrate their negative impact on folding reliability. To address this challenge, we propose UCCS (Utility-Controlled Contrastive Steering), which steers protein generation with a constrained dataset. Instead of naively contrasting high- vs. low-repetition sequences, we construct contrastive sets that maximize differences in repetition while tightly controlling for structural utility. This disentanglement yields steering vectors that specifically target repetition without degrading foldability. Injected at inference, these vectors consistently reduce repetition without retraining or heuristic decoding. Experiments with ESM-3 and ProtGPT2 in CATH, UniRef50, and SCOP show that our method outperforms decoding penalties and other baselines, substantially lowering repetition while preserving AlphaFold confidence scores. Our results establish repetition control as a central challenge for PLMs and highlight dataset-guided steering as a principled approach for reliable protein generation.

## 1 INTRODUCTION

Protein language models (PLMs) are trained on large corpora of protein sequences in a manner analogous to large language models (LLMs) in natural language processing (Consortium, 2024). They have enabled impressive advances in structure prediction and *de novo* protein design (Lin et al., 2022; Hayes et al., 2025; Ferruz et al., 2022; Madani et al., 2023; Li et al., 2024; Hesslow et al., 2022). Despite these successes, PLMs may suffer characteristic generation failures. Among them, the most prominent is **pathological repetition**, where outputs collapse into redundant motifs or long homopolymers, severely limiting the biological utility of generated sequences. To illustrate this phenomenon, we present a representative case in Table 1.

Repetition is not unique to proteins: in natural language generation, LLMs frequently produce low-diversity continuations that harm readability (Holtzman et al., 2020; Welleck et al., 2020; Meister et al., 2023). However, in the protein domain, the consequences are substantially more severe. Repetitive amino acid sequences lack structural diversity, often resulting in unstable folds and non-functional proteins. Moreover, repetition in proteins is more complex than in text because it directly interacts with structural constraints: suppressing it too aggressively may degrade folding reliability and reduce biological plausibility. Surprisingly, this issue has remained largely overlooked in the protein modeling community.

In natural language processing, two classes of approaches have been explored to mitigate repetition. The first are *decoding-based heuristics*, including stochastic sampling strategies (Fan et al., 2018;

---

[*]The first two authors contributed equally to this work.
[†]Correspondence to Lijie Hu (`lijie.hu@mbzuai.ac.ae`).

Table 1: Representative cases of natural proteins vs. PLM generations arranged by sequence length (rows) and dataset/model (columns). Color indicates pLDDT confidence: blue = high (> 90), cyan/green = medium (70–90), orange/red = low (< 70).[1]

| length (aa) | CATH | SCOP | Uniref50 | ESM3 | ProtGPT2 |
|---|---|---|---|---|---|
| 128 | | | | | |
| 256 | | | | | |
| 512 | | | | | |

Holtzman et al., 2020), $n$-gram blocking (Kulikov et al., 2019), and repetition penalties. More recently, *interpretability-inspired methods* leverage internal representations to identify and steer away from degenerate behaviors (Yao et al., 2025). In PLMs, decoding-time interventions such as repetition penalties, temperature scaling, and sampling adjustments have been adopted, largely by directly transferring techniques from NLP (Ferruz et al., 2022; Chen et al., 2024). Some of these heuristics may incidentally reduce repetition, but they are not specifically designed to address this problem in proteins. Moreover, they often introduce undesirable side effects, such as lowering structural confidence (e.g., reduced AlphaFold pLDDT (Jumper et al., 2021)), thereby undermining biological viability. This trade-off highlights the need for approaches that are not only tailored to the protein domain but also more principled and interpretable.

Addressing this problem poses several challenges. First, repetition and structural utility are tightly entangled, such that naïve reductions in repetition often degrade folding reliability. Second, PLMs provide no explicit mechanism to disentangle repetition from other generative factors. Third, conventional text-based repetition metrics capture only surface-level diversity and fail to reflect the biological consequences of repetitive patterns. Since pathological repetition is a newly recognized issue in protein generation, the first step is to formally define this failure mode and establish evaluation metrics. To this end, we introduce two complementary metrics. The first is a unified **repetition score** $R(x)$, integrating token-level entropy, motif-level $n$-gram diversity, and homopolymer penalties into a single measure of degeneracy. The second is a **utility score** $U(x)$, derived from AlphaFold-based structural proxies (pLDDT, pTM), which quantifies whether generated sequences remain plausibly foldable. These metrics allow us to formulate repetition control as a *constrained optimization problem*: reduce degeneracy without compromising structural plausibility. This unified perspective not only provides principled evaluation, but also forms the basis of our method.

After we unify this problem, we seek to overcome this trade-off between repetition and folding reliability. Specifically, we propose *Utility-Controlled Contrastive Steering (UCCS)*, a representation-level intervention that disentangles repetition from structural plausibility. UCCS builds contrastive datasets matched in utility but separated in repetition, derives steering vectors from hidden activations, and injects them during generation to suppress degeneracy without damaging foldability. We extensively evaluated UCCS on two representative PLMs (ESM-3 (Hayes et al., 2025) (MLM) and PROTGPT2 (Ferruz et al., 2022) (AR)) across three standard protein datasets (CATH (Knudsen & Wiuf, 2010), UniRef50(Consortium, 2024), SCOP(Murzin et al., 1995)) in two generation settings (unconditional and conditional). Results demonstrate that UCCS consistently achieves the best trade-off: it significantly improves the repetition score $R(x)$ while maintaining or even enhancing the utility score $U(x)$, outperforming decoding heuristics (e.g., temperature, top-$p$ sampling) and

---

[1]Columns 2–4 of Table 1 show **natural proteins**, while columns 5–6 display **PLM-generated sequences**. Generated cases often exhibit repetition artifacts (e.g., long homopolymer stretches in orange/red, indicative of poor foldability), whereas natural proteins display diverse and structurally coherent patterns.

mechanism-level baselines. To the best of our knowledge, this work presents the **first systematic study** that identifies, characterizes, and unifies pathological repetition as a critical failure mode in protein generation. We summarize our contributions as follows.

- We present the first systematic study of pathological repetition in PLMs, introducing principled metrics that capture both motif-level and homopolymer repetition, and validating their ability to reflect folding reliability.

- We introduce **Utility-Controlled Contrastive Steering (UCCS)**, a dataset-guided steering method that disentangles repetition control from biological utility.

- We conduct a comprehensive evaluation across modeling paradigms (auto-encoding and autoregressive PLMs), generation settings (unconditional and conditional), and multiple protein datasets (CATH, UniRef50, and SCOP). Results show that UCCS consistently reduces repetition while preserving foldability, outperforming decoding heuristics and other baselines.

## 2 RELATED WORK

**Generative PLMs for protein design.** Recent advances in protein language models (PLMs) have enabled impressive progress in *de novo* protein design. By training on large-scale sequence databases such as UniProt (Consortium, 2024), models like ESM (Hayes et al., 2025), Prot-GPT2 (Ferruz et al., 2022), and ProGen (Madani et al., 2023) learn rich statistical representations of amino acid usage and have been applied to generate novel, diverse, and partially functional proteins. These works demonstrate the potential of generative PLMs as powerful priors for protein engineering, showing applications in structure prediction (Lin et al., 2022), mutational effect estimation (Li et al., 2024), and controlled design with conditioning signals (Hesslow et al., 2022). However, previous studies overwhelmingly emphasize positive outcomes of PLM generations, while systematic *failure modes* in particular pathological repetition has received little attention. This motivates us to look at related phenomena in natural language generation, where repetition and degeneration have been extensively studied (see more related work in the Appendix A).

**Representation steering.** Beyond decoding heuristics, recent work in NLP and vision has explored manipulating internal representations to steer model behavior (Chen et al., 2025; Li et al., 2025; Hu et al., 2025; Yu et al., 2025; Hu et al., 2024). Activation steering methods identify directions in hidden states correlated with specific attributes and adjust them to amplify or suppress those attributes (Turner et al., 2024; Rimsky et al., 2024; Hong et al., 2024). Related approaches such as task arithmetic (Ilharco et al., 2023) and representation engineering (Meng et al., 2022; Dai et al., 2022; Jiang et al., 2025; Zhang et al., 2024; Yang et al., 2024) demonstrate that linear operations in embedding or activation space can achieve controlled changes without retraining. These findings suggest that neural representations encode disentangled factors of variation, but to our knowledge, no prior work has attempted to isolate or control repetition-specific directions in protein language models. Beyond these computational approaches, insights from biological systems further underscore the importance of repetition control, as discussed next.

**Biological perspectives on repetition.** While most previous discussions focus on computational aspects, biological evidence also supports the importance of controlling repetition. Although natural proteins can contain ordered repeats (e.g., $\beta$-propellers or coiled coils (Kajava, 2012)), uncontrolled repetition is typically deleterious: pathological repeat expansions such as polyglutamine tracts in Huntington's disease drive misfolding and aggregation (Gatchel & Zoghbi, 2005), while dipeptide repeats from *C9orf72* expansions in ALS/FTD disrupt cellular homeostasis (Freibaum et al., 2015). Even modest low-complexity repeats are enriched in proteins prone to aggregation or aberrant phase separation (Hughes et al., 2018). These biological findings reinforce our motivation: reducing pathological repetition in PLMs is not only a modeling concern but also critical for maintaining foldability and realistic protein design. For a broader discussion covering interpretability-oriented PLM analyses and parallels to repetition in natural language generation, please refer to Appendix A.

## 3 UNIFYING REPETITION IN PLMS

### 3.1 PROTEIN LANGUAGE MODELS

Let $\mathcal{A}$ denote the amino acid alphabet, and $x = (x_1, \ldots, x_T)$ a protein sequence of length $T$, where each $x_t \in \mathcal{A}$. A pretrained protein language model (PLM) $M$ can be trained in either of two

paradigms: (i) **Masked language modeling (MLM)**: The model estimates conditional probabilities of the form $M(x_t \mid x_{\backslash t})$, where $x_{\backslash t}$ denotes the sequence with position $t$ masked out. (ii) **Autoregressive (AR)**: The model estimates conditional probabilities $M(x_t \mid x_{<t})$ of the next token given its prefix. In generation, PLMs can operate in the *unconditional* setting (starting from a special beginning token) or in the *conditional* setting, where a prefix $p$ is provided.

### 3.2 PATHOLOGICAL REPETITION IN PLMS

We focus on a characteristic failure mode of PLMs known as **pathological repetition**. Generated sequences often collapse into degenerate, low-complexity patterns that reduce diversity and impair folding into stable 3D structures. We categorize these failures into two canonical forms.

- **Motif-level repetition:** Short $n$-gram fragments recur repeatedly (e.g., AGAGAG). This corresponds to local looping behavior, where the model fixates on a short subsequence and generates it cyclically.
- **Homopolymer repetition:** A single residue is extended into long runs (e.g., AAAAAA). This represents a global collapse into a low-complexity sequence dominated by one amino acid type.

We argue that these two categories suffice to characterize pathological repetition in PLMs. In contrast to natural language, where a large vocabulary enables many types of semantic redundancy, protein sequences are restricted to an alphabet of only 20 amino acids. This limited vocabulary constrains the space of degenerate behaviors, which therefore manifest in relatively simple but severe collapse patterns. Our empirical analysis (Appendix F.2, Fig. 1) confirms that virtually all repetitive failures produced by PLMs fall into either motif-level looping or homopolymer expansion, supporting the sufficiency of this taxonomy. The two categories are complementary: motif-level repetition corresponds to *local cyclic degeneracy*, while homopolymer repetition represents a *global collapse* into low-complexity sequences. They capture the dominant failure modes that undermine the structural plausibility of PLM-generated proteins.

### 3.3 QUANTIFYING PATHOLOGICAL REPETITION

To systematically study pathological repetition, we adopt and extend diversity metrics originally used in NLP, while adapting them to the specific characteristics of protein sequences. Token-level entropy and distinct-$n$ have proven to be effective in capturing global imbalance and local motif recurrence in natural language, and provide a natural starting point for analyzing degeneracy in PLMs. However, directly relying on these measures is insufficient for proteins: due to the much smaller amino acid vocabulary, PLMs exhibit extreme collapse behaviors, most notably long homopolymer runs, which are rarely observed in text. To address this, we design three complementary metrics: entropy, distinct-$n$, and a novel homopolymer diversity score. Each is motivated by a distinct failure mode. We will discuss them in details as following.

**Token-level entropy** ($H_{\text{norm}}$)**.** PLM generations often collapse into a small set of residues (e.g., dominated by alanine and glycine). To measure such a global imbalance, we compute the normalized Shannon entropy of the empirical unigram distribution.

$$H_{\text{norm}}(x) = \frac{-\sum_{a \in \mathcal{A}} p(a) \log_2 p(a)}{\log_2 |\mathcal{A}|}, \quad H_{\text{norm}}(x) \in [0, 1],$$

where $p(a)$ is the frequency of amino acids $a$ in sequence $x$. High entropy indicates balanced amino acid usage, while low entropy reflects global collapse into a few residues. As further demonstrated in Appendix C.2 ( Fig 6), PLM-generated sequences indeed exhibit skewed amino acid distributions, with over-representation of a few residues (e.g., alanine $A$ in ESM3). This empirical evidence supports the use of $H_{\text{norm}}$ as a biologically meaningful measure of global degeneracy(Fig 1).

$n$**-gram diversity (Distinct-$n$).** To quantify local motif repetition, we use the distinct-$n$ metric (Li et al., 2016), defined as

$$\text{Distinct-}n(x) = \frac{|\text{uniq\_ngrams}_n(x)|}{|\text{ngrams}_n(x)|}.$$

We focus on $n = 2, 3$ for both methodological and biological reasons. In NLP, Distinct-1/2 are standard indicators of short-range redundancy, whereas Distinct-1 is less informative in protein modeling due to the small amino acid vocabulary. Biologically, dipeptides and tripeptides constitute

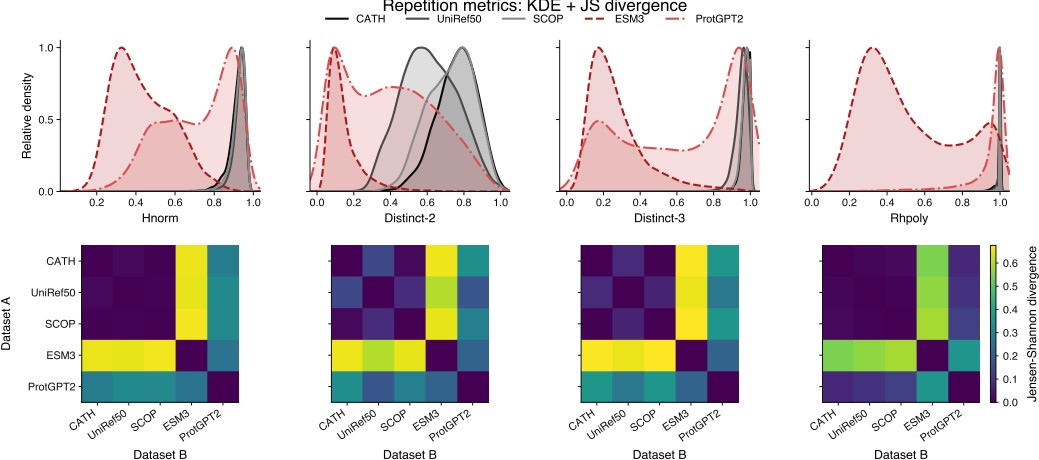

Figure 1: **Repetition metrics reveal distinct failure modes in PLMs.** Top: Kernel density estimates (relative density, peak = 1) of repetition metrics across natural (CATH, UniRef50, SCOP) and PLM-generated (ESM3, ProtGPT2) datasets. Bottom: Jensen–Shannon divergence heatmaps comparing dataset pairs. Large JS divergence between PLMs and natural datasets confirms that our metrics cleanly separate repetitive, degenerate PLM outputs from biologically diverse natural proteins. $H_{\mathrm{norm}}$ separates both PLMs from natural proteins, confirming global repetition pathology. Distinct-2/3 highlight motif-level repetition characteristic of autoregressive models like ProtGPT2, while $R_{\mathrm{hpoly}}$ captures long-run homopolymer collapse unique to masked models like ESM3.

fundamental structural motifs (e.g., `PG` loops, `Gly-X-Gly` signatures). Empirically, PLMs often degenerate into repeated short patterns (e.g., `AGAGAG`), which sharply reduces Distinct-2/3 values. In contrast, natural proteins maintain consistently higher scores (see Appendix F.2, Figs. 1). These observations support Distinct-2/3 as effective measures of motif-level diversity in protein sequences.

**Homopolymer diversity score** ($R_{\mathrm{hpoly}}$). Unlike natural language, PLMs often degenerate into long homopolymer stretches (e.g., `AAAAAAAA`), which are not captured by $n$-gram diversity: repeating a single residue trivially yields high $n$-gram overlap. To address this, we introduce the homopolymer diversity score:

$$R_{\mathrm{hpoly}}(x) = 1 - \frac{1}{T} \sum_i \ell_i \cdot \mathbf{1}(\ell_i \geq k),$$

where $\ell_i$ is the length of the $i$-th homopolymer run, and $k$ is a threshold (default $k = 4$). The choice $k = 4$ is biologically motivated: natural proteins occasionally contain short runs of 2–3 identical residues (e.g., `AA` or `GGG`), which are not pathological, but the lengths $\geq 4$ almost always correspond to low-complexity, unstable regions. Empirically, we find that such long runs are pervasive in PLM generations but rare in natural proteins. As shown in Fig. 1, Appendix F.2, $R_{\mathrm{hpoly}}$ sharply separates natural and synthetic datasets, confirming its effectiveness as a dedicated measure of homopolymer collapse.

**Aggregation.** Each metric captures a different granularity of degeneracy: $H_{\mathrm{norm}}$ reflects global diversity, Distinct-2/3 capture local motif recurrence, and $R_{\mathrm{hpoly}}$ captures long-run collapse. Individually, none of them suffices: for instance, low entropy may arise from either motif repetition or homopolymer collapse, while Distinct-2/3 fail to penalize homopolymers. We therefore aggregate them into a unified *repetition score* $R(x)$ (see Appendix E.1 for details), which provides a comprehensive quantification of degeneracy. In Section 4, we pair $R(x)$ with a complementary *utility score* $U(x)$ derived from structural proxies (e.g., AlphaFold pLDDT), ensuring that repetition control is evaluated jointly with foldability. Importantly, the validity of our repetition metrics is empirically supported: as shown in Appendix F.2, they sharply separate natural proteins (POS) from PLM generations (NEG), confirming their ability to capture pathological repetition in practice.

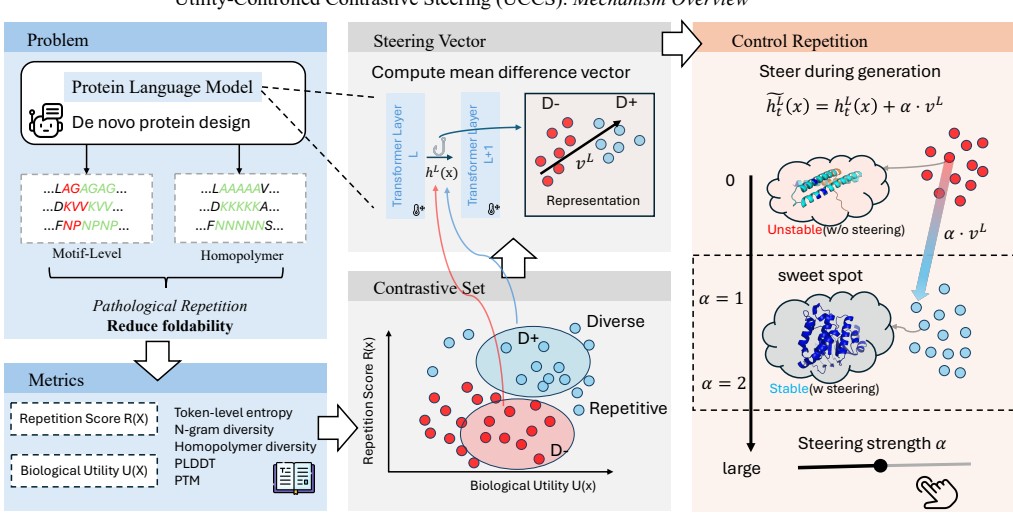

Figure 2: Overview of **Utility-Controlled Contrastive Steering (UCCS)**. The diagram illustrates (1) the identification of pathological repetition modes (motif-level and homopolymer), (2) the construction of contrastive pairs $(D^+, D^-)$ and computation of the steering vector $v^L$, and (3) inference-time control of hidden representations via $\alpha v^L$ to mitigate repetition while preserving biological utility.

# 4 UTILITY-CONTROLLED CONTRASTIVE STEERING

Decoding heuristics such as sampling adjustments (Fan et al., 2018; Holtzman et al., 2020) or $n$-gram penalties (Kulikov et al., 2019) can reduce repetition, but often at the cost of lowering structural utility. We instead intervene at the representation level by extracting latent directions that specifically encode repetition. Our method *Utility-Controlled Contrastive Steering (UCCS)* constructs paired datasets to disentangle repetition from utility, computes mean difference vectors from hidden activations, and injects them into selected layers to control generation. This provides a simple, model-agnostic mechanism for repetition control without retraining or sampling tricks. An overview of the proposed approach is shown in Figure 2.

## 4.1 PROBLEM DEFINITION

We define *repetition control* in protein language models (PLMs) as the task of reducing degenerate sequence redundancy while preserving structural plausibility. Given a pretrained PLM $M$ and an optional prefix $p$, the model generates a sequence $x = (x_1, \ldots, x_T)$. We evaluate each sequence with two complementary scores: a *repetition score* $R(x)$, capturing motif and homopolymer level collapse, and a *utility score* $U(x)$, reflecting biological plausibility based on AlphaFold confidence. The atomic metrics underlying $R(x)$ and $U(x)$ were introduced in Section 3; Their aggregation, weighting, and biological justification are provided in Appendix E. The objective is to design a modification $f$ to $M$ such that

$$\min_f \; R(f(M,p)) \quad \text{s.t.} \quad U(f(M,p)) \geq U(M,p) - \epsilon,$$

where $\epsilon$ is a small tolerance for utility degradation. This formulation is model and method-agnostic, i.e., any intervention from decoding heuristics to latent-space steering can be assessed under the same constrained optimization framework.

## 4.2 DATASET CONSTRUCTION FOR CONTRASTIVE STEERING

**Motivation.** Activation steering requires contrastive sets that differ along the target attribute while remaining aligned with potential confounders. However, in proteins, repetition and biological plausibility are naturally entangled: PLM generations with extreme repetition almost always exhibit low

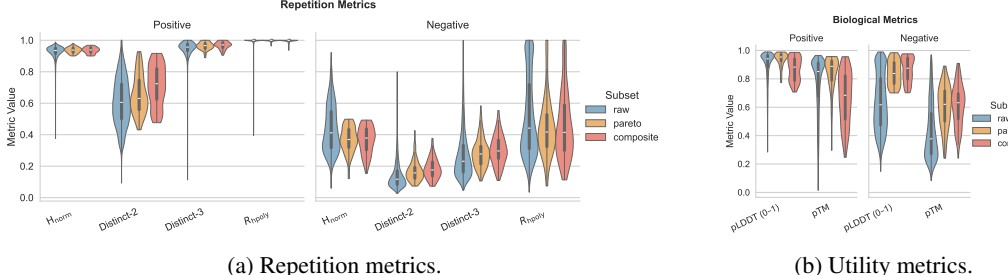

(a) Repetition metrics.       (b) Utility metrics.

Figure 3: **Effect of contrastive selection.** Raw negatives: ESM3 ($\mathcal{P}^-$, $n = 10$k); raw positives: UniRef50 ($\mathcal{P}^+$, $n = 10$k). After selection ($n = 100$ each): (a) repetition separation enlarges, (b) utility remains aligned.

structural confidence. Therefore, a naive comparison of positives and negatives would conflate $R(x)$ (repetition) with $U(x)$ (utility), making it unclear whether steering improves diversity, plausibility, or both. Our goal is to *decouple repetition from utility* so that the steering captures a clean signal of pathological repetition.

**Candidate pools.** We construct a raw candidate (Appendix F.1) from two complementary sources: (i) natural proteins sampled from CATH, SCOP, and UniRef50, and (ii) synthetic proteins generated by ESM3 and PROTGPT2. These provide sufficient coverage of low and high repetition regions across masked and autoregressive PLMs.

**Scoring and filtering.** Each sequence is annotated with $(R(x), U(x))$ as defined in Section 3.3. To weaken the intrinsic correlation between the two dimensions, we first filter out sequences with $U(x)$ outside a tolerance band around the reference mean $\bar{U}$, defined as the average utility score across the candidate pool (computed separately within each length bin; see Appendix F.3). This step discards trivial cases with extremely poor folding, ensuring that utility is not the dominant separating factor.

**Contrastive selection.** We then select $\mathcal{D}^+$ (low repetition) and $\mathcal{D}^-$ (high repetition) by maximizing their separation in $R(x)$, the repetition score, subject to a utility constraint. Formally, let

$$\Delta R = \mathbb{E}_{x \in \mathcal{D}^+}[R(x)] - \mathbb{E}_{x \in \mathcal{D}^-}[R(x)], \qquad \Delta U = \left| \mathbb{E}_{x \in \mathcal{D}^+}[U(x)] - \mathbb{E}_{x \in \mathcal{D}^-}[U(x)] \right|,$$

where $U(x)$ is the biological utility score and $\epsilon$ is a tolerance threshold. We then solve:

$$(\mathcal{D}^+, \mathcal{D}^-) = \arg \max_{\mathcal{D}^+, \mathcal{D}^-} \Delta R \quad \text{s.t.} \quad \Delta U \leq \epsilon.$$

In practice, we instantiate this procedure using either a Pareto-based frontier or a composite score ranking, with random sampling as a baseline. Algorithm details, length/utility binning, and pseudocode are provided in Appendix F.

This pipeline yields contrastive datasets that are sharply separated in repetition but comparable in utility (see Fig. 3 for dataset-level statistics). As a result, the steering vectors extracted in Section 4.3 capture latent directions of pathological repetition rather than foldability artifacts, providing a principled foundation for our method.

## 4.3 CONTRASTIVE STEERING VECTOR

**Preliminaries on representations.** Let $h_t^L(x) \in \mathbb{R}^d$ denote the hidden representation at layer $L$ for token $x_t$ in sequence $x$ of length $T$. We define a sequence-level summary representation $\phi^L(x)$ differently for masked language models (MLMs) and autoregressive language models (AR-LMs):

$$\phi^L(x) = \begin{cases} \frac{1}{T} \sum_{t=1}^{T} h_t^L(x), & \text{(MLM: mean-pooled across tokens)}, \\ h_T^L(x), & \text{(AR-LM: last-token embedding)}. \end{cases}$$

This choice aligns with the training objectives of the two paradigms, ensuring $\phi^L(x)$ is faithful to how each PLM encodes global context: mean pooling reflects the bidirectional nature of MLMs,

while last-token embeddings in AR-LMs are known to concentrate predictive information of the whole prefix (Xing et al., 2025).

**Mean difference vector.** Given contrastive datasets $\mathcal{D}^+$ (low repetition) and $\mathcal{D}^-$ (high repetition) constructed in Section 4.2, we compute the steering vector as the difference in mean sequence-level representations:

$$v^L = \mathbb{E}_{x \in \mathcal{D}^+}[\phi^L(x)] \ - \ \mathbb{E}_{x \in \mathcal{D}^-}[\phi^L(x)].$$

By construction, $v^L$ captures the latent direction that best separates high-$R(x)$ from low-$R(x)$ sequences, while controlling for $U(x)$. We refer to $v^L$ as *contrastive steering vector* in the layer $L$.

**Injection.** To control repetition during generation, we modify hidden states by injecting $v^L$ at the chosen layer $L$. At each decoding step, hidden activations are updated as:

$$\tilde{h}_t^L(x) = h_t^L(x) + \alpha \cdot v^L,$$

where $\alpha \in \mathbb{R}$ is a scalar coefficient that controls the strength and direction of steering. In all main experiments, we set $\alpha = 1$ for simplicity, while ablation studies (Section 5.3) investigate the sensitivity of results to different $\alpha$ values. This procedure requires no retraining and can be applied plug-and-play, making it an efficient and interpretable mechanism for repetition control under the framework defined in Section 4.1.

## 5 EXPERIMENTS

### 5.1 SETUP

**Datasets.** We evaluate our method on three widely used protein sequence databases: CATH (Knudsen & Wiuf, 2010), UniRef50 (Consortium, 2024), and the SCOP database (Murzin et al., 1995). From each dataset, we randomly sample approximately 10,000 protein sequences, retaining only those shorter than 1024 residues. Detailed preprocessing steps are provided in Appendix F.

**Models.** We experiment with two classes of protein language models (PLMs): masked language models (MLMs) and autoregressive language models (AR-LMs). Specifically, we adopt ESM-3 (HAYES ET AL., 2025) for the MLM setting and PROTGPT2 FERRUZ ET AL., 2022 for the AR-LM setting, with default generation configurations listed in Appendix G.1.

**Baselines.** We compare against three categories of baselines: (1) The original models without steering, using standard decoding with no intervention. (2) Decoding-level heuristics, where for ESM-3 we evaluate entropy-based unmasking (from the original release), top-$p$, and temperature sampling, and for PROTGPT2 we implement top-$p$, temperature, repetition penalty, and $n$-gram penalty strategies (details in Appendix G.2). (3) Mechanism-level interpretability methods, including neuron-based deactivation(Radford et al., 2017) and probe-based steering vectors(Elazar et al., 2021). (Appendix G.3).

**Tasks.** We consider two sequence generation paradigms: (i) *unconditional generation (uniform-50–512)*, where models generate novel protein sequences with length uniformly sampled between 50 and 512 residues; and (ii) *conditional generation (prefix-10)*, where models generate sequences conditioned on a prefix extracted from natural proteins. For each test sequence, the first 10% of residues are used as the prefix, and the remaining residues are generated with the total target length uniformly sampled between 50 and 512 residues. In both the **MLM** and **AR-LM** settings, we evaluate on both tasks, which balance computational efficiency and biological plausibility.

**Metrics.** We evaluate outputs with two composite scores: the **repetition score** $R(x)$, aggregating entropy, motif-level $n$-gram diversity, and homopolymer penalties; and the **utility score** $U(x)$, the average of pLDDT and pTM from AlphaFold confidence. Details of metric construction are given in Appendix E. Results are averaged over 5 random seeds with standard deviations.For each method, hyperparameters are selected according to the harmonic mean of $R(x)$ and $U(x)$ (Appendix G.4), ensuring a balanced trade-off between repetition reduction and biological plausibility.

**Implementation Details.** All experiments are conducted on an AutoDL server with NVIDIA RTX 5090 GPUs. We use the open-source ESM3 (esm3-sm-open-v1) and PROTGPT2 (ProtGPT2) models from HuggingFace. Default generation parameters are listed in Appendix G.1. Each experiment is repeated with 5 random seeds, and we report the mean $\pm$ standard deviation.

Table 2: ProtGPT2 results for two tasks with utility constraint: (a) unconditional generation, (b) conditional generation. [2]

| Method | CATH | | UniRef50 | | SCOP | |
|---|---|---|---|---|---|---|
| | $R \uparrow$ | $U \uparrow$ | $R \uparrow$ | $U \uparrow$ | $R \uparrow$ | $U \uparrow$ |
| **(a) Unconditional generation** | | | | | | |
| Original Model | $0.728_{\pm 0.010}$ | $0.621_{\pm 0.022}$ ✓ | $0.728_{\pm 0.010}$ | $0.621_{\pm 0.022}$ ✓ | $0.728_{\pm 0.010}$ | $0.621_{\pm 0.022}$ ✓ |
| Temperature Sampling | $0.756_{\pm 0.012}$ | $0.612_{\pm 0.019}$ | $0.756_{\pm 0.012}$ | $0.612_{\pm 0.019}$ | $0.756_{\pm 0.012}$ | $0.612_{\pm 0.019}$ |
| Top-p Sampling | $0.714_{\pm 0.013}$ | $0.608_{\pm 0.008}$ | $0.714_{\pm 0.013}$ | $0.608_{\pm 0.008}$ | $0.714_{\pm 0.013}$ | $0.608_{\pm 0.008}$ |
| No Repeat N-gram | $0.736_{\pm 0.010}$ | $0.613_{\pm 0.018}$ | $0.736_{\pm 0.010}$ | $0.613_{\pm 0.018}$ | $0.736_{\pm 0.010}$ | $0.613_{\pm 0.018}$ |
| Repetition Penalty | $0.780_{\pm 0.003}$ | $0.622_{\pm 0.018}$ ✓ | $0.780_{\pm 0.003}$ | $0.622_{\pm 0.018}$ ✓ | $0.780_{\pm 0.003}$ | $0.622_{\pm 0.018}$ ✓ |
| Neuron Deactivation | $0.719_{\pm 0.017}$ | $0.610_{\pm 0.013}$ | $0.718_{\pm 0.019}$ | $0.614_{\pm 0.029}$ | $0.734_{\pm 0.009}$ | $0.615_{\pm 0.015}$ |
| Probe Steering | $0.722_{\pm 0.017}$ | $0.607_{\pm 0.017}$ | $0.732_{\pm 0.015}$ | $0.624_{\pm 0.029}$ ✓ | $0.735_{\pm 0.017}$ | $0.619_{\pm 0.016}$ |
| UCCS(ours) | $\mathbf{0.845}_{\pm 0.010}$ | $\mathbf{0.711}_{\pm 0.004}$ ✓ | $\mathbf{0.824}_{\pm 0.012}$ | $\mathbf{0.703}_{\pm 0.018}$ ✓ | $\mathbf{0.835}_{\pm 0.008}$ | $\mathbf{0.722}_{\pm 0.008}$ ✓ |
| **(b) Conditional generation** | | | | | | |
| Original Model | $0.836_{\pm 0.018}$ | $0.704_{\pm 0.023}$ ✓ | $0.836_{\pm 0.018}$ | $0.704_{\pm 0.023}$ ✓ | $0.836_{\pm 0.018}$ | $0.704_{\pm 0.023}$ ✓ |
| Temperature Sampling | $0.847_{\pm 0.013}$ | $0.700_{\pm 0.021}$ | $0.847_{\pm 0.013}$ | $0.700_{\pm 0.021}$ | $0.847_{\pm 0.013}$ | $0.700_{\pm 0.021}$ |
| Top-p Sampling | $0.827_{\pm 0.014}$ | $0.710_{\pm 0.008}$ ✓ | $0.827_{\pm 0.014}$ | $0.710_{\pm 0.008}$ ✓ | $0.827_{\pm 0.014}$ | $0.710_{\pm 0.008}$ ✓ |
| No Repeat N-gram | $0.828_{\pm 0.013}$ | $0.708_{\pm 0.025}$ ✓ | $0.828_{\pm 0.013}$ | $0.708_{\pm 0.025}$ ✓ | $0.828_{\pm 0.013}$ | $0.708_{\pm 0.025}$ ✓ |
| Repetition Penalty | $0.843_{\pm 0.013}$ | $0.716_{\pm 0.021}$ ✓ | $0.843_{\pm 0.013}$ | $0.716_{\pm 0.021}$ ✓ | $0.843_{\pm 0.013}$ | $0.716_{\pm 0.021}$ ✓ |
| Neuron Deactivation | $0.836_{\pm 0.011}$ | $0.709_{\pm 0.012}$ ✓ | $0.828_{\pm 0.009}$ | $0.706_{\pm 0.015}$ ✓ | $0.827_{\pm 0.004}$ | $0.701_{\pm 0.011}$ |
| Probe Steering | $0.829_{\pm 0.019}$ | $0.702_{\pm 0.020}$ | $0.829_{\pm 0.009}$ | $0.690_{\pm 0.010}$ | $0.824_{\pm 0.008}$ | $0.686_{\pm 0.006}$ |
| UCCS(ours) | $\mathbf{0.877}_{\pm 0.009}$ | $\mathbf{0.743}_{\pm 0.013}$ ✓ | $\mathbf{0.880}_{\pm 0.011}$ | $\mathbf{0.725}_{\pm 0.022}$ ✓ | $\mathbf{0.890}_{\pm 0.008}$ | $\mathbf{0.737}_{\pm 0.014}$ ✓ |

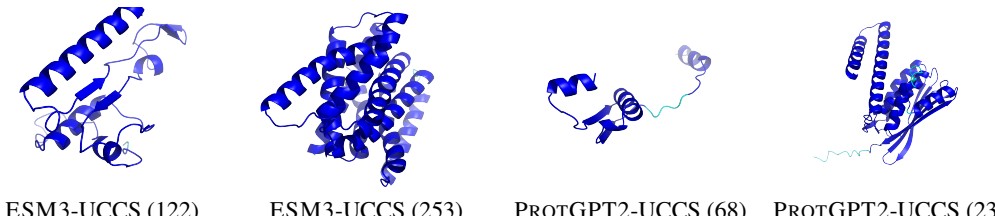

ESM3-UCCS (122)    ESM3-UCCS (253)    PROTGPT2-UCCS (68)    PROTGPT2-UCCS (236)

Figure 4: **Generation showcase of UCCS.** Unconditional protein generations with UCCS.[3] All cases exhibit high structural confidence (dominantly blue regions, $pLDDT > 90$) and diverse folds, highlighting reduced repetition and improved foldability compared to raw generations in Table 1.

## 5.2    MAIN RESULTS

Tables 2–10 and Figures 8–9 report results across both backbones (ESM3, PROTGPT2), datasets (CATH, UniRef50, SCOP), and tasks (unconditional vs. conditional generation). We evaluate repetition $R(x)$ and biological utility $U(x)$, marking $U \geq U_{\text{Orig}}$ with a checkmark. For brevity, the main text shows scatter plots for ESM3 (Figure 8), while full PROTGPT2 results are in Appendix H.1.

**UCCS achieves the best trade-off.** Our method consistently improves $R(x)$ while maintaining or enhancing $U(x)$ across all settings. As demonstrated in Table 10(a), on ESM3, UCCS yields large gains in $R(x)$ compared to the original model (e.g., $+55\%$ under unconditional generation on CATH), while preserving or even slightly improving $U(x)$ (see: Fig 4 and Appendix K for visualization cases). On PROTGPT2 (see Table 2(a)), UCCS achieves the highest repetition control (up to $+20\%$ relative to temperature sampling) and is the only method that consistently satisfies the utility constraint across datasets and tasks. These results confirm our central claim: dataset-guided contrastive steering offers a plug-and-play yet robust mechanism to reduce pathological repetition without compromising foldability. *Compared to other mechanism-level approaches, UCCS explicitly decouples utility from repetition when constructing the mean-difference vector(section 4), ensuring that the injected direction corresponds to a genuine repetition signal rather than entangled features. This alignment explains why UCCS provides stable gains in $R(x)$ without sacrificing $U(x)$.*

---

[2]*Notes.* $R$: repetition score; $U$: biological utility. We annotate cells with ✓ when $U \geq U_{\text{Orig}}$ (utility constraint satisfied). Decoding-level heuristics and original models do not require supervision; therefore their behavior remains consistent across datasets. In particular, the choice of dataset does not affect unconditional generation.

[3]Lengths are shown in parentheses after each case ID.

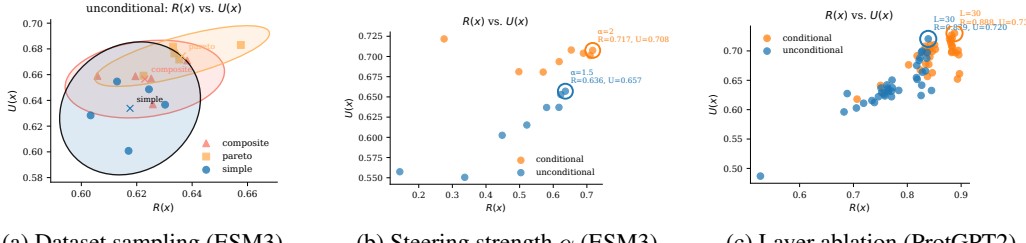

(a) Dataset sampling (ESM3)  (b) Steering strength $\alpha$ (ESM3)  (c) Layer ablation (ProtGPT2)

Figure 5: **Representative ablations.** (a) dataset sampling, (b) steering strength, (c) steering layer. Full results (both models, unconditional/conditional) in Appendix H.2.

**Decoding heuristics reduce repetition but harm foldability.** Sampling strategies such as increasing temperature or applying top-$p$ thresholds can raise $R(x)$ substantially (e.g., temperature sampling at $T = 1.3$, using entropy based sampling for ESM3, as shown in Figure 8(a)), but they consistently reduce $U(x)$, indicating worse structural reliability. The reason is that such heuristics operate only at the output distribution: they encourage **surface-level diversity** by flattening token probabilities, yet remain agnostic to the internal relationships among residues. As a result, higher temperatures often produce locally diverse but globally incoherent motifs, disrupting the residue patterns that underlie stable protein folds. This suggests that *token-level diversity does not automatically translate into structural plausibility*, and explains why decoding heuristics, while effective in NLP, can be detrimental in protein generation. In contrast, UCCS directly disentangles repetition from biological utility, steering generation along directions that preserve foldability while reducing degeneracy.

**Mechanism-level baselines are unstable.** Neuron deactivation and probe steering sometimes improve $R(x)$, but gains are inconsistent and often come at the cost of reduced $U(x)$. For instance, in the unconditional task on ESM3, neuron deactivation shows some promise: across all three datasets it moderately increases both $R$ and $U$ (see Figure 8(b–d)), suggesting that suppressing a small subset of high-activation neurons can reduce degeneracy in this setting. However, the same method performs poorly in the conditional task: on ESM3-SCOP, it lowers repetition and thus raises $R$, but simultaneously damages $U$, leading to worse overall trade-offs (Figure 8(e)). Probe steering is even less reliable: while occasionally yielding minor improvements, its effects fluctuate strongly across datasets and settings, offering no consistent pattern of benefit. These observations reinforce our claim that mechanism-level baselines, which manipulate isolated features without explicitly decoupling repetition from utility, lack robustness compared to dataset-guided contrastive steering.

### 5.3  ABLATION STUDY

To assess the contributions of different components, we conduct ablations along three axes: (i) contrastive dataset construction, (ii) steering strength $\alpha$, and (iii) injection layer $L$. Across both ESM3 and ProtGPT2 backbones, three consistent trends emerge: (1) structured dataset construction strategies (*Pareto* and *Composite*) outperform simple random sampling (see Fig. 5a, detailed in Fig. 10 and Tables 15–16 in Appendix H.2); (2) moderate steering strength ($\alpha \approx 1 \sim 2$) yields the most balanced trade-off (see Fig. 5b, with full results in Figs. 11–12 and Tables 17–18); and (3) effective steering typically arises in later layers, with the optimal range depending on model depth (see Fig. 5c, with per-layer curves and tables in Figs. 13–14 and Tables 21–20). In practice, Pareto/Composite sampling not only improves mean performance but also reduces variance across seeds, especially in the unconditional setting. For $\alpha$, both backbones show a unimodal trend: performance improves up to $\alpha \approx 1 \sim 2$, after which repetition control saturates while utility declines, with conditional sampling remaining more robust. For layer ablations, ESM3 benefits most from steering at the final 2–3 layers, while ProtGPT2 shows a smoother optimum in the last third of its stack, consistent with their architectural differences. Comprehensive visualizations and results are provided in Appendix H.2.

## 6  CONCLUSION

We conducted the first systematic study of pathological repetition in protein language models (PLMs) and introduced metrics framing it as a constrained optimization problem. To tackle this, we proposed **Utility-Controlled Contrastive Steering (UCCS)**, which disentangles repetition from structural utility via dataset-guided activation edits. Experiments across diverse PLMs and datasets show that UCCS mitigates degeneracy while preserving foldability, outperforming existing heuristics and interventions.

## ACKNOWLEDGMENT

Lijie Hu, Jiahao Zhang, and Zeqing Zhang are supported by the funding BF0100 from Mohamed bin Zayed University of Artificial Intelligence (MBZUAI). Di Wang is supported in part by the funding BAS/1/1689-01-01,RGC/3/7125-01-01, FCC/1/5940-20-05, FCC/1/5940-06-02, and King Abdullah University of Science and Technology (KAUST) – Center of Excellence for Generative AI, under award number 5940 and a gift from Google.

## ETHICS STATEMENT

This work does not involve human subjects, personally identifiable information, or sensitive user data. All datasets used (CATH, SCOP, UniRef50) are publicly available protein sequence databases. Generated sequences are synthetic and intended solely for methodological evaluation; they are not directly deployable for therapeutic or industrial applications.

## REPRODUCIBILITY STATEMENT

We have made every effort to ensure reproducibility. All datasets used are public and detailed pre-processing instructions are provided in Appendix F. Model configurations, hyperparameters, and evaluation metrics are fully documented in Appendices G.1 and E. Pseudocode for dataset construction and steering is included in Appendix F.4. All code, along with configuration files and scripts for reproducing figures and tables, will be released upon publication.

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

## A   ADDITIONAL RELATED WORK

To complement the main related work in Section 2, we provide additional discussions that span three interconnected areas: (i) explainable AI (XAI) techniques for interpreting and steering protein language models (PLMs); (ii) repetition and degeneration in natural language generation, including decoding and representational interventions.

**XAI for protein language model.**   Recent studies have begun to apply explainable AI (XAI) methods to uncover how protein language models (PLMs) encode biological knowledge. Attribution and attention-based analyzes demonstrate that PLMs highlight residues at functional motifs or binding sites, offering interpretable evidence of biologically meaningful signal capture (Sun et al., 2025). To probe model internals more systematically, researchers have leveraged sparse autoencoders to decompose PLM representations into human-interpretable features corresponding to structural or functional categories (Simon & Zou, 2024; Parsan et al., 2025), and neuron-level studies reveal "knowledge neurons" associated with enzyme classes and other biochemical properties (Nori et al., 2023). Beyond post-hoc interpretation, intervention-based strategies are emerging: recent work shows that editing activations can steer PLMs toward desired sequence properties (Huang et al., 2025), while concept bottleneck architectures introduce interpretable intermediate variables that enable controllable protein design (Ismail et al., 2025). These advances illustrate a growing toolkit for interpreting, probing, and steering PLMs, laying the groundwork for more transparent and biologically informed protein design.

**Repetition and degeneration in NLP generation.**   While not discussed in detail in the main text, repetition has long been recognized as a failure mode in natural language generation. Here, we provide a more focused overview of this literature, which informs our approach to repetition control in protein sequence generation. Repetition in large language models often leads to low-entropy loops or degenerate text with reduced diversity (Holtzman et al., 2020; Welleck et al., 2020). To mitigate these issues, the NLP community has explored decoding strategies such as top-$k$ or nucleus sampling, $n$-gram blocking, and temperature scaling (Graves, 2014; Fan et al., 2018; Kulikov et al., 2019). More recent work investigates internal mechanisms of repetition and proposes neuron-level interventions (Yao et al., 2025). Yet these techniques were developed in the context of natural language, where repetition mainly reduces readability. In proteins, by contrast, uncontrolled repetition can collapse structural folding and render sequences biologically non-functional. While most NLP work has focused on decoding-level strategies, another line of research investigates steering generation through internal representations, which inspires our approach.

## B   NOTATION

We summarize here the notations used throughout the paper.

## C   REPETITION IN PLM: EMPIRICAL OBSERVATION

### C.1   RAW POS VS. PLM RAW NEG

As previewed in Table 1 of the Introduction, here we provide a more detailed analysis of raw positive pool $\mathbf{P}^+$ (natural proteins) and negative pool $\mathbf{P}^-$ (PLM-generated proteins). Natural proteins exhibit rich compositional balance and non-trivial motif structures, while PLM generations frequently collapse into homopolymers (e.g., long stretches of alanine or glycine) or short repetitive motifs. Structural prediction with AlphaFold further reveals that natural proteins achieve high pLDDT confidence, whereas PLM outputs often result in unstable folds with poor confidence, confirming their limited biological plausibility.

### C.2   TOKEN FREQUENCY DISTRIBUTIONS: PLM VS. NATURAL PROTEINS

To further support our observation of degeneracy in PLM outputs, we compare the overall amino acid frequency distributions between natural proteins ($\mathbf{P}^+$) and PLM-generated sequences ($\mathbf{P}^-$).

| Symbol | Description |
|---|---|
| *Basic objects* | |
| $\mathcal{A}$ | Amino acid alphabet |
| $x = (x_1, \ldots, x_T)$ | Protein sequence of length $T$, with $x_t \in \mathcal{A}$ |
| $T$ | Sequence length |
| $p(a)$ | Empirical unigram distribution of residue $a \in \mathcal{A}$ in $x$ |
| *Models and generation* | |
| $M$ | Pretrained protein language model (PLM) |
| $M(x_t \mid x_{\backslash t})$ | Prediction of a masked token (MLM setting) |
| $M(x_t \mid x_{<t})$ | Prediction of the next token (AR setting) |
| $p$ | Prefix used for conditional generation |
| $f$ | Modification function applied to $M$ for repetition control |
| *Optimization problem* | |
| $\epsilon$ | Tolerance for utility degradation |
| Objective | $\min_f R(f(M,p))$ s.t. $U(f(M,p)) \geq U(M,p) - \epsilon$ |
| *Metrics* | |
| $R(x)$ | Repetition score integrating $H_{\mathrm{norm}}(x)$, Distinct-2/3, and $R_{\mathrm{hpoly}}(x)$ |
| $U(x)$ | Utility score combining $\mathrm{pLDDT}(x)$ and $\mathrm{pTM}(x)$ |
| *Representations and steering* | |
| $h_t^L(x) \in \mathbb{R}^d$ | Hidden representation at layer $L$ for token $x_t$ |
| $\phi^L(x)$ | Sequence-level summary of hidden states (mean for MLM, last token for AR) |
| $v^L$ | Contrastive steering vector, $\mathbb{E}_{x \in \mathcal{D}^-}[\phi^L(x)] - \mathbb{E}_{x \in \mathcal{D}^+}[\phi^L(x)]$ |
| $\alpha$ | Steering strength coefficient |
| $\tilde{h}_t^L(x)$ | Steered hidden state: $h_t^L(x) + \alpha v^L$ |
| *Datasets and construction* | |
| $\mathcal{P}^+, \mathcal{P}^-$ | Raw positive (low-repetition) and negative (high-repetition) candidate pools |
| $\mathcal{D}^+, \mathcal{D}^-$ | Filtered contrastive datasets from $\mathcal{P}^+, \mathcal{P}^-$ |
| $\Delta R, \Delta U$ | Differences in repetition/utility between $\mathcal{D}^+$ and $\mathcal{D}^-$ |

Table 3: Summary of notation used in the paper.

Figure 6 presents grouped bar charts of amino acid compositions, where the horizontal axis enumerates amino acid types (standard 20 amino acids, represented by their single-letter codes), and, for each amino acid, different colored bars correspond to different datasets (POS vs. NEG). Natural proteins show a relatively balanced usage of amino acids across different physicochemical categories, while PLM generations are strongly skewed. In particular, in ESM3 outputs **alanine (A)** accounts for more than 40% of residues and **leucine (L)** is also overrepresented, whereas in ProtGPT2 outputs **alanine (A)** and **proline (P)** are highly enriched. This compositional collapse corroborates the repetition phenomena reported in the main text, and motivates the definition of our repetition-related evaluation metrics ($H_{\mathrm{norm}}$, Distinct-$n$, $R_{\mathrm{hpoly}}$).

# D  STRUCTURAL UTILITY METRICS

## D.1  PER-RESIDUE LDDT AND PLDDT.

Following AlphaFold (Jumper et al., 2021), we measure local structural accuracy using the (non–superposition-based) local Distance Difference Test (lDDT) (Mariani et al., 2013) score. For a residue $i$, let $\mathcal{N}_i$ be the set of residues whose $C_\alpha$ atoms lie within a cutoff $r_{\mathrm{nb}}$ in the *reference* structure (and $|i - j| > \Delta$ to ignore trivial neighbors along the chain). Let $d_{ij}$ and $d_{ij}^\star$ denote the predicted and reference inter-residue distances. The lDDT at residue $i$ averages tolerance tests over

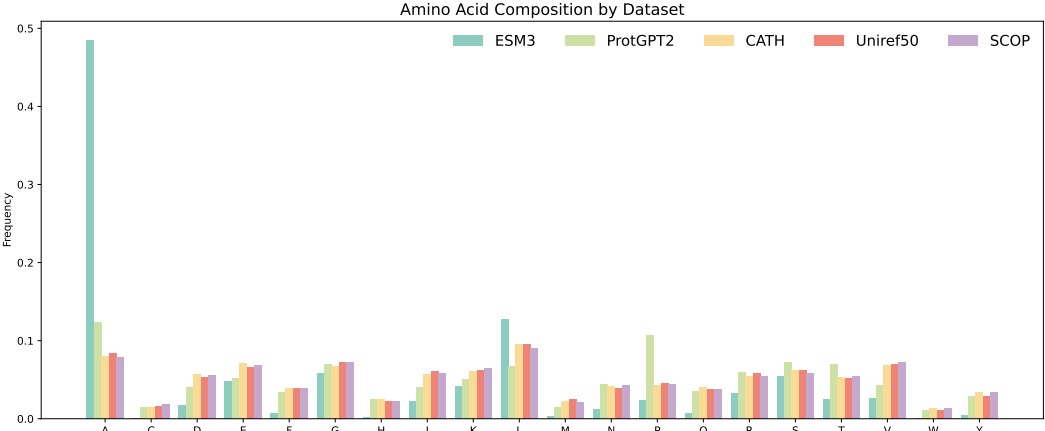

Figure 6: Grouped bar charts of amino acid frequency distributions across natural proteins (CATH, Uniref50, SCOP) and PLM-generated proteins (ESM3, ProtGPT2). Natural datasets show balanced usage across amino acids, whereas PLM generations are dominated by a few residues (e.g., alanine (A) and leucine (L) in ESM3; alanine (A) and proline (P) in ProtGPT2).

a set of thresholds $\mathcal{T} = \{0.5, 1.0, 2.0, 4.0\}$ Å:

$$\text{lDDT}_i = \frac{1}{|\mathcal{N}_i| \, |\mathcal{T}|} \sum_{j \in \mathcal{N}_i} \sum_{\tau \in \mathcal{T}} \mathbf{1}\big(|d_{ij} - d_{ij}^\star| < \tau\big).$$

The chain-level lDDT is the mean over residues, $\text{lDDT} = \frac{1}{L} \sum_i \text{lDDT}_i$. AlphaFold's *predicted* lDDT (pLDDT) is obtained by a confidence head that outputs a categorical distribution over binned lDDT values; pLDDT is the (scaled) expectation:

$$\text{pLDDT}_i = 100 \cdot \mathbb{E}_{q_\theta(\text{lDDT}_i)}[\text{lDDT}_i], \qquad \text{pLDDT} = \frac{1}{L} \sum_{i=1}^{L} \text{pLDDT}_i \in [0, 100].$$

### D.2 GLOBAL TOPOLOGY TM-SCORE AND pTM.

Global fold accuracy is summarized by the TM-score (Zhang & Skolnick, 2004) for an optimal alignment $\mathcal{A}$ between predicted and reference structures:

$$\text{TM}(\mathcal{A}) = \frac{1}{L} \sum_{i=1}^{L} \frac{1}{1 + \big(\frac{d_i(\mathcal{A})}{d_0(L)}\big)^2}, \qquad d_0(L) = 1.24 \sqrt[3]{L - 15} - 1.8,$$

where $d_i(\mathcal{A})$ is the aligned $C_\alpha$ distance at residue $i$, and $d_0(L)$ is the TM normalization. AlphaFold's *predicted* TM-score (pTM) uses the *predicted alignment error* (PAE), i.e., a distribution over pairwise alignment deviations. Let $\mathbb{E}[d_i]$ denote the expected aligned error at residue $i$ under the PAE. Then pTM is an expectation/relaxation of TM-score:

$$\text{pTM} \approx \max_{\mathcal{A}} \frac{1}{L} \sum_{i=1}^{L} \frac{1}{1 + \big(\frac{\mathbb{E}[d_i(\mathcal{A})]}{d_0(L)}\big)^2} \in [0, 1].$$

Intuitively, pLDDT captures *local* geometric confidence while pTM captures *global* topology confidence.

### D.3 OUR OPERATIONAL UTILITY.

In our experiments we use

$$U(x) = \tfrac{1}{2}\Big(\text{pLDDT}(x) \, / \, 100 \, + \, \text{pTM}(x)\Big),$$

i.e., the mean of (scaled) pLDDT and pTM, to summarize both local and global structural plausibility for a generated sequence $x$.

# E   METRIC DETAILS

This appendix provides detailed definitions and motivations for the metrics used throughout the paper. While the main text only presents their compact forms (Section 4), here we explain the construction rationale and discuss why they provide reliable signals for evaluating repetition and biological plausibility in protein sequences.

## E.1   REPETITION SCORE CONSTRUCTION

We define the overall repetition score $R(x)$ for a protein sequence $x$ as a normalized aggregation of three complementary measures:

$$R(x) = \tfrac{1}{3}\Big(H_{\mathrm{norm}}(x) + \tfrac{1}{2}\big(\text{Distinct-2}(x) + \text{Distinct-3}(x)\big) + R_{\mathrm{hpoly}}(x)\Big).$$

**Entropy ($H_{\mathrm{norm}}$).**   Normalized unigram entropy captures the global residue distribution. Low entropy indicates collapse into a small subset of amino acids, a hallmark of low-complexity sequences.

$n$**-gram diversity (Distinct-$n$).**   Distinct-2/3 measures local motif diversity. Recurrent dipeptide or tripeptide fragments (e.g., AGAGAG, PQAPQA) lead to low Distinct-$n$, signaling motif-level repetition.

**Homopolymer diversity ($R_{\mathrm{hpoly}}$).**   This score penalizes runs of length $\geq 4$ for any residue, detecting extended homopolymerization (e.g., AAAAAA). Such runs are a canonical form of pathological collapse in PLM generations.

**Why an average?**   Each component captures a different failure mode: entropy (global collapse), Distinct-$n$ (motif recurrence), and $R_{\mathrm{hpoly}}$ (long runs). By averaging, we obtain a balanced metric that is sensitive to diverse artifacts without overemphasizing any single mode. Empirically, we observed that sequences with high $R(x)$ align with natural proteins, while PLM-generated sequences with low $R(x)$ exhibit visible collapse patterns (see Appendix F.2).

## E.2   UTILITY SCORE CONSTRUCTION

To measure biological plausibility, we define the utility score as

$$U(x) = \tfrac{1}{2}\big(\mathrm{pLDDT}(x)/100 + \mathrm{pTM}(x)\big),$$

where pLDDT is the residue-level local confidence, and pTM estimates global fold topology.

**Why combine pLDDT and pTM?**   Relying on a single metric can be misleading:

- pLDDT is highly sensitive to local residue environments but blind to global misfolding.
- pTM captures overall fold similarity but may overestimate local confidence.

By averaging them, we obtain a score that balances local accuracy and global topology.

## E.3   SUMMARY

In summary:

- The **repetition score** $R(x)$ integrates entropy, motif-level diversity, and homopolymer penalties, offering a comprehensive view of pathological repetition.
- The **utility score** $U(x)$ combines local (pLDDT) and global (pTM) structure metrics, providing a balanced estimate of biological plausibility.

Together, these scores allow us to construct contrastive datasets that differ primarily along the repetition dimension while remaining matched in structural plausibility, enabling effective application of our proposed Utility-Controlled Contrastive Steering (UCCS).

# F    DATASET CONSTRUCTION DETAILS

We describe the preprocessing steps, filtering rules, and sampling strategies used to construct the datasets evaluated in Section 5.

## F.1    DETAILS OF RAW DATASET

### F.1.1    POSITIVE DATASET

We randomly sample 10,000 sequences from widely used protein databases including **CATH (Knudsen & Wiuf, 2010)**(NR20 Non-redundant Subset), **UniRef50 (Consortium, 2024)**, an **SCOP (Murzin et al., 1995)**. All sequences are annotated along two dimensions: (1) *Repetition*, measured by $R(x)$ and its components (see Section 3.3), and (2) *Biological Utility*, measured by $U(x)$, combining $\text{pLDDT}(x)$ and $\text{pTM}(x)$ (Appendix D).

### F.1.2    NEGATIVE DATASET

Negative examples are *synthetic* rather than natural proteins. Two independent generation procedures were used:

- **ESM-3 unconditional sampling**: sequences are generated by unconditional sampling from the pretrained *ESM-3* model using the official inference client, with random target lengths ($50 \leq T \leq 1024$).

- **ProtGPT2 autoregressive sampling**: sequences are generated using the ProtGPT2 language model, from an empty prefix, with random target lengths ($50 \leq T \leq 1024$).

Together, these sources provide a diverse negative pool with low or ambiguous biological plausibility. Filtering ensures all negatives pass minimal length and pLDDT constraints while exhibiting low entropy. **Notably, the synthetic negatives tend to display much higher repetition levels than natural positives**, making them complementary counterparts for balanced contrastive evaluation.

## F.2    STATISTICS

To ensure that our repetition and structure metrics are biologically meaningful rather than ad-hoc definitions, we compute them on both natural proteins (positive sets $\mathbf{P}^+$) and PLM-generated sequences (negative sets $\mathbf{P}^-$). This analysis serves a dual purpose: (i) characterizing the distributional properties of our datasets, and (ii) validating that $H_{\text{norm}}$, Distinct-2/3, $R_{\text{hpoly}}$, and structure-based scores ($\text{pLDDT}$, $\text{pTM}$) reliably distinguish repetitive, degenerate PLM outputs from diverse, foldable natural proteins.

**Tabular Summary.**    We report the following metrics consistently with Appendix B:

- $H_{\text{norm}}(x)$: normalized entropy,

- Distinct-2/3: proportion of unique $n$-grams,

- $R_{\text{hpoly}}(x)$: 4-gram polynomial entropy,

- $\text{pLDDT}(x)$ and $\text{pTM}(x)$: structural confidence scores.

**Density distributions.**    For each metric, we plot kernel density estimates (KDEs) for both positive (natural proteins) and negative (PLM generations) datasets. To enable comparison across disparate scales, we report *relative density*, i.e., each KDE is normalized such that its maximum equals 1. These visualizations highlight clear contrasts between natural and synthetic sequences: (i) repetition metrics ($H_{\text{norm}}$, Distinct-2/3, $R_{\text{hpoly}}$; see Figure 1) indicate that negatives exhibit higher redundancy and lower lexical diversity; and (ii) structure utilities ($\text{pLDDT}$, $\text{pTM}$; see Figure 7) cleanly separate natural sequences—shifted toward high-confidence regions—from PLM outputs with degraded or unstable folds.

Table 4: mean and variance of repetition and bio-utility metrics across pos (natural proteins) and neg (plm generations).

| metric | ESM (neg) | ProtGPT2 (neg) | CATH (pos) | Uniref50 (pos) | SCOP (pos) |
|---|---|---|---|---|---|
| $H_{\text{norm}}$ | $0.436 \pm 0.149$ | $0.711 \pm 0.182$ | $0.917 \pm 0.041$ | $0.926 \pm 0.042$ | $0.924 \pm 0.036$ |
| Distinct-2 | $0.144 \pm 0.092$ | $0.389 \pm 0.243$ | $0.760 \pm 0.111$ | $0.612 \pm 0.150$ | $0.727 \pm 0.125$ |
| Distinct-3 | $0.272 \pm 0.156$ | $0.625 \pm 0.313$ | $0.974 \pm 0.031$ | $0.949 \pm 0.045$ | $0.973 \pm 0.024$ |
| $R_{\text{hpoly}}$ | $0.516 \pm 0.252$ | $0.939 \pm 0.128$ | $0.994 \pm 0.022$ | $0.996 \pm 0.016$ | $0.999 \pm 0.006$ |
| pLDDT (0–100) | $63.45 \pm 19.75$ | $75.51 \pm 20.83$ | $89.94 \pm 8.56$ | $90.14 \pm 10.15$ | $91.62 \pm 9.22$ |
| pTM | $0.427 \pm 0.194$ | $0.514 \pm 0.269$ | $0.763 \pm 0.154$ | $0.767 \pm 0.197$ | $0.811 \pm 0.146$ |

*notes.* pos: natural proteins; neg: plm generations. repetition metrics (e.g., $H_{\text{norm}}$, distinct-$n$, $R_{\text{hpoly}}$) consistently separate pos vs neg, validating their effectiveness.

**Interpretation of Results.** The quantitative results in Table 4 reinforce the patterns observed in the density plots. Synthetic negatives generated by PLMs (ESM-3, ProtGPT2) exhibit substantially lower values across all repetition metrics—including $H_{\text{norm}}$, Distinct-2/3, and $R_{\text{hpoly}}$—indicating higher redundancy and reduced sequence diversity. In contrast, natural datasets (CATH, UniRef50, SCOP) consistently score higher on entropy and diversity, reflecting richer lexical and structural complexity.

From a structural perspective, natural proteins achieve significantly higher pLDDT and pTM scores, reflecting greater foldability and evolutionary plausibility. Meanwhile, synthetic negatives not only show lower means but also greater variance, suggesting instability and reduced structural confidence.

Figure 1 demonstrates that our repetition metrics not only distinguish natural versus synthetic proteins, but also reveal distinct *failure modes* across PLM architectures. $H_{\text{norm}}$ globally separates both PLMs from natural proteins, confirming the presence of pathological repetition in general. Distinct-2/3 expose local motif-level redundancy specific to autoregressive models like ProtGPT2 ("AGA-GAG...") patterns), whereas $R_{\text{hpoly}}$ captures long-run homopolymer collapse ("AAAAAA...") unique to masked LMs like ESM3. Overall, both PLMs diverge sharply from natural proteins in global repetition metrics, underscoring that pathological repetition is a shared phenomenon manifested differently across model classes.

### F.3 DETAILS OF THE RANKING ALGORITHM

**Defining metrics.** In our dataset construction, we adopt the **repetition score** $R(x)$ as the primary metric for distinguishing between positive and negative samples. Unlike raw repetition measures (e.g., homopolymer runs), the repetition score is a composite metric that integrates three complementary components:

- Normalized sequence entropy $H_{\text{norm}}(x)$;
- Average $n$-gram diversity $\frac{1}{2}(\text{Distinct-2}(x) + \text{Distinct-3}(x))$;
- Homopolymer diversity score $R_{\text{hpoly}}^{(k=4)}(x)$, which penalizes runs of length $\geq 4$.

Formally,

$$R(x) = \tfrac{1}{3}\Big(H_{\text{norm}}(x) + \tfrac{1}{2}\big(\text{Distinct-2}(x) + \text{Distinct-3}(x)\big) + R_{\text{hpoly}}(x)\Big).$$

A higher $R(x)$ indicates greater sequence diversity and reduced redundancy, which aligns with biological plausibility. Positive samples are therefore expected to have *higher* $R(x)$, while negatives tend to exhibit low diversity and strong repetitive patterns.

The **utility score** $U(x)$ reflects structural plausibility, computed as:

$$U(x) = \tfrac{1}{2}\big(\text{pLDDT}(x)/100 + \text{pTM}(x)\big),$$

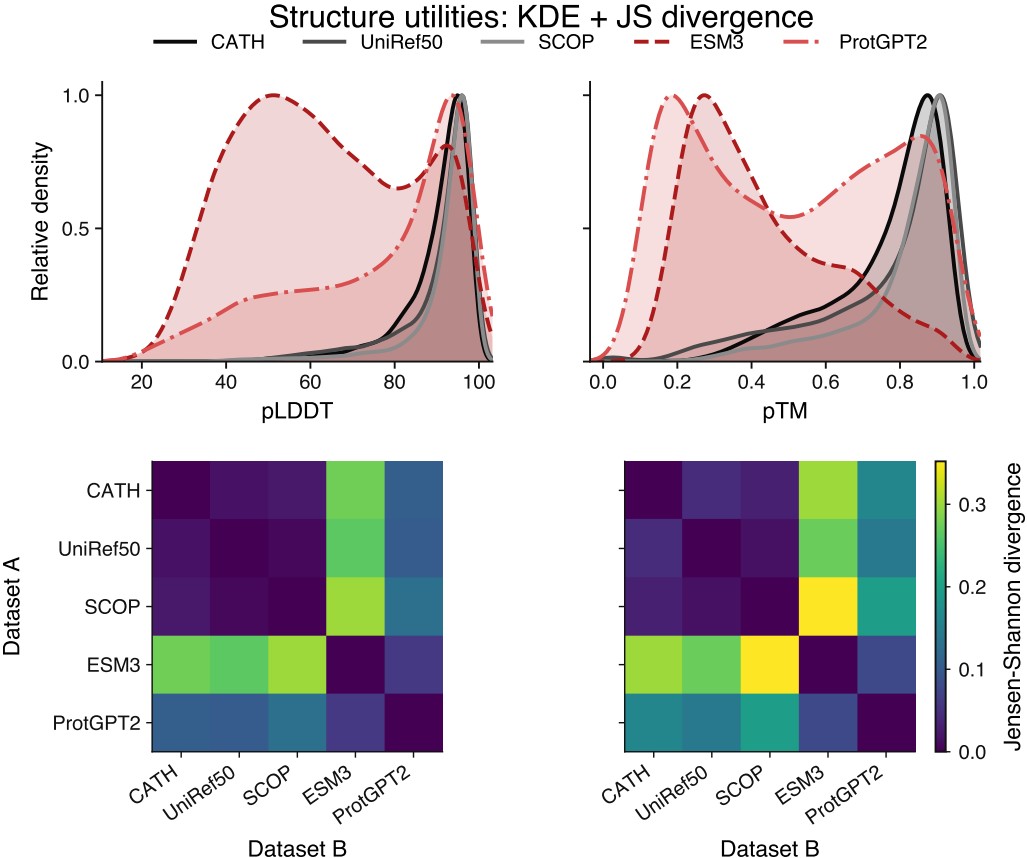

Figure 7: **Structure utilities: KDE + JS divergence.** Top: Kernel density estimates (relative density, peak = 1) of structure-based metrics (pLDDT, pTM) across natural (CATH, UniRef50, SCOP) and PLM-generated (ESM3, ProtGPT2) datasets. Bottom: Pairwise Jensen–Shannon divergence heatmaps between datasets. High divergence between PLM and natural sets confirms that our structure metrics ($U(x)$) effectively separate foldable natural proteins from unstable PLM generations. Natural datasets cluster tightly, while PLM sets diverge—indicating reduced structural reliability. Together with repetition metrics, these results demonstrate that our metric suite captures both structural and lexical degradation in PLM outputs.

where pLDDT reports residue-level local confidence and pTM measures fold-level topology (Jumper et al., 2021).

### F.3.1 LENGTH AND BIO-UTILITY BINNING

Before ranking, we bucketize sequences along two orthogonal axes:

1. **Length bins**: determined by pre-defined thresholds (50, 129, 257, 513, 1025). Each sequence is assigned to the corresponding length interval.

2. **Quantile-based bio-utility bins**: within each length bin, sequences are further split into $Q$ bins by quantiles of $U(x)$ (using either pTM or pLDDT depending on availability).

This two-level bucketing ensures that positives and negatives are compared on matched subsets, removing trivial confounders such as sequence length.

### F.3.2 Pareto-based Selection

We use an approximate two-objective selection that operates *within each* length–utility bin. Let $s = +1$ for positive candidates and $s = -1$ for negative candidates. Inside the current bin, we robustly scale scores using

$$\text{rs}_{\mathcal{B}}(z) = \frac{z - \text{median}_{x \in \mathcal{B}} z(x)}{\max\{\text{IQR}_{x \in \mathcal{B}} z(x), \, \varepsilon\}},$$

with a small $\varepsilon > 0$ for stability. Define two oriented objectives:

$$A(x) = s \cdot \text{rs}_{\mathcal{B}}\big(R(x)\big) \quad \text{(repetition-oriented)}, \qquad B(x) = -\text{rs}_{\mathcal{B}}\Big(|U(x) - \bar{U}_{\text{ref}}|\Big) \quad \text{(utility alignment)}.$$

Here $\bar{U}_{\text{ref}}$ is the mean $U(\cdot)$ of the *opposite* side in the same bin.

**Front extraction (bin quota $q$).** Rather than running a full nondominated sort, we apply a single-pass frontier approximation: (i) sort all candidates in the bin by the pair $(A(x), B(x))$ in descending lexicographic order; (ii) scan once in that order, keeping any point whose $B(x)$ is within $\epsilon$ of the running maximum $B_{\max}$ (updating $B_{\max}$ as we go), which yields a high-$B$ envelope under high $A$; (iii) if the envelope exceeds the quota, keep the top $q$ by $A(x) + B(x)$; if it falls short, fill from the remaining candidates in sort order until reaching $q$. After selecting the positive side, we update $\bar{U}_{\text{ref}}$ using the selected positives and then select the negatives (and vice versa). This procedure is repeated until per-bin quotas are satisfied.

### F.3.3 Composite Score Selection

As a simpler alternative, we form a weighted composite score:

$$S(x) = \alpha \cdot (\pm \hat{R}(x)) + \beta \cdot \hat{G}(x),$$

where $\hat{R}(x)$ is the robustly scaled repetition score, $\hat{G}(x)$ is the scaled deviation term $-|U(x) - \bar{U}_{\text{ref}}|$, and $(\alpha, \beta)$ balance the trade-off. This produces a single scalar ranking per bin.

### F.3.4 Baseline: Random Sampling

We also consider random baselines:

- **Simple random**: global uniform sampling, ignoring bins;

### F.3.5 Summary

In summary, ranking enforces matched distributions in length and bio-utility, while maximizing contrast in repetition. The Pareto strategy provides a principled multi-objective approximation, the composite score offers a tunable scalarization, and random sampling serves as a control. We report dataset-level statistics and violin plots of the three repetition components and two bio-utility metrics in Section 5.

### F.4 Pseudo Algorithm of Dataset Construction

For completeness, we provide pseudocode for the dataset construction procedure described in Section 4.2. This algorithm implements the optimization objective

$$(\mathcal{D}^+, \mathcal{D}^-) = \arg \max_{\mathcal{D}^+, \mathcal{D}^-} \Delta R \quad \text{s.t.} \quad \Delta U \leq \epsilon,$$

with multiple selection strategies (Pareto frontier, composite scoring, and random selection as baseline).

## G Implementation Details and Hyperparameters

This section provides additional details on model setups, baseline implementations, and hyperparameter choices.

---

**Algorithm 1** Dataset Construction for Contrastive Steering

---

1: **Inputs:**
- Raw positive pool $\mathcal{P}^+$ (low repetition, unconstrained utility)
- Raw negative pool $\mathcal{P}^-$ (high repetition, unconstrained utility)
- Repetition scorer $R(\cdot)$, Utility scorer $U(\cdot)$
- Tolerance $\epsilon$, target sizes $n_+, n_-$

2: **Outputs:** $\mathcal{D}^+$ (filtered positives), $\mathcal{D}^-$ (filtered negatives)

3: Compute $(R(x), U(x))$ for all sequences in $\mathcal{P}^+ \cup \mathcal{P}^-$
4: Filter out sequences with $|U(x) - \bar{U}| > \epsilon$
5: Partition candidates into bins $\mathcal{B}$ by sequence length and bio-utility quantiles
6: **if** method = Pareto **then**
7:     **for** each bin $\mathcal{B}$ with quota $k$ **do**
8:         Scale $R(x), U(x)$ within $\mathcal{B}$
9:         Select positives maximizing $R(x)$ while aligning $U(x)$ to $\bar{U}_{\text{ref}}^{(+)}$ (Pareto frontier)
10:         Update $\bar{U}_{\text{ref}}^{(-)}$
11:         Select negatives minimizing $R(x)$ while aligning $U(x)$ to $\bar{U}_{\text{ref}}^{(-)}$ (Pareto frontier)
12:     **end for**
13: **else if** method = Composite **then**
14:     **for** each bin $\mathcal{B}$ with quota $k$ **do**
15:         Scale $R(x), U(x)$ within $\mathcal{B}$
16:         Positives: score $S(x) = +\alpha R(x) - \beta |U(x) - \bar{U}_{\text{ref}}^{(+)}|$, take top $k$
17:         Update $\bar{U}_{\text{ref}}^{(-)}$
18:         Negatives: score $S(x) = -\alpha R(x) - \beta |U(x) - \bar{U}_{\text{ref}}^{(-)}|$, take top $k$
19:     **end for**
20: **else**                                                 ▷ Random baseline
21:     Uniformly sample $n_+$ positives and $n_-$ negatives from global pools
22: **end if**
23: **Return:** $\mathcal{D}^+, \mathcal{D}^-$

---

Table 5: hyperparameter settings for ESM-3 sequence generation.

| parameter | default value |
|---|---|
| schedule (`schedule`) | `cosine` |
| unmasking strategy (`strategy`) | `random` |
| decoding steps (`num_steps`) | 20 |
| temperature (`temperature`) | 1.0 |
| temperature annealing (`temperature_annealing`) | `True` |
| top-$p$ (`top_p`) | 1.0 |

### G.1   MODEL CONFIGURATIONS

We summarize the default configurations for ESM-3 (masked LM) and PROTGPT2 (autoregressive LM). Unless otherwise specified, all other hyperparameters follow the official implementations.

### G.1.1   ESM-3

**Sequence generation.** The default iterative unmasking settings are given in Table 5.

**Structure folding.** For structure prediction tasks, we vary the number of iterative steps depending on usage (Table 6).

Table 6: settings for structure folding with ESM-3.

| usage | value of num_steps |
|---|---|
| dataset construction | 20 |
| main experiments | 8 |

Table 7: hyperparameter settings for PROTGPT2 sequence generation.

| parameter | default value |
|---|---|
| temperature (temperature) | 1.0 |
| top-$p$ (top_p) | 1.0 |
| repetition penalty (repetition_penalty) | 1.0 |
| no-repeat $n$-gram size (no_repeat_ngram_size) | 0 |

### G.1.2 PROTGPT2

**Sequence generation.** We use HuggingFace's `generate()` API with the parameters listed in Table 7.

To strictly control sequence length, we set

$$\texttt{max\_length} = \texttt{min\_length} = \text{target sequence length}.$$

## G.2 DECODING STRATEGY BASELINES

### G.2.1 ESM DECODING STRATEGY

Unless otherwise noted, parameters follow Appendix G.1.1, with only the sampling policy varied.

**Generative unmasking (default).** Random positions are unmasked with annealed temperature:

$$\texttt{schedule} = \texttt{cosine}, \quad \texttt{strategy} = \texttt{random}, \quad \texttt{temperature\_annealing} = \texttt{True}.$$

**Entropy-based unmasking.** Lowest-entropy positions are unmasked first. To isolate this effect, annealing is disabled:

$$\texttt{schedule} = \texttt{cosine}, \quad \texttt{strategy} = \texttt{entropy}, \quad \texttt{temperature\_annealing} = \texttt{False}.$$

**Temperature sampling.** We sweep the sampling temperature:

$$\tau \in \{0.7, 1.0, 1.3\}.$$

**Top-$p$ sampling.** We vary the nucleus probability cutoff:

$$p \in \{0.80, 0.85, 0.90, 0.95, 0.98, 1.00\}.$$

### G.2.2 EFFECT OF DECODING STEPS ON REPETITION

To examine whether the observed repetition behavior of ESM-3 arises from an insufficient number of decoding iterations, we conducted an ablation varying the number of unmasking steps. Note that ESM-3 performs *parallel unmasking* rather than stepwise autoregressive decoding; the default configuration of 20 steps follows the official recommendation balancing quality and efficiency. We nonetheless tested a wide range of decoding depths to verify the robustness of this choice.

Specifically, we varied the number of iterative unmasking steps

$$\texttt{num\_steps} \in \{10, 20, 40, 100, 200\},$$

under unconditional generation (target length = 200 aa, 100 samples, 5 random seeds). Table 8 reports the mean and standard deviation of structure and diversity metrics.

Table 8: Ablation on the number of decoding steps (`num_steps`) for ESM-3 sequence generation. Repetition-related metrics ($H_{norm}$, Distinct-2/3, $R_{hpoly}$) remain stable, indicating that repetition is intrinsic to the model rather than caused by limited decoding depth.

| Steps | pLDDT | pTM | $H_{norm}$ | Distinct-2 | Distinct-3 | $R_{hpoly}$ |
|-------|-------|-----|------------|------------|------------|-------------|
| 10 | 80.65±12.95 | 0.607±0.177 | 0.454±0.142 | 0.212±0.086 | 0.371±0.160 | 0.559±0.237 |
| 20 | 80.41±13.89 | 0.581±0.187 | 0.442±0.164 | 0.217±0.103 | 0.373±0.182 | 0.557±0.270 |
| 40 | 82.42±12.39 | 0.598±0.179 | 0.429±0.153 | 0.198±0.089 | 0.344±0.167 | 0.555±0.279 |
| 100 | 83.55±13.30 | 0.628±0.204 | 0.460±0.193 | 0.222±0.125 | 0.381±0.220 | 0.560±0.305 |
| 200 | 80.30±14.99 | 0.608±0.194 | 0.454±0.199 | 0.221±0.125 | 0.381±0.228 | 0.563±0.302 |

Table 9: Composite scores versus decoding steps.

| Steps | Repetition Score | Biological Utility |
|-------|------------------|--------------------|
| 10 | 0.435 | 0.707 |
| 20 | 0.431 | 0.692 |
| 40 | 0.419 | 0.711 |
| 100 | 0.440 | 0.732 |
| 200 | 0.439 | 0.706 |

**Summary.** Across decoding depths, repetition metrics remain nearly constant, while structure quality (pLDDT, pTM) varies only within normal statistical noise. The composite repetition–utility scores (Table 9) show no consistent improvement as the number of steps increases, confirming that repetition is not primarily determined by decoding depth.

Overall, increasing the number of unmasking iterations does not meaningfully reduce repetition, suggesting that the observed degeneracy is an inherent property of the pretrained model rather than a consequence of limited decoding steps.

### G.2.3 PROTGPT2 DECODING STRATEGY

PROTGPT2 is an autoregressive protein LM. The official example command includes a non-trivial repetition penalty (`repetition_penalty=1.2`):

```
sequences = protgpt2(
    "<|endoftext|>", max_length=100, do_sample=True,
    top_k=950, repetition_penalty=1.2,
    num_return_sequences=10, eos_token_id=0)
```

Our goal is to study repetition *without* decoding-specific heuristics. Thus, we set `repetition_penalty=1.0` in all baselines and compare standard decoding methods:

**Top-$p$ (nucleus) sampling.** We truncate to the top cumulative probability mass $p$, sweeping

$$p \in \{0.80, 0.85, 0.90, 0.95, 0.98, 1.00\}.$$

**Temperature sampling.** We vary the softmax temperature:

$$\tau \in \{0.7, 1.0, 1.3\}.$$

**Repetition penalty (for comparison).** For completeness, we also include ablations with non-trivial repetition penalties:

$$\texttt{repetition\_penalty} \in \{1.1, 1.2, 1.3\}.$$

These settings are reported only for comparison and are *not* part of our main baselines.

**$n$-gram blocking.** We additionally evaluate $n$-gram blocking, where repeated $n$-grams are disallowed during generation:

$$n \in \{2, 3, 4\}.$$

### G.3  MECHANISM INTERPRETABILITY BASELINES

We provide implementation details for two mechanism-level baselines: *Neuron-based Deactivation* and *Probe-based Steering*. Both operate on sequence-level hidden activations $h^L(x) \in \mathbb{R}^d$ of a transformer language model, where $L$ denotes the layer index and $d$ the hidden dimension. For masked language models (MLMs), $h^L(x)$ is obtained via mean pooling across tokens, while for autoregressive models (AR-LMs) we use the last-token embedding (see Appendix B).

#### G.3.1  NEURON-BASED DEACTIVATION

**Definition of a neuron.**  At layer $L$, the activation for a sequence $x$ is $h^L(x) \in \mathbb{R}^d$. A *neuron* corresponds to a single coordinate $j \in \{1, \ldots, d\}$, i.e. the scalar value $h_j^L(x)$.

**Supervision signal.**  We construct contrastive sets $\mathcal{D}^+$ (low repetition) and $\mathcal{D}^-$ (high repetition). For each sequence $x$, we compute the normalized entropy $H_{\text{norm}}(x)$ (Section 4.1; Appendix B) as a scalar supervision signal: lower entropy indicates higher repetition.

**Scoring.**  For every layer $L$ and neuron $j$, we collect activations $\{h_j^L(x)\}$ across sequences, align them with supervision values $\{H_{\text{norm}}(x)\}$, and compute the Pearson correlation coefficient:

$$s^{(L,j)} = \text{corr}\big(h_j^L(x),\, H_{\text{norm}}(x)\big).$$

We then rank all neurons globally by $|s^{(L,j)}|$.

**Deactivation.**  We select the top-$K$ most correlated neurons and set their activations to zero:

$$h_j^L(x) \mapsto 0, \quad j \in \mathcal{S}_K.$$

**Configuration.**  We sweep $K$ across $\{8, 64, 256, 1024, 4096\}$.

---

**Algorithm 2** Neuron-based Deactivation

---

**Require:** Activations $\{h^L(x)\}$, supervision $\{H_{\text{norm}}(x)\}$
1: **for** each layer $L$ and neuron $j$ **do**
2:     Collect $\{h_j^L(x)\}_x$ over dataset
3:     $s^{(L,j)} \leftarrow \text{corr}(h_j^L(x), H_{\text{norm}}(x))$
4: **end for**
5: Rank neurons by $|s^{(L,j)}|$, select top-$K$ indices $\mathcal{S}_K$
6: **for** each $(L, j) \in \mathcal{S}_K$ **do**
7:     Set $h_j^L(x) \leftarrow 0 \;\; \forall x$
8: **end for**

---

**Summary.**  This baseline evaluates how strongly neurons correlate with entropy-based repetition signals $H_{\text{norm}}(x)$, and ablates the most predictive dimensions by zeroing them out.

#### G.3.2  PROBE-BASED STEERING

**Setup.**  Given contrastive sets $\mathcal{D}^+$ (low repetition) and $\mathcal{D}^-$ (high repetition), we train a binary probe at a chosen layer $L$ to separate the two sets. The probe's learned weight vector is then used as a *steering direction*.

**Model.**  Let the activations at layer $L$ be $\{h^L(x)\}_{x \in \mathcal{D}^+ \cup \mathcal{D}^-} \subset \mathbb{R}^d$. We form a labeled dataset:

$$X = \big[h^L(x^+); h^L(x^-)\big], \qquad y = \big[\mathbf{1}_{|\mathcal{D}^+|}; \mathbf{0}_{|\mathcal{D}^-|}\big].$$

A logistic regression probe is trained with binary cross-entropy:

$$z = w^\top h + b, \qquad \mathcal{L} = \text{BCEWithLogits}(z, y).$$

**Steering.** After training, we extract the weight vector $w \in \mathbb{R}^d$. Optionally normalize it as $\hat{w} = w/\|w\|_2$. At inference, activations are updated as:

$$h' = h + \alpha \cdot \tilde{w}, \qquad \tilde{w} = \begin{cases} \hat{w}, & \text{if normalize=True,} \\ w, & \text{otherwise.} \end{cases}$$

where $\alpha \in \mathbb{R}$ controls the steering strength (Appendix B).

**Configuration.** We control the probe-based steering through three hyperparameters:

- **Layer** $L$: the layer at which activations are probed and edited.

- **Normalization**: whether to L2-normalize the weight vector $w$ (default: `False`).

- **Steering strength** $\alpha$: the magnitude of the applied edit (default: 1.0).

---

**Algorithm 3** Probe-based Steering

---

**Require:** Activations $\{h^L(x)\}$, contrastive sets $\mathcal{D}^+, \mathcal{D}^-$
 1: Build dataset $(X, y)$ from $\mathcal{D}^+$ and $\mathcal{D}^-$
 2: Train logistic regressor $(w, b)$ on $(X, y)$          $\triangleright$ epochs=200, lr=$10^{-2}$, weight_decay=0
 3: $\tilde{w} \leftarrow w/\|w\|_2$ if normalize=True else $w$
 4: At inference: $h' \leftarrow h + \alpha \cdot \tilde{w}$

---

**Summary.** This baseline learns a linear probe distinguishing high- vs. low-repetition activations, and uses its weight vector as a steering direction to reduce $R(x)$ while aiming to preserve $U(x)$.

### G.4 How to select the best parameters

In this section, we describe how we select the optimal hyperparameters for each method. Recall from Section 4 that we evaluate generated sequences using the repetition score $R(x)$ and the utility score $U(x)$ (see Appendix B). For each method, we choose the hyperparameters that maximize the harmonic mean of $R(x)$ and $U(x)$:

$$\text{Score}(x) = \frac{2 \cdot R(x) \cdot U(x)}{R(x) + U(x)}.$$

This criterion balances between reducing repetition and maintaining biological plausibility, ensuring that the selected parameters yield sequences that are both diverse and structurally sound.

## H Details of Experiment Results

### H.1 Main Results

We provide complete results across all three datasets (CATH, UniRef50, SCOP), both generation tasks (conditional, unconditional), and both model backbones (ESM3, ProtGPT2). Figures 8–9 present $R$–$U$ scatter plots, and Tables 12–13 report the full set of metrics underlying $R(x)$ and $U(x)$. Specifically, each table provides values for pLDDT, pTM, $H_{\text{norm}}$, Distinct-2, Distinct-3, and $R_{\text{hpoly}}$.

### H.2 Ablation Study Results

As introduced in Sec. 5.3, our ablation experiments probe the two main components of the method—(i) contrastive dataset construction and (ii) steering vector injection—along three axes: dataset sampling, steering strength $\alpha$, and steering layer $L$.

Table 10: ESM3 results for two tasks with utility constraint: (a) unconditional generation, (b) conditional generation.

| Method | CATH | | UniRef50 | | SCOP | |
|---|---|---|---|---|---|---|
| | $R \uparrow$ | $U \uparrow$ | $R \uparrow$ | $U \uparrow$ | $R \uparrow$ | $U \uparrow$ |
| **(a) Unconditional generation** | | | | | | |
| Original Model | $0.423_{\pm 0.024}$ | $0.576_{\pm 0.027}$ ✓ | $0.423_{\pm 0.024}$ | $0.576_{\pm 0.027}$ ✓ | $0.423_{\pm 0.024}$ | $0.576_{\pm 0.027}$ ✓ |
| Temperature Sampling | $0.751_{\pm 0.006}$ | $0.566_{\pm 0.022}$ | $0.751_{\pm 0.006}$ | $0.566_{\pm 0.022}$ | $0.751_{\pm 0.006}$ | $0.566_{\pm 0.022}$ |
| Top-p Sampling | $0.419_{\pm 0.030}$ | $0.590_{\pm 0.037}$ ✓ | $0.419_{\pm 0.030}$ | $0.590_{\pm 0.037}$ ✓ | $0.419_{\pm 0.030}$ | $0.590_{\pm 0.037}$ ✓ |
| Entropy Based Sampling | $\mathbf{0.904}_{\pm 0.002}$ | $0.477_{\pm 0.014}$ | $\mathbf{0.904}_{\pm 0.002}$ | $0.477_{\pm 0.014}$ | $\mathbf{0.904}_{\pm 0.002}$ | $0.477_{\pm 0.014}$ |
| Neuron Deactivation | $0.551_{\pm 0.071}$ | $0.507_{\pm 0.048}$ | $0.448_{\pm 0.040}$ | $0.594_{\pm 0.022}$ ✓ | $0.452_{\pm 0.020}$ | $0.597_{\pm 0.035}$ ✓ |
| Probe Steering | $0.415_{\pm 0.010}$ | $0.587_{\pm 0.011}$ ✓ | $0.416_{\pm 0.010}$ | $0.586_{\pm 0.010}$ ✓ | $0.415_{\pm 0.010}$ | $0.585_{\pm 0.011}$ ✓ |
| UCCS(ours) | $0.645_{\pm 0.013}$ | $\mathbf{0.652}_{\pm 0.020}$ ✓ | $0.631_{\pm 0.016}$ | $\mathbf{0.646}_{\pm 0.011}$ ✓ | $0.641_{\pm 0.014}$ | $\mathbf{0.646}_{\pm 0.012}$ ✓ |
| **(b) Conditional generation** | | | | | | |
| Original Model | $0.542_{\pm 0.014}$ | $0.686_{\pm 0.018}$ ✓ | $0.542_{\pm 0.014}$ | $0.686_{\pm 0.018}$ ✓ | $0.542_{\pm 0.014}$ | $0.686_{\pm 0.018}$ ✓ |
| Temperature Sampling | $0.792_{\pm 0.006}$ | $0.652_{\pm 0.021}$ | $0.792_{\pm 0.006}$ | $0.652_{\pm 0.021}$ | $0.792_{\pm 0.006}$ | $0.652_{\pm 0.021}$ |
| Top-p Sampling | $0.539_{\pm 0.016}$ | $0.681_{\pm 0.012}$ | $0.539_{\pm 0.016}$ | $0.681_{\pm 0.012}$ | $0.539_{\pm 0.016}$ | $0.681_{\pm 0.012}$ |
| Entropy Based Sampling | $\mathbf{0.923}_{\pm 0.003}$ | $0.578_{\pm 0.011}$ | $\mathbf{0.923}_{\pm 0.003}$ | $0.578_{\pm 0.011}$ | $\mathbf{0.923}_{\pm 0.003}$ | $0.578_{\pm 0.011}$ |
| Neuron Deactivation | $0.637_{\pm 0.024}$ | $0.636_{\pm 0.031}$ | $0.590_{\pm 0.100}$ | $0.638_{\pm 0.012}$ | $0.557_{\pm 0.013}$ | $0.673_{\pm 0.030}$ |
| Probe Steering | $0.546_{\pm 0.005}$ | $0.685_{\pm 0.024}$ | $0.546_{\pm 0.005}$ | $0.685_{\pm 0.022}$ | $0.546_{\pm 0.005}$ | $0.686_{\pm 0.023}$ |
| UCCS(ours) | $0.707_{\pm 0.019}$ | $\mathbf{0.720}_{\pm 0.021}$ ✓ | $0.707_{\pm 0.022}$ | $\mathbf{0.726}_{\pm 0.014}$ ✓ | $0.707_{\pm 0.021}$ | $\mathbf{0.717}_{\pm 0.018}$ ✓ |

*Notes.* $R$: repetition score; $U$: biological utility. we annotate cells with ✓ when $U \geq U_{\text{Orig}}$ (utility constraint satisfied).

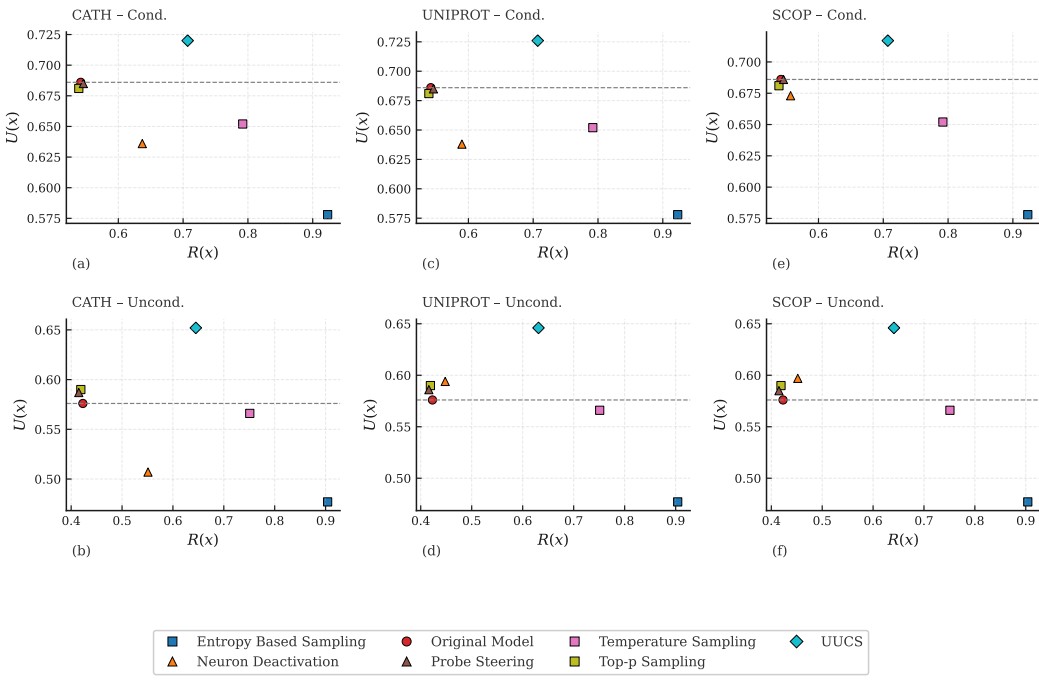

Figure 8: Main results of ESM3. $R$–$U$ scatter plots across three datasets (CATH, UniRef50, SCOP) and two generation conditions (conditional and unconditional), summarized into a single $2 \times 3$ panel.

**Setup.** Unless otherwise noted, we use UniRef50 (Consortium, 2024). For each ablation we construct contrastive sets $\mathcal{D}^+$ and $\mathcal{D}^-$ (100 sequences each) from natural proteins and PLM generations with high repetition. Results are averaged over five random seeds and evaluated under both *unconditional* and *conditional* generation. We report repetition score $R$ (higher is better) and bio-utility score $U$ (higher is better). Shaded areas and error bars indicate variability across seeds.

Below we present detailed visualizations and tabular results for each axis of ablation.

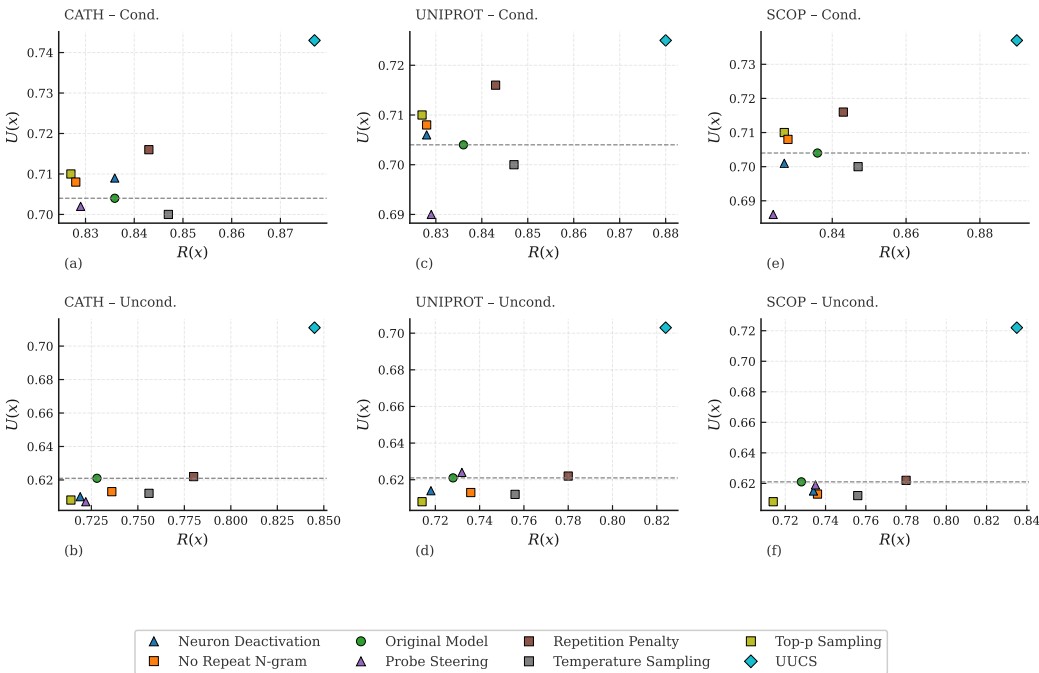

Figure 9: Main results of ProtGPT2. $R$–$U$ scatter plots across three datasets (CATH, UniRef50, SCOP) and two generation conditions (conditional and unconditional), summarized into a single $2{\times}3$ panel.

Table 11: ESM3 Results: Unconditional Generation

| Dataset | Method | Parameter | | | | Metrics | | |
|---------|--------|-----------|-------|-----|----------------|------------|------------|-----------------|
| | | | pLDDT | pTM | $H_{\text{norm}}$ | Distinct-2 | Distinct-3 | $R_{\text{hpoly}}$ |
| CATH | Original Model | / | 0.686 | 0.466 | 0.445 | 0.200 | 0.347 | 0.549 |
| CATH | Temperature Sampling | Temperature = 1.3 | 0.687 | 0.444 | 0.763 | 0.416 | 0.739 | 0.913 |
| CATH | Top-p Sampling | Top p = 0.95 | 0.699 | 0.482 | 0.440 | 0.193 | 0.339 | 0.550 |
| CATH | Entropy Based Sampling | / | 0.596 | 0.358 | 0.923 | 0.626 | 0.955 | 0.998 |
| CATH | Neuron Deactivation | Top K = 1024.0 | 0.643 | 0.372 | 0.545 | 0.216 | 0.429 | 0.785 |
| CATH | Probe Steering | Layer = 47 | 0.692 | 0.482 | 0.442 | 0.196 | 0.342 | 0.535 |
| CATH | UCCS(ours) | Layer = 47 | 0.780 | 0.524 | 0.632 | 0.296 | 0.549 | 0.880 |
| Uniref50 | Original Model | / | 0.686 | 0.466 | 0.445 | 0.200 | 0.347 | 0.549 |
| Uniref50 | Temperature Sampling | Temperature = 1.3 | 0.687 | 0.444 | 0.763 | 0.416 | 0.739 | 0.913 |
| Uniref50 | Top-p Sampling | Top p = 0.95 | 0.699 | 0.482 | 0.440 | 0.193 | 0.339 | 0.550 |
| Uniref50 | Entropy Based Sampling | / | 0.596 | 0.358 | 0.923 | 0.626 | 0.955 | 0.998 |
| Uniref50 | Neuron Deactivation | Top K = 256.0 | 0.700 | 0.487 | 0.474 | 0.204 | 0.353 | 0.592 |
| Uniref50 | Probe Steering | Layer = 47 | 0.691 | 0.480 | 0.442 | 0.196 | 0.343 | 0.536 |
| Uniref50 | UCCS(ours) | Layer = 47 | 0.778 | 0.513 | 0.619 | 0.286 | 0.531 | 0.866 |
| SCOP | Original Model | / | 0.686 | 0.466 | 0.445 | 0.200 | 0.347 | 0.549 |
| SCOP | Temperature Sampling | Temperature = 1.3 | 0.687 | 0.444 | 0.763 | 0.416 | 0.739 | 0.913 |
| SCOP | Top-p Sampling | Top p = 0.95 | 0.699 | 0.482 | 0.440 | 0.193 | 0.339 | 0.550 |
| SCOP | Entropy Based Sampling | / | 0.596 | 0.358 | 0.923 | 0.626 | 0.955 | 0.998 |
| SCOP | Neuron Deactivation | Top K = 256.0 | 0.707 | 0.487 | 0.479 | 0.202 | 0.351 | 0.602 |
| SCOP | Probe Steering | Layer = 47 | 0.691 | 0.479 | 0.441 | 0.196 | 0.342 | 0.535 |
| SCOP | UCCS(ours) | Layer = 47 | 0.773 | 0.519 | 0.628 | 0.294 | 0.546 | 0.876 |

Table 12: ESM3 Results: Conditional Generation

| Dataset | Method | Parameter | Metrics | | | | | |
|---|---|---|---|---|---|---|---|---|
| | | | pLDDT | pTM | $H_{\text{norm}}$ | Distinct-2 | Distinct-3 | $R_{\text{hpoly}}$ |
| CATH | Original Model | / | 0.808 | 0.565 | 0.545 | 0.339 | 0.515 | 0.655 |
| CATH | Temperature Sampling | Temperature = 1.3 | 0.785 | 0.519 | 0.768 | 0.549 | 0.824 | 0.921 |
| CATH | Top-p Sampling | Top p = 0.95 | 0.807 | 0.554 | 0.538 | 0.330 | 0.505 | 0.660 |
| CATH | Entropy Based Sampling | / | 0.720 | 0.435 | 0.909 | 0.755 | 0.974 | 0.996 |
| CATH | Neuron Deactivation | Top K = 1024.0 | 0.782 | 0.491 | 0.601 | 0.349 | 0.570 | 0.850 |
| CATH | Probe Steering | Layer = 47 | 0.813 | 0.558 | 0.548 | 0.339 | 0.517 | 0.663 |
| CATH | UCCS(ours) | Layer = 47 | 0.851 | 0.589 | 0.669 | 0.438 | 0.691 | 0.886 |
| Uniref50 | Original Model | / | 0.808 | 0.565 | 0.545 | 0.339 | 0.515 | 0.655 |
| Uniref50 | Temperature Sampling | Temperature = 1.3 | 0.785 | 0.519 | 0.768 | 0.549 | 0.824 | 0.921 |
| Uniref50 | Top-p Sampling | Top p = 0.95 | 0.807 | 0.554 | 0.538 | 0.330 | 0.505 | 0.660 |
| Uniref50 | Entropy Based Sampling | / | 0.720 | 0.435 | 0.909 | 0.755 | 0.974 | 0.996 |
| Uniref50 | Neuron Deactivation | Top K = 1024.0 | 0.785 | 0.491 | 0.573 | 0.339 | 0.539 | 0.758 |
| Uniref50 | Probe Steering | Layer = 47 | 0.813 | 0.557 | 0.548 | 0.339 | 0.517 | 0.663 |
| Uniref50 | UCCS(ours) | Layer = 47 | 0.857 | 0.595 | 0.672 | 0.439 | 0.691 | 0.884 |
| SCOP | Original Model | / | 0.808 | 0.565 | 0.545 | 0.339 | 0.515 | 0.655 |
| SCOP | Temperature Sampling | Temperature = 1.3 | 0.785 | 0.519 | 0.768 | 0.549 | 0.824 | 0.921 |
| SCOP | Top-p Sampling | Top p = 0.95 | 0.807 | 0.554 | 0.538 | 0.330 | 0.505 | 0.660 |
| SCOP | Entropy Based Sampling | / | 0.720 | 0.435 | 0.909 | 0.755 | 0.974 | 0.996 |
| SCOP | Neuron Deactivation | Top K = 64.0 | 0.799 | 0.547 | 0.559 | 0.335 | 0.506 | 0.692 |
| SCOP | Probe Steering | Layer = 47 | 0.814 | 0.558 | 0.547 | 0.339 | 0.516 | 0.663 |
| SCOP | UCCS(ours) | Layer = 47 | 0.849 | 0.585 | 0.671 | 0.440 | 0.695 | 0.884 |

Table 13: ProtGPT2 Results: Unconditional Generation

| Dataset | Method | Parameter | Metrics | | | | | |
|---|---|---|---|---|---|---|---|---|
| | | | pLDDT | pTM | $H_{\text{norm}}$ | Distinct-2 | Distinct-3 | $R_{\text{hpoly}}$ |
| CATH | Original Model | / | 0.747 | 0.495 | 0.721 | 0.406 | 0.647 | 0.936 |
| CATH | Temperature Sampling | Temperature = 1.3 | 0.746 | 0.478 | 0.745 | 0.430 | 0.707 | 0.954 |
| CATH | Top-p Sampling | Top p = 0.98 | 0.736 | 0.479 | 0.706 | 0.391 | 0.624 | 0.929 |
| CATH | No Repeat N-gram | no repeat 3.0 gram | 0.741 | 0.485 | 0.727 | 0.404 | 0.663 | 0.947 |
| CATH | Repetition Penalty | repetition penalty = 1.2 | 0.753 | 0.492 | 0.777 | 0.458 | 0.759 | 0.955 |
| CATH | Neuron Deactivation | Top K = 8.0 | 0.734 | 0.487 | 0.708 | 0.393 | 0.627 | 0.940 |
| CATH | Probe Steering | Layer = 30 | 0.739 | 0.475 | 0.711 | 0.408 | 0.649 | 0.926 |
| CATH | UCCS(ours) | Layer = 30 | 0.820 | 0.602 | 0.855 | 0.553 | 0.826 | 0.991 |
| Uniref50 | Original Model | / | 0.747 | 0.495 | 0.721 | 0.406 | 0.647 | 0.936 |
| Uniref50 | Temperature Sampling | Temperature = 1.3 | 0.746 | 0.478 | 0.745 | 0.430 | 0.707 | 0.954 |
| Uniref50 | Top-p Sampling | Top p = 0.98 | 0.736 | 0.479 | 0.706 | 0.391 | 0.624 | 0.929 |
| Uniref50 | No Repeat N-gram | no repeat 3.0 gram | 0.741 | 0.485 | 0.727 | 0.404 | 0.663 | 0.947 |
| Uniref50 | Repetition Penalty | repetition penalty = 1.2 | 0.753 | 0.492 | 0.777 | 0.458 | 0.759 | 0.955 |
| Uniref50 | Neuron Deactivation | Top K = 8.0 | 0.746 | 0.481 | 0.705 | 0.389 | 0.632 | 0.940 |
| Uniref50 | Probe Steering | Layer = 30 | 0.749 | 0.498 | 0.723 | 0.411 | 0.649 | 0.945 |
| Uniref50 | UCCS(ours) | Layer = 30 | 0.807 | 0.599 | 0.828 | 0.510 | 0.788 | 0.994 |
| SCOP | Original Model | / | 0.747 | 0.495 | 0.721 | 0.406 | 0.647 | 0.936 |
| SCOP | Temperature Sampling | Temperature = 1.3 | 0.746 | 0.478 | 0.745 | 0.430 | 0.707 | 0.954 |
| SCOP | Top-p Sampling | Top p = 0.98 | 0.736 | 0.479 | 0.706 | 0.391 | 0.624 | 0.929 |
| SCOP | No Repeat N-gram | no repeat 3.0 gram | 0.741 | 0.485 | 0.727 | 0.404 | 0.663 | 0.947 |
| SCOP | Repetition Penalty | repetition penalty = 1.2 | 0.753 | 0.492 | 0.777 | 0.458 | 0.759 | 0.955 |
| SCOP | Neuron Deactivation | Top K = 64.0 | 0.742 | 0.487 | 0.727 | 0.412 | 0.664 | 0.938 |
| SCOP | Probe Steering | Layer = 30 | 0.746 | 0.492 | 0.728 | 0.415 | 0.661 | 0.939 |
| SCOP | UCCS(ours) | Layer = 30 | 0.823 | 0.621 | 0.844 | 0.530 | 0.805 | 0.993 |

### H.2.1 DATASET SAMPLING ABLATION

This section expands the summary in Sec. 5.3 by detailing the effect of different sampling strategies (*Simple random*, *Pareto*, and *Composite*) for constructing contrastive datasets. We report both unconditional and conditional results for ESM3 and ProtGPT2.

Table 14: ProtGPT2 Results: Conditional Generation

| Dataset | Method | Parameter | Metrics | | | | | |
|---------|--------|-----------|---------|-----|--------------|-----------|-----------|----------------|
| | | | pLDDT | pTM | $H_{\mathrm{norm}}$ | Distinct-2 | Distinct-3 | $R_{\mathrm{hpoly}}$ |
| CATH | Original Model | / | 0.836 | 0.572 | 0.800 | 0.620 | 0.850 | 0.973 |
| CATH | Temperature Sampling | Temperature = 1.3 | 0.829 | 0.571 | 0.811 | 0.634 | 0.872 | 0.978 |
| CATH | Top-p Sampling | Top p = 0.98 | 0.837 | 0.583 | 0.794 | 0.612 | 0.833 | 0.966 |
| CATH | No Repeat N-gram | no repeat 2.0 gram | 0.839 | 0.577 | 0.793 | 0.612 | 0.840 | 0.965 |
| CATH | Repetition Penalty | repetition penalty = 1.1 | 0.841 | 0.591 | 0.807 | 0.632 | 0.871 | 0.971 |
| CATH | Neuron Deactivation | Top K = 64.0 | 0.837 | 0.581 | 0.802 | 0.626 | 0.847 | 0.971 |
| CATH | Probe Steering | Layer = 30 | 0.834 | 0.569 | 0.795 | 0.608 | 0.833 | 0.970 |
| CATH | UCCS(ours) | Layer = 30 | 0.851 | 0.635 | 0.860 | 0.672 | 0.891 | 0.991 |
| Uniref50 | Original Model | / | 0.836 | 0.572 | 0.800 | 0.620 | 0.850 | 0.973 |
| Uniref50 | Temperature Sampling | Temperature = 1.3 | 0.829 | 0.571 | 0.811 | 0.634 | 0.872 | 0.978 |
| Uniref50 | Top-p Sampling | Top p = 0.98 | 0.837 | 0.583 | 0.794 | 0.612 | 0.833 | 0.966 |
| Uniref50 | No Repeat N-gram | no repeat 2.0 gram | 0.839 | 0.577 | 0.793 | 0.612 | 0.840 | 0.965 |
| Uniref50 | Repetition Penalty | repetition penalty = 1.1 | 0.841 | 0.591 | 0.807 | 0.632 | 0.871 | 0.971 |
| Uniref50 | Neuron Deactivation | Top K = 8.0 | 0.834 | 0.577 | 0.793 | 0.603 | 0.837 | 0.973 |
| Uniref50 | Probe Steering | Layer = 30 | 0.828 | 0.552 | 0.793 | 0.614 | 0.837 | 0.968 |
| Uniref50 | UCCS(ours) | Layer = 30 | 0.834 | 0.616 | 0.859 | 0.675 | 0.895 | 0.995 |
| SCOP | Original Model | / | 0.836 | 0.572 | 0.800 | 0.620 | 0.850 | 0.973 |
| SCOP | Temperature Sampling | Temperature = 1.3 | 0.829 | 0.571 | 0.811 | 0.634 | 0.872 | 0.978 |
| SCOP | Top-p Sampling | Top p = 0.98 | 0.837 | 0.583 | 0.794 | 0.612 | 0.833 | 0.966 |
| SCOP | No Repeat N-gram | no repeat 2.0 gram | 0.839 | 0.577 | 0.793 | 0.612 | 0.840 | 0.965 |
| SCOP | Repetition Penalty | repetition penalty = 1.1 | 0.841 | 0.591 | 0.807 | 0.632 | 0.871 | 0.971 |
| SCOP | Neuron Deactivation | Top K = 64.0 | 0.831 | 0.571 | 0.794 | 0.616 | 0.839 | 0.962 |
| SCOP | Probe Steering | Layer = 30 | 0.821 | 0.551 | 0.789 | 0.607 | 0.828 | 0.966 |
| SCOP | UCCS(ours) | Layer = 30 | 0.846 | 0.627 | 0.873 | 0.693 | 0.918 | 0.991 |

Table 15: dataset sampling ablation: per-task performance across seed runs (mean $\pm$ std), model: ESM3, dataset: Uniref50.

| **dataset** | R | U | $HM(\mathrm{R}, \mathrm{U})$ | $n$ |
|---------|---|---|------|---|
| *unconditional* | | | | |
| pareto | $0.636_{\pm 0.013}$ | $0.674_{\pm 0.010}$ | 0.655 | 5 |
| composite | $0.623_{\pm 0.012}$ | $0.657_{\pm 0.012}$ | 0.639 | 5 |
| simple (random) | $0.618_{\pm 0.010}$ | $0.634_{\pm 0.021}$ | 0.626 | 5 |
| *conditional* | | | | |
| pareto | $0.703_{\pm 0.008}$ | $0.715_{\pm 0.016}$ | 0.709 | 5 |
| composite | $0.693_{\pm 0.022}$ | $0.718_{\pm 0.012}$ | 0.705 | 5 |
| simple (random) | $0.692_{\pm 0.007}$ | $0.706_{\pm 0.020}$ | 0.699 | 5 |

*notes.* R denotes the repetition score $R(x)$, and U denotes the bio-utility $U(x)$.

**Visualization.** We visualize the repetition score ($R(x)$)–utility score ($U(x)$) trade-off under three dataset sampling strategies (*Simple random*, *Pareto*, and *Composite*) for both tasks (unconditional vs. conditional) and both models (ESM3 vs. ProtGPT2). Each marker corresponds to a seed, and shaded ellipses indicate the 95% covariance region across seeds for each method.

**Results.** Table 15 and Table 16 report the per-task mean±std for repetition and utility metrics, together with the harmonic mean $HM(R, U)$. These tables complement the visualizations by providing robust averages and variability across seeds.

**Findings.** Overall, our *Pareto* and *Composite* strategies outperform simple random sampling. This is especially evident in the **unconditional task**, where the error ellipses are tighter (indicating greater

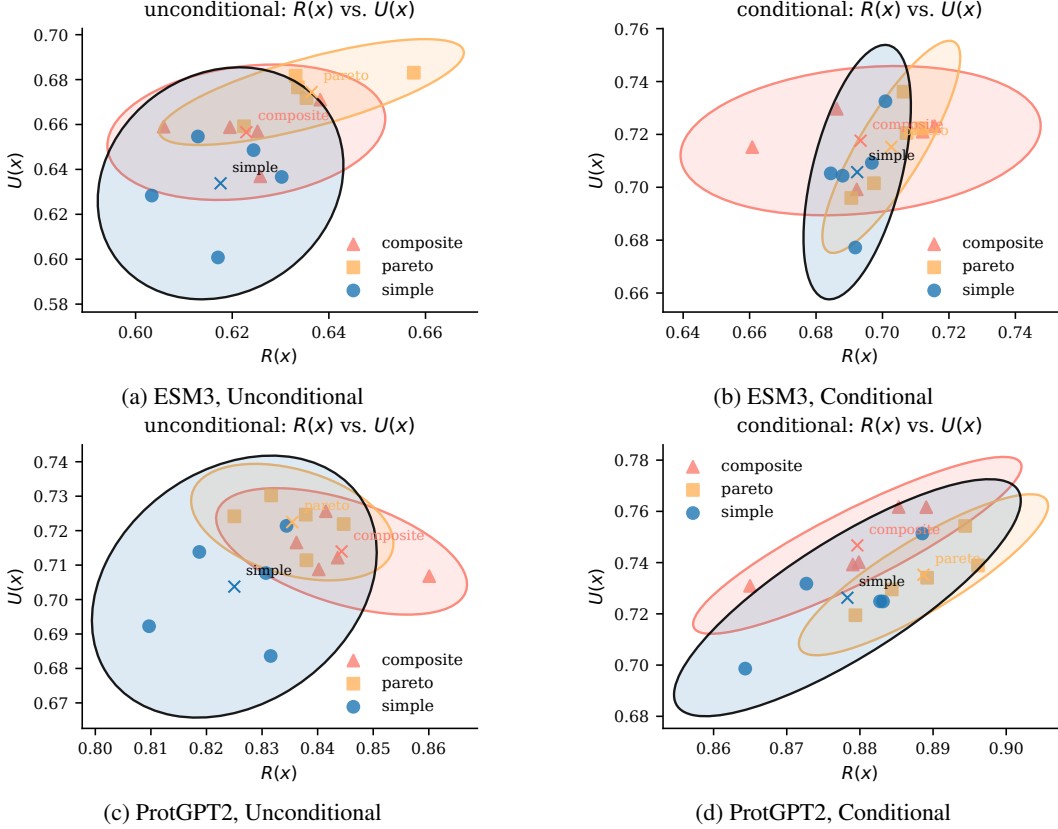

Figure 10: Dataset sampling ablation for ESM3 and ProtGPT2 under unconditional and conditional generation. Each point is a seed, with ellipses showing 95% seed covariance.

Table 16: dataset sampling ablation: per-task performance across seed runs (mean ± std), model: PROTGPT2, dataset: Uniref50.

| dataset | R | U | $HM(\text{R}, \text{U})$ | $n$ |
|---|---|---|---|---|
| *unconditional* | | | | |
| pareto | $0.835_{\pm 0.007}$ | $0.722_{\pm 0.007}$ | 0.775 | 5 |
| composite | $0.844_{\pm 0.009}$ | $0.714_{\pm 0.007}$ | 0.774 | 5 |
| simple (random) | $0.825_{\pm 0.010}$ | $0.704_{\pm 0.016}$ | 0.760 | 5 |
| *conditional* | | | | |
| composite | $0.880_{\pm 0.009}$ | $0.747_{\pm 0.014}$ | 0.808 | 5 |
| pareto | $0.889_{\pm 0.007}$ | $0.735_{\pm 0.013}$ | 0.805 | 5 |
| simple (random) | $0.878_{\pm 0.010}$ | $0.726_{\pm 0.019}$ | 0.795 | 5 |

*notes.* R denotes the repetition score $R(x)$, and U denotes the bio-utility $U(x)$.

stability across seeds) and both $R$ and $U$ values are consistently higher. In conditional generation, differences are smaller but Pareto/Composite still maintain a clear edge.

### H.2.2 STEERING STRENGTH ABLATION

This section provides the full results for the steering strength($\alpha$ in section4.3) ablation summarized in Sec. 5.3. We investigate how injecting the steering vector at different strengths affects $R$ and $U$ in ESM3 and ProtGPT2.

**Visualization.** We provide three complementary plots per model: (i) $\log_2 \alpha$ vs. repetition $R(x)$, (ii) $\log_2 \alpha$ vs. utility $U(x)$, and (iii) the $R(x)$–$U(x)$ trade-off scatter. Shaded bands indicate 95% confidence intervals across seeds, and markers highlight the selected best operating points.

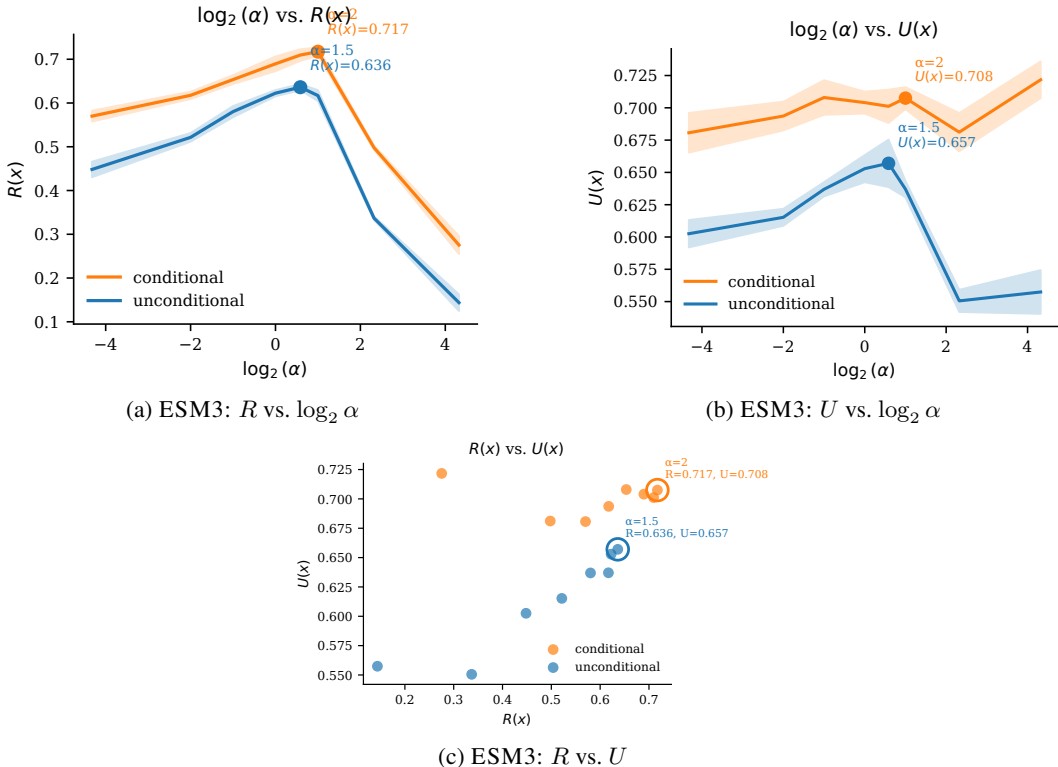

(a) ESM3: $R$ vs. $\log_2 \alpha$        (b) ESM3: $U$ vs. $\log_2 \alpha$

(c) ESM3: $R$ vs. $U$

Figure 11: Steering-strength ($\alpha$) ablation on ESM3. Conditional sampling consistently traces the upper envelope in the $R$–$U$ plane.

**Results.** Tables 17 and 18 summarize the per-task means ($\pm$ std) across seeds for each tested $\alpha$, reporting atomic metrics (e.g., $H_{\text{norm}}$, Distinct-$n$, $R_{\text{hpoly}}$, pLDDT, pTM) together with the aggregates $R$ and $U$.

**Observations.** Across both backbones, $\log_2(\alpha)$ exhibits a unimodal trend: increasing from negative values to roughly $0.5 \sim 1$ ($\alpha \approx 1.4 \sim 2$) improves both $R$ and $U$; further increases cause a marked drop in the *unconditional* setting, while the *conditional* setting is comparatively robust (Figs. 11–12).

**ESM3.** As shown in Table 17, unconditional performance peaks around $\alpha = 1.5$ ($\log_2 \alpha = 0.585$), where $R = 0.636$ and $U = 0.657$. Conditional sampling attains its joint optimum near $\alpha = 2$ ($R = 0.717$, $U = 0.708$). Very large $\alpha$ (e.g., 20) yields imbalanced trade-offs: some structure metrics increase, but $R$ collapses.

**ProtGPT2.** When selecting the operating point based on the harmonic mean of $R$ and $U$ (Section G.4), the optimal setting is $\alpha = 1.0$. In the unconditional case, this yields $R = 0.833$ and $U = 0.706$, while for conditional sampling we obtain $R = 0.891$ and $U = 0.742$. Although larger values of $\alpha$ (e.g., $\alpha = 1.5$) further increase $R$, this comes at the cost of a noticeable drop in $U$. Overall, conditional curves dominate the Pareto frontier, and $\alpha = 1.0$ offers the most balanced trade-off between repetition reduction and bio-utility (Figs. 11c,12c).

**Recommendation.** Both models exhibit stable trade-offs when $\alpha$ lies in the range 1–2. In practice, we recommend selecting $\alpha = 1.0$, which maximizes the harmonic mean of $R$ and $U$ (Appendix G.4) and thus provides the most balanced operating point. For conditional sampling, values in $[1, 2]$ re-

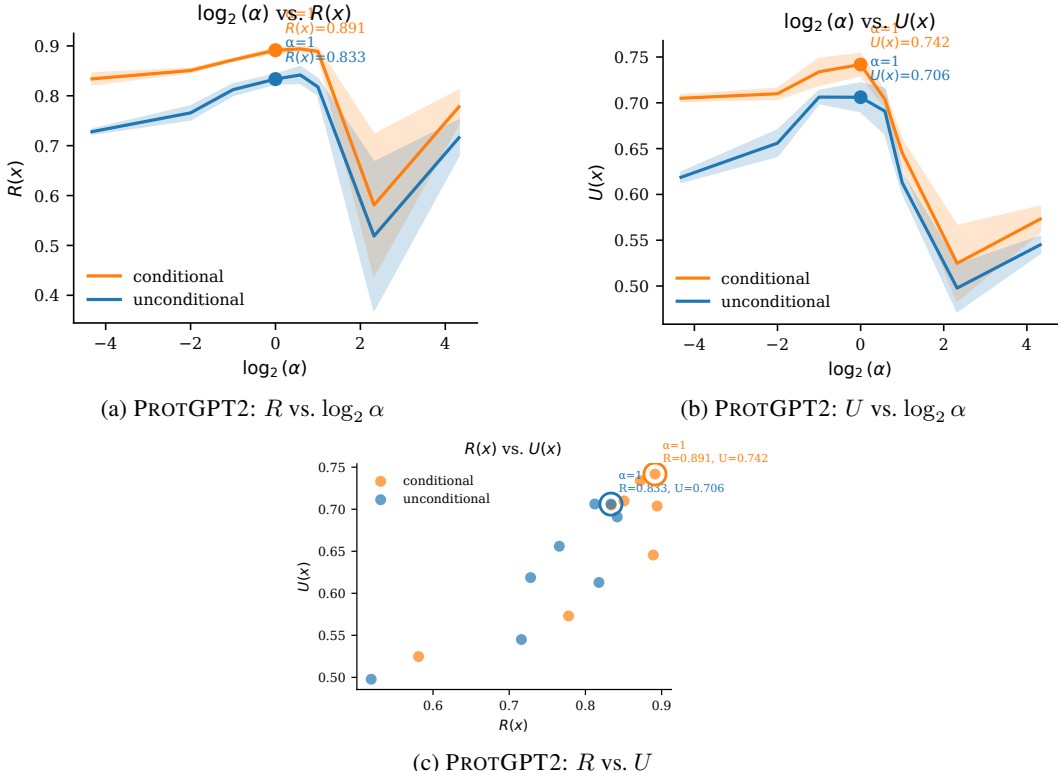

Figure 12: Steering-strength ($\alpha$) ablation on PROTGPT2. Shaded areas denote seed variability. Markers highlight the selected operating points discussed below.

Table 17: Alpha ablation: repetition and bio-utility metrics per $\alpha$ (mean $\pm$ std across seeds). Model: ESM3, Dataset: Uniref50

| $\alpha$ | $\log_2(\alpha)$ | $H_{norm}$ | Dist-2 | Dist-3 | $R_{hpoly}$ | pLDDT | pTM | R | U |
|---|---|---|---|---|---|---|---|---|---|
| *Unconditional* | | | | | | | | | |
| 0.05 | -4.32 | $0.472_{\pm 0.013}$ | $0.207_{\pm 0.014}$ | $0.366_{\pm 0.024}$ | $0.586_{\pm 0.032}$ | $0.712_{\pm 0.012}$ | $0.493_{\pm 0.013}$ | $0.448_{\pm 0.020}$ | $0.603_{\pm 0.012}$ |
| 0.25 | -2 | $0.540_{\pm 0.011}$ | $0.239_{\pm 0.009}$ | $0.431_{\pm 0.013}$ | $0.689_{\pm 0.015}$ | $0.737_{\pm 0.010}$ | $0.493_{\pm 0.009}$ | $0.521_{\pm 0.011}$ | $0.615_{\pm 0.007}$ |
| 0.5 | -1 | $0.590_{\pm 0.012}$ | $0.265_{\pm 0.007}$ | $0.487_{\pm 0.010}$ | $0.775_{\pm 0.025}$ | $0.766_{\pm 0.006}$ | $0.508_{\pm 0.011}$ | $0.580_{\pm 0.014}$ | $0.637_{\pm 0.006}$ |
| 1 | 0 | $0.616_{\pm 0.007}$ | $0.281_{\pm 0.004}$ | $0.528_{\pm 0.004}$ | $0.847_{\pm 0.016}$ | $0.781_{\pm 0.012}$ | $0.525_{\pm 0.013}$ | $0.622_{\pm 0.008}$ | $0.653_{\pm 0.012}$ |
| 1.5 | 0.585 | $0.616_{\pm 0.008}$ | $0.285_{\pm 0.012}$ | $0.534_{\pm 0.014}$ | $0.883_{\pm 0.009}$ | $0.776_{\pm 0.020}$ | $0.538_{\pm 0.022}$ | $0.636_{\pm 0.008}$ | $0.657_{\pm 0.021}$ |
| 2 | 1 | $0.588_{\pm 0.014}$ | $0.269_{\pm 0.011}$ | $0.503_{\pm 0.018}$ | $0.877_{\pm 0.015}$ | $0.759_{\pm 0.007}$ | $0.515_{\pm 0.010}$ | $0.617_{\pm 0.014}$ | $0.637_{\pm 0.007}$ |
| 5 | 2.32 | $0.344_{\pm 0.005}$ | $0.153_{\pm 0.007}$ | $0.267_{\pm 0.012}$ | $0.456_{\pm 0.020}$ | $0.707_{\pm 0.011}$ | $0.394_{\pm 0.010}$ | $0.337_{\pm 0.005}$ | $0.551_{\pm 0.010}$ |
| 20 | 4.32 | $0.164_{\pm 0.017}$ | $0.073_{\pm 0.003}$ | $0.115_{\pm 0.007}$ | $0.173_{\pm 0.043}$ | $0.711_{\pm 0.023}$ | $0.404_{\pm 0.017}$ | $0.144_{\pm 0.021}$ | $0.557_{\pm 0.019}$ |
| *Conditional* | | | | | | | | | |
| 0.05 | -4.32 | $0.566_{\pm 0.010}$ | $0.349_{\pm 0.004}$ | $0.535_{\pm 0.009}$ | $0.702_{\pm 0.031}$ | $0.814_{\pm 0.014}$ | $0.548_{\pm 0.023}$ | $0.570_{\pm 0.014}$ | $0.681_{\pm 0.018}$ |
| 0.25 | -2 | $0.605_{\pm 0.010}$ | $0.376_{\pm 0.005}$ | $0.584_{\pm 0.011}$ | $0.768_{\pm 0.017}$ | $0.830_{\pm 0.010}$ | $0.557_{\pm 0.017}$ | $0.618_{\pm 0.010}$ | $0.694_{\pm 0.013}$ |
| 0.5 | -1 | $0.634_{\pm 0.014}$ | $0.403_{\pm 0.009}$ | $0.627_{\pm 0.015}$ | $0.811_{\pm 0.011}$ | $0.849_{\pm 0.010}$ | $0.567_{\pm 0.023}$ | $0.654_{\pm 0.011}$ | $0.708_{\pm 0.015}$ |
| 1 | 0 | $0.657_{\pm 0.020}$ | $0.426_{\pm 0.010}$ | $0.667_{\pm 0.017}$ | $0.865_{\pm 0.025}$ | $0.842_{\pm 0.012}$ | $0.566_{\pm 0.017}$ | $0.690_{\pm 0.019}$ | $0.704_{\pm 0.010}$ |
| 1.5 | 0.585 | $0.667_{\pm 0.015}$ | $0.439_{\pm 0.007}$ | $0.684_{\pm 0.018}$ | $0.901_{\pm 0.019}$ | $0.839_{\pm 0.013}$ | $0.563_{\pm 0.021}$ | $0.710_{\pm 0.015}$ | $0.701_{\pm 0.015}$ |
| 2 | 1 | $0.666_{\pm 0.010}$ | $0.437_{\pm 0.010}$ | $0.686_{\pm 0.013}$ | $0.924_{\pm 0.013}$ | $0.841_{\pm 0.012}$ | $0.574_{\pm 0.011}$ | $0.717_{\pm 0.011}$ | $0.708_{\pm 0.010}$ |
| 5 | 2.32 | $0.489_{\pm 0.006}$ | $0.315_{\pm 0.005}$ | $0.463_{\pm 0.008}$ | $0.615_{\pm 0.019}$ | $0.849_{\pm 0.018}$ | $0.514_{\pm 0.016}$ | $0.498_{\pm 0.006}$ | $0.681_{\pm 0.017}$ |
| 20 | 4.32 | $0.312_{\pm 0.017}$ | $0.193_{\pm 0.006}$ | $0.253_{\pm 0.011}$ | $0.291_{\pm 0.045}$ | $0.875_{\pm 0.015}$ | $0.568_{\pm 0.019}$ | $0.275_{\pm 0.023}$ | $0.722_{\pm 0.016}$ |

main competitive: $\alpha \approx 1$ favors higher utility $U$, whereas $\alpha \approx 1.5$–$2$ emphasizes repetition reduction $R$ at the cost of some $U$. Very small or large values of $\alpha$ lead to inferior performance in both settings.

Table 18: Alpha ablation: repetition and bio-utility metrics per $\alpha$ (mean $\pm$ std across seeds). Model: ProtGPT2, Dataset: Uniref50

| $\alpha$ | $\log_2(\alpha)$ | $H_{norm}$ | Dist-2 | Dist-3 | $R_{hpoly}$ | pLDDT | pTM | R | U |
|---|---|---|---|---|---|---|---|---|---|
| *Unconditional* | | | | | | | | | |
| 0.05 | -4.32 | $0.719_{\pm 0.007}$ | $0.407_{\pm 0.013}$ | $0.649_{\pm 0.018}$ | $0.937_{\pm 0.012}$ | $0.744_{\pm 0.007}$ | $0.494_{\pm 0.008}$ | $0.728_{\pm 0.005}$ | $0.619_{\pm 0.006}$ |
| 0.25 | -2 | $0.756_{\pm 0.015}$ | $0.440_{\pm 0.027}$ | $0.706_{\pm 0.025}$ | $0.968_{\pm 0.010}$ | $0.776_{\pm 0.015}$ | $0.535_{\pm 0.018}$ | $0.766_{\pm 0.015}$ | $0.656_{\pm 0.016}$ |
| 0.5 | -1 | $0.814_{\pm 0.014}$ | $0.505_{\pm 0.027}$ | $0.780_{\pm 0.024}$ | $0.981_{\pm 0.001}$ | $0.814_{\pm 0.007}$ | $0.598_{\pm 0.009}$ | $0.812_{\pm 0.013}$ | $0.706_{\pm 0.008}$ |
| 1 | 0 | $0.838_{\pm 0.011}$ | $0.534_{\pm 0.025}$ | $0.807_{\pm 0.018}$ | $0.992_{\pm 0.006}$ | $0.811_{\pm 0.015}$ | $0.601_{\pm 0.021}$ | $0.833_{\pm 0.011}$ | $0.706_{\pm 0.018}$ |
| 1.5 | 0.585 | $0.847_{\pm 0.020}$ | $0.548_{\pm 0.037}$ | $0.815_{\pm 0.035}$ | $0.997_{\pm 0.001}$ | $0.803_{\pm 0.022}$ | $0.579_{\pm 0.035}$ | $0.842_{\pm 0.019}$ | $0.691_{\pm 0.028}$ |
| 2 | 1 | $0.816_{\pm 0.015}$ | $0.510_{\pm 0.042}$ | $0.777_{\pm 0.041}$ | $0.994_{\pm 0.001}$ | $0.753_{\pm 0.005}$ | $0.472_{\pm 0.020}$ | $0.818_{\pm 0.018}$ | $0.613_{\pm 0.012}$ |
| 5 | 2.32 | $0.736_{\pm 0.039}$ | $0.311_{\pm 0.114}$ | $0.425_{\pm 0.232}$ | $0.453_{\pm 0.305}$ | $0.656_{\pm 0.012}$ | $0.340_{\pm 0.047}$ | $0.519_{\pm 0.169}$ | $0.498_{\pm 0.029}$ |
| 20 | 4.32 | $0.697_{\pm 0.051}$ | $0.321_{\pm 0.057}$ | $0.587_{\pm 0.072}$ | $0.997_{\pm 0.007}$ | $0.630_{\pm 0.014}$ | $0.460_{\pm 0.012}$ | $0.716_{\pm 0.040}$ | $0.545_{\pm 0.010}$ |
| *Conditional* | | | | | | | | | |
| 0.05 | -4.32 | $0.802_{\pm 0.011}$ | $0.611_{\pm 0.032}$ | $0.841_{\pm 0.025}$ | $0.974_{\pm 0.007}$ | $0.833_{\pm 0.006}$ | $0.577_{\pm 0.012}$ | $0.834_{\pm 0.013}$ | $0.705_{\pm 0.004}$ |
| 0.25 | -2 | $0.821_{\pm 0.010}$ | $0.639_{\pm 0.003}$ | $0.866_{\pm 0.009}$ | $0.979_{\pm 0.007}$ | $0.833_{\pm 0.005}$ | $0.587_{\pm 0.013}$ | $0.851_{\pm 0.005}$ | $0.710_{\pm 0.007}$ |
| 0.5 | -1 | $0.843_{\pm 0.009}$ | $0.670_{\pm 0.010}$ | $0.896_{\pm 0.009}$ | $0.990_{\pm 0.002}$ | $0.848_{\pm 0.015}$ | $0.619_{\pm 0.018}$ | $0.872_{\pm 0.003}$ | $0.734_{\pm 0.016}$ |
| 1 | 0 | $0.868_{\pm 0.011}$ | $0.703_{\pm 0.014}$ | $0.920_{\pm 0.012}$ | $0.994_{\pm 0.003}$ | $0.853_{\pm 0.012}$ | $0.631_{\pm 0.021}$ | $0.891_{\pm 0.006}$ | $0.742_{\pm 0.014}$ |
| 1.5 | 0.585 | $0.874_{\pm 0.013}$ | $0.710_{\pm 0.011}$ | $0.913_{\pm 0.010}$ | $0.996_{\pm 0.002}$ | $0.827_{\pm 0.011}$ | $0.580_{\pm 0.013}$ | $0.894_{\pm 0.005}$ | $0.704_{\pm 0.009}$ |
| 2 | 1 | $0.864_{\pm 0.014}$ | $0.706_{\pm 0.013}$ | $0.910_{\pm 0.016}$ | $0.995_{\pm 0.003}$ | $0.792_{\pm 0.011}$ | $0.499_{\pm 0.025}$ | $0.889_{\pm 0.007}$ | $0.646_{\pm 0.015}$ |
| 5 | 2.32 | $0.798_{\pm 0.025}$ | $0.409_{\pm 0.150}$ | $0.493_{\pm 0.233}$ | $0.495_{\pm 0.276}$ | $0.703_{\pm 0.031}$ | $0.346_{\pm 0.064}$ | $0.581_{\pm 0.161}$ | $0.525_{\pm 0.047}$ |
| 20 | 4.32 | $0.748_{\pm 0.044}$ | $0.468_{\pm 0.058}$ | $0.715_{\pm 0.068}$ | $0.995_{\pm 0.008}$ | $0.690_{\pm 0.030}$ | $0.456_{\pm 0.014}$ | $0.778_{\pm 0.038}$ | $0.573_{\pm 0.016}$ |

### H.2.3 STEERING LAYER ABLATION

We further study the effect of inserting the mean difference vector at different transformer layers. Results are reported for both ESM3 (48-layer auto-encoding) and PROTGPT2 (36-layer autoregressive) backbones.

**Visualization.** We provide three complementary plots per model: (i) layer index $L$ vs. repetition $R(x)$, (ii) layer index $L$ vs. utility $U(x)$, and (iii) the $R(x)$–$U(x)$ trade-off scatter. As before, shaded bands indicate 95% confidence intervals across seeds, and markers denote the best-performing layers.

**Results.** Tables 21–19 report, for the unconditional setting, the per-task averages ($\pm$ standard deviation across seeds) of each steering layer. The tables include both the atomic metrics ($H_{norm}$, Distinct-$n$, $R_{hpoly}$, pLDDT, pTM) and the aggregate scores $R$ and $U$. The corresponding results for the conditional setting are presented in Tables 22–20.

**Observations.** For ESM3, steering at early or middle layers ($L < 40$) has little effect on either $R$ or $U$. A marked improvement appears only in the final layers, peaking at $L = 46$ where $R = 0.652$ and $U = 0.660$ in the unconditional setting(Table 21, and $R = 0.723$, $U = 0.721$ in the conditional setting(Table 20. This indicates that late-layer intervention is most effective for the auto-encoding backbone.

For PROTGPT2, repetition control is already strong overall. Steering in very early layers (e.g., $L = 2$) or around mid layers ($L = 18$) tends to exacerbate repetition and slightly reduce $U$. In contrast, steering after $L = 20$ gradually improves both metrics, with a clear optimum around $L = 30$ ($R = 0.888$, $U = 0.730$ in the conditional setting, see Table 20).

Overall, both models demonstrate that effective steering occurs in the later portion of the network, but the precise "sweet spot" depends on architecture depth: the last few layers for ESM3, and the last third of the stack for PROTGPT2. Conditional sampling consistently traces the upper $R$–$U$ envelope across layers.

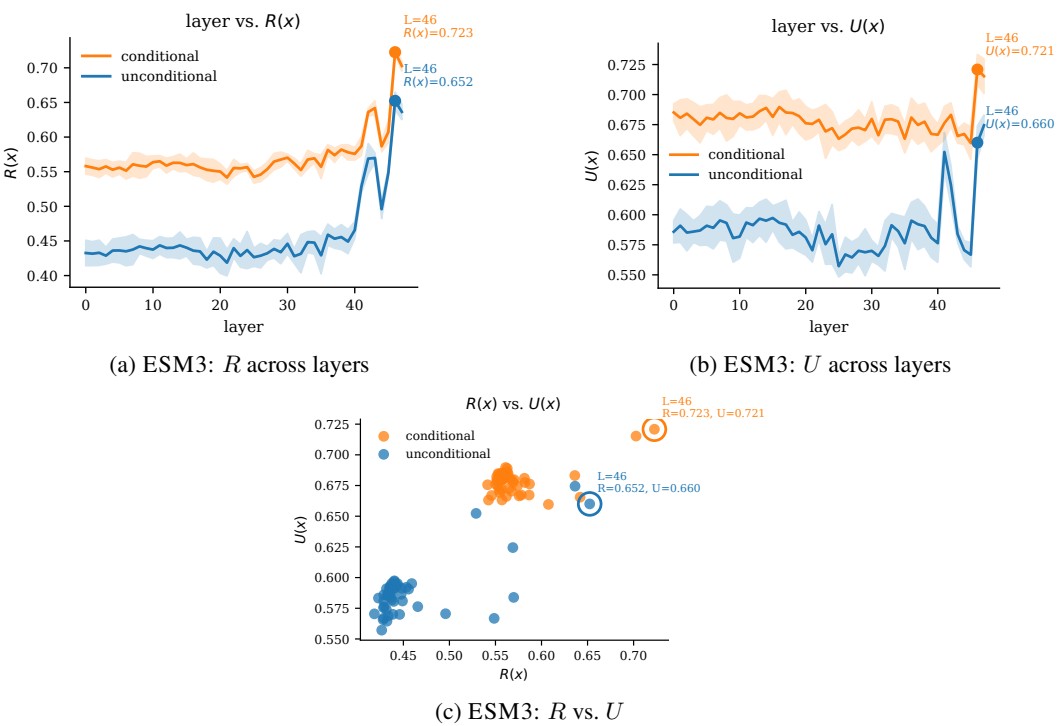

(a) ESM3: $R$ across layers

(b) ESM3: $U$ across layers

(c) ESM3: $R$ vs. $U$

Figure 13: Layer-wise steering ablation on ESM3. Conditional sampling tends to trace the upper envelope in the $R$–$U$ plane.

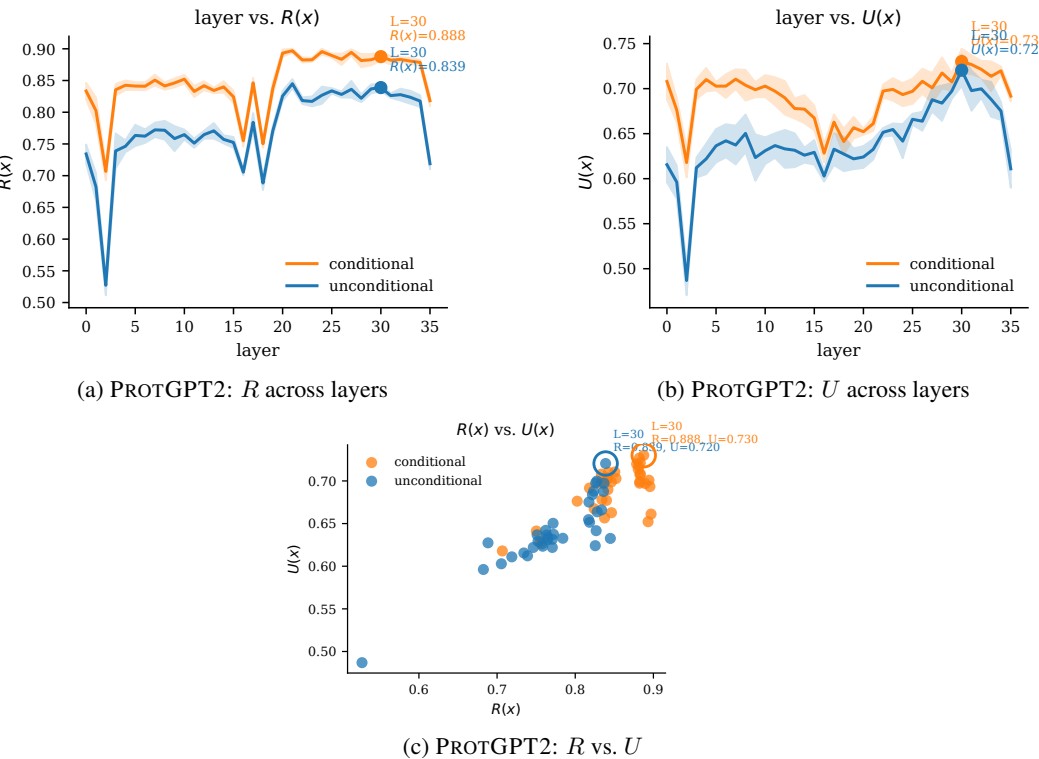

(a) PROTGPT2: $R$ across layers

(b) PROTGPT2: $U$ across layers

(c) PROTGPT2: $R$ vs. $U$

Figure 14: Layer-wise steering ablation on PROTGPT2. Shaded areas denote seed variability.

Table 19: Layer ablation (unconditional) result of ProtGPT2 on Uniref50: repetition and bio-utility metrics per layer (mean $\pm$ std across seeds).

| layer | $H_{norm}$ | Dist-2 | Dist-3 | $R_{hpoly}$ | pLDDT | pTM | R | U |
|---|---|---|---|---|---|---|---|---|
| 0 | $0.728_{\pm0.020}$ | $0.410_{\pm0.026}$ | $0.656_{\pm0.037}$ | $0.942_{\pm0.010}$ | $0.743_{\pm0.017}$ | $0.488_{\pm0.028}$ | $0.734_{\pm0.017}$ | $0.616_{\pm0.022}$ |
| 1 | $0.670_{\pm0.026}$ | $0.349_{\pm0.031}$ | $0.565_{\pm0.039}$ | $0.921_{\pm0.020}$ | $0.721_{\pm0.018}$ | $0.472_{\pm0.025}$ | $0.682_{\pm0.025}$ | $0.596_{\pm0.021}$ |
| 2 | $0.498_{\pm0.015}$ | $0.167_{\pm0.015}$ | $0.286_{\pm0.025}$ | $0.857_{\pm0.022}$ | $0.624_{\pm0.021}$ | $0.350_{\pm0.017}$ | $0.527_{\pm0.018}$ | $0.487_{\pm0.019}$ |
| 3 | $0.730_{\pm0.029}$ | $0.416_{\pm0.039}$ | $0.658_{\pm0.051}$ | $0.950_{\pm0.022}$ | $0.735_{\pm0.014}$ | $0.490_{\pm0.014}$ | $0.739_{\pm0.032}$ | $0.612_{\pm0.013}$ |
| 4 | $0.733_{\pm0.012}$ | $0.420_{\pm0.014}$ | $0.669_{\pm0.014}$ | $0.961_{\pm0.008}$ | $0.747_{\pm0.017}$ | $0.497_{\pm0.023}$ | $0.746_{\pm0.011}$ | $0.622_{\pm0.019}$ |
| 5 | $0.751_{\pm0.019}$ | $0.445_{\pm0.024}$ | $0.703_{\pm0.029}$ | $0.965_{\pm0.015}$ | $0.754_{\pm0.016}$ | $0.519_{\pm0.027}$ | $0.763_{\pm0.019}$ | $0.637_{\pm0.021}$ |
| 6 | $0.757_{\pm0.011}$ | $0.445_{\pm0.019}$ | $0.698_{\pm0.028}$ | $0.958_{\pm0.009}$ | $0.765_{\pm0.016}$ | $0.520_{\pm0.026}$ | $0.762_{\pm0.012}$ | $0.642_{\pm0.021}$ |
| 7 | $0.764_{\pm0.016}$ | $0.451_{\pm0.023}$ | $0.710_{\pm0.023}$ | $0.972_{\pm0.005}$ | $0.757_{\pm0.019}$ | $0.517_{\pm0.030}$ | $0.772_{\pm0.013}$ | $0.637_{\pm0.025}$ |
| 8 | $0.764_{\pm0.016}$ | $0.459_{\pm0.026}$ | $0.715_{\pm0.035}$ | $0.964_{\pm0.009}$ | $0.770_{\pm0.022}$ | $0.531_{\pm0.026}$ | $0.772_{\pm0.017}$ | $0.650_{\pm0.024}$ |
| 9 | $0.750_{\pm0.024}$ | $0.436_{\pm0.035}$ | $0.683_{\pm0.041}$ | $0.965_{\pm0.008}$ | $0.749_{\pm0.023}$ | $0.498_{\pm0.037}$ | $0.758_{\pm0.023}$ | $0.623_{\pm0.029}$ |
| 10 | $0.759_{\pm0.011}$ | $0.443_{\pm0.018}$ | $0.692_{\pm0.023}$ | $0.967_{\pm0.007}$ | $0.754_{\pm0.011}$ | $0.508_{\pm0.024}$ | $0.765_{\pm0.009}$ | $0.631_{\pm0.017}$ |
| 11 | $0.744_{\pm0.011}$ | $0.418_{\pm0.015}$ | $0.656_{\pm0.019}$ | $0.972_{\pm0.005}$ | $0.758_{\pm0.016}$ | $0.516_{\pm0.019}$ | $0.751_{\pm0.008}$ | $0.637_{\pm0.016}$ |
| 12 | $0.758_{\pm0.010}$ | $0.434_{\pm0.008}$ | $0.682_{\pm0.015}$ | $0.978_{\pm0.007}$ | $0.764_{\pm0.019}$ | $0.502_{\pm0.031}$ | $0.765_{\pm0.007}$ | $0.633_{\pm0.024}$ |
| 13 | $0.769_{\pm0.014}$ | $0.442_{\pm0.021}$ | $0.686_{\pm0.031}$ | $0.979_{\pm0.004}$ | $0.763_{\pm0.023}$ | $0.501_{\pm0.021}$ | $0.771_{\pm0.014}$ | $0.632_{\pm0.021}$ |
| 14 | $0.759_{\pm0.010}$ | $0.410_{\pm0.012}$ | $0.639_{\pm0.014}$ | $0.988_{\pm0.009}$ | $0.755_{\pm0.008}$ | $0.497_{\pm0.013}$ | $0.757_{\pm0.007}$ | $0.626_{\pm0.010}$ |
| 15 | $0.759_{\pm0.010}$ | $0.401_{\pm0.014}$ | $0.614_{\pm0.021}$ | $0.991_{\pm0.004}$ | $0.758_{\pm0.016}$ | $0.501_{\pm0.017}$ | $0.752_{\pm0.010}$ | $0.629_{\pm0.016}$ |
| 16 | $0.738_{\pm0.011}$ | $0.311_{\pm0.012}$ | $0.456_{\pm0.017}$ | $0.995_{\pm0.002}$ | $0.725_{\pm0.014}$ | $0.481_{\pm0.007}$ | $0.705_{\pm0.006}$ | $0.603_{\pm0.007}$ |
| 17 | $0.794_{\pm0.021}$ | $0.453_{\pm0.025}$ | $0.682_{\pm0.036}$ | $0.991_{\pm0.003}$ | $0.755_{\pm0.020}$ | $0.511_{\pm0.023}$ | $0.784_{\pm0.017}$ | $0.633_{\pm0.020}$ |
| 18 | $0.738_{\pm0.015}$ | $0.275_{\pm0.020}$ | $0.388_{\pm0.029}$ | $0.995_{\pm0.002}$ | $0.729_{\pm0.022}$ | $0.526_{\pm0.022}$ | $0.688_{\pm0.013}$ | $0.627_{\pm0.022}$ |
| 19 | $0.786_{\pm0.012}$ | $0.419_{\pm0.013}$ | $0.655_{\pm0.017}$ | $0.989_{\pm0.003}$ | $0.745_{\pm0.011}$ | $0.499_{\pm0.018}$ | $0.771_{\pm0.009}$ | $0.622_{\pm0.013}$ |
| 20 | $0.820_{\pm0.009}$ | $0.530_{\pm0.016}$ | $0.808_{\pm0.022}$ | $0.988_{\pm0.003}$ | $0.754_{\pm0.017}$ | $0.494_{\pm0.011}$ | $0.826_{\pm0.009}$ | $0.624_{\pm0.014}$ |
| 21 | $0.843_{\pm0.011}$ | $0.559_{\pm0.014}$ | $0.843_{\pm0.014}$ | $0.991_{\pm0.002}$ | $0.755_{\pm0.011}$ | $0.510_{\pm0.017}$ | $0.845_{\pm0.008}$ | $0.633_{\pm0.013}$ |
| 22 | $0.808_{\pm0.008}$ | $0.516_{\pm0.013}$ | $0.804_{\pm0.014}$ | $0.987_{\pm0.005}$ | $0.785_{\pm0.010}$ | $0.518_{\pm0.007}$ | $0.818_{\pm0.006}$ | $0.652_{\pm0.005}$ |
| 23 | $0.805_{\pm0.013}$ | $0.515_{\pm0.011}$ | $0.805_{\pm0.012}$ | $0.987_{\pm0.005}$ | $0.785_{\pm0.009}$ | $0.524_{\pm0.011}$ | $0.817_{\pm0.009}$ | $0.655_{\pm0.008}$ |
| 24 | $0.818_{\pm0.020}$ | $0.533_{\pm0.029}$ | $0.819_{\pm0.022}$ | $0.986_{\pm0.008}$ | $0.770_{\pm0.020}$ | $0.513_{\pm0.024}$ | $0.827_{\pm0.015}$ | $0.642_{\pm0.022}$ |
| 25 | $0.824_{\pm0.011}$ | $0.542_{\pm0.007}$ | $0.827_{\pm0.007}$ | $0.992_{\pm0.003}$ | $0.788_{\pm0.012}$ | $0.544_{\pm0.005}$ | $0.834_{\pm0.005}$ | $0.666_{\pm0.007}$ |
| 26 | $0.821_{\pm0.010}$ | $0.533_{\pm0.007}$ | $0.814_{\pm0.008}$ | $0.989_{\pm0.003}$ | $0.791_{\pm0.010}$ | $0.537_{\pm0.016}$ | $0.828_{\pm0.005}$ | $0.664_{\pm0.012}$ |
| 27 | $0.834_{\pm0.021}$ | $0.545_{\pm0.028}$ | $0.821_{\pm0.027}$ | $0.991_{\pm0.004}$ | $0.799_{\pm0.016}$ | $0.577_{\pm0.016}$ | $0.836_{\pm0.015}$ | $0.688_{\pm0.015}$ |
| 28 | $0.821_{\pm0.008}$ | $0.515_{\pm0.013}$ | $0.794_{\pm0.012}$ | $0.991_{\pm0.004}$ | $0.805_{\pm0.019}$ | $0.563_{\pm0.026}$ | $0.822_{\pm0.007}$ | $0.684_{\pm0.022}$ |
| 29 | $0.837_{\pm0.007}$ | $0.545_{\pm0.016}$ | $0.819_{\pm0.020}$ | $0.991_{\pm0.002}$ | $0.810_{\pm0.008}$ | $0.584_{\pm0.016}$ | $0.837_{\pm0.008}$ | $0.697_{\pm0.010}$ |
| 30 | $0.845_{\pm0.012}$ | $0.546_{\pm0.013}$ | $0.815_{\pm0.015}$ | $0.991_{\pm0.002}$ | $0.828_{\pm0.017}$ | $0.613_{\pm0.024}$ | $0.839_{\pm0.007}$ | $0.720_{\pm0.020}$ |
| 31 | $0.829_{\pm0.005}$ | $0.521_{\pm0.013}$ | $0.796_{\pm0.012}$ | $0.991_{\pm0.004}$ | $0.818_{\pm0.020}$ | $0.578_{\pm0.032}$ | $0.826_{\pm0.004}$ | $0.698_{\pm0.026}$ |
| 32 | $0.834_{\pm0.010}$ | $0.530_{\pm0.017}$ | $0.793_{\pm0.018}$ | $0.988_{\pm0.008}$ | $0.817_{\pm0.011}$ | $0.583_{\pm0.013}$ | $0.828_{\pm0.011}$ | $0.700_{\pm0.011}$ |
| 33 | $0.827_{\pm0.013}$ | $0.524_{\pm0.021}$ | $0.788_{\pm0.026}$ | $0.988_{\pm0.003}$ | $0.805_{\pm0.019}$ | $0.571_{\pm0.026}$ | $0.824_{\pm0.012}$ | $0.688_{\pm0.022}$ |
| 34 | $0.823_{\pm0.012}$ | $0.518_{\pm0.019}$ | $0.774_{\pm0.019}$ | $0.984_{\pm0.003}$ | $0.794_{\pm0.012}$ | $0.556_{\pm0.014}$ | $0.818_{\pm0.011}$ | $0.675_{\pm0.011}$ |
| 35 | $0.708_{\pm0.011}$ | $0.388_{\pm0.017}$ | $0.627_{\pm0.017}$ | $0.941_{\pm0.011}$ | $0.742_{\pm0.023}$ | $0.480_{\pm0.026}$ | $0.719_{\pm0.010}$ | $0.611_{\pm0.024}$ |

# I MECHANISTIC ANALYSIS OF REPETITION IN PROTEIN LANGUAGE MODELS

To address why certain protein PLMs suffer more severely from repetitive degeneration, and to provide concrete hypotheses for the community, we conduct a systematic mechanistic analysis at two complementary levels: (i) layer-level probing, to trace how repetition-related information emerges and evolves across depth; and (ii) neuron-level correlation analysis, to examine whether repetition is localized or distributed, and how its strength differs across architectures. Together, these analyses reveal which models struggle most with repetition and offer insights into the underlying representational geometry.

## I.1 LAYER-LEVEL EVIDENCE OF MODEL-DEPENDENT REPETITION DYNAMICS

We begin by examining how repetition-related information is represented throughout the network using layer-wise linear probes. We construct a balanced dataset of positive (low-repetition) and

Table 20: Layer ablation (conditional) result of ProtGPT2 on Uniref50: repetition and bio-utility metrics per layer (mean $\pm$ std across seeds).

| layer | $H_{norm}$ | Dist-2 | Dist-3 | $R_{hpoly}$ | pLDDT | pTM | R | U |
|---|---|---|---|---|---|---|---|---|
| 0 | $0.805_{\pm0.025}$ | $0.610_{\pm0.018}$ | $0.836_{\pm0.019}$ | $0.972_{\pm0.010}$ | $0.838_{\pm0.017}$ | $0.577_{\pm0.035}$ | $0.833_{\pm0.015}$ | $0.708_{\pm0.023}$ |
| 1 | $0.774_{\pm0.026}$ | $0.570_{\pm0.030}$ | $0.791_{\pm0.031}$ | $0.954_{\pm0.018}$ | $0.807_{\pm0.006}$ | $0.546_{\pm0.025}$ | $0.802_{\pm0.024}$ | $0.676_{\pm0.015}$ |
| 2 | $0.672_{\pm0.021}$ | $0.425_{\pm0.019}$ | $0.635_{\pm0.031}$ | $0.918_{\pm0.010}$ | $0.769_{\pm0.015}$ | $0.467_{\pm0.025}$ | $0.707_{\pm0.016}$ | $0.618_{\pm0.019}$ |
| 3 | $0.806_{\pm0.012}$ | $0.613_{\pm0.021}$ | $0.846_{\pm0.016}$ | $0.971_{\pm0.008}$ | $0.827_{\pm0.009}$ | $0.572_{\pm0.017}$ | $0.835_{\pm0.011}$ | $0.699_{\pm0.012}$ |
| 4 | $0.809_{\pm0.010}$ | $0.624_{\pm0.014}$ | $0.857_{\pm0.014}$ | $0.978_{\pm0.007}$ | $0.835_{\pm0.012}$ | $0.585_{\pm0.021}$ | $0.842_{\pm0.009}$ | $0.710_{\pm0.015}$ |
| 5 | $0.812_{\pm0.016}$ | $0.623_{\pm0.015}$ | $0.852_{\pm0.009}$ | $0.974_{\pm0.009}$ | $0.830_{\pm0.010}$ | $0.576_{\pm0.009}$ | $0.841_{\pm0.007}$ | $0.703_{\pm0.005}$ |
| 6 | $0.813_{\pm0.007}$ | $0.623_{\pm0.018}$ | $0.852_{\pm0.014}$ | $0.972_{\pm0.007}$ | $0.830_{\pm0.014}$ | $0.576_{\pm0.021}$ | $0.841_{\pm0.007}$ | $0.703_{\pm0.015}$ |
| 7 | $0.822_{\pm0.013}$ | $0.642_{\pm0.020}$ | $0.864_{\pm0.016}$ | $0.977_{\pm0.004}$ | $0.831_{\pm0.009}$ | $0.591_{\pm0.018}$ | $0.851_{\pm0.010}$ | $0.711_{\pm0.013}$ |
| 8 | $0.812_{\pm0.009}$ | $0.624_{\pm0.014}$ | $0.853_{\pm0.010}$ | $0.974_{\pm0.004}$ | $0.827_{\pm0.010}$ | $0.581_{\pm0.030}$ | $0.841_{\pm0.007}$ | $0.704_{\pm0.020}$ |
| 9 | $0.817_{\pm0.009}$ | $0.632_{\pm0.014}$ | $0.856_{\pm0.011}$ | $0.977_{\pm0.010}$ | $0.821_{\pm0.007}$ | $0.576_{\pm0.017}$ | $0.846_{\pm0.008}$ | $0.699_{\pm0.010}$ |
| 10 | $0.823_{\pm0.008}$ | $0.640_{\pm0.022}$ | $0.867_{\pm0.017}$ | $0.981_{\pm0.003}$ | $0.827_{\pm0.013}$ | $0.579_{\pm0.020}$ | $0.852_{\pm0.008}$ | $0.703_{\pm0.014}$ |
| 11 | $0.807_{\pm0.012}$ | $0.604_{\pm0.019}$ | $0.828_{\pm0.016}$ | $0.975_{\pm0.008}$ | $0.823_{\pm0.021}$ | $0.572_{\pm0.016}$ | $0.833_{\pm0.010}$ | $0.697_{\pm0.018}$ |
| 12 | $0.815_{\pm0.011}$ | $0.618_{\pm0.015}$ | $0.838_{\pm0.014}$ | $0.982_{\pm0.004}$ | $0.818_{\pm0.013}$ | $0.562_{\pm0.031}$ | $0.842_{\pm0.008}$ | $0.690_{\pm0.021}$ |
| 13 | $0.807_{\pm0.010}$ | $0.603_{\pm0.015}$ | $0.818_{\pm0.009}$ | $0.985_{\pm0.003}$ | $0.811_{\pm0.014}$ | $0.545_{\pm0.013}$ | $0.834_{\pm0.005}$ | $0.678_{\pm0.013}$ |
| 14 | $0.818_{\pm0.009}$ | $0.604_{\pm0.023}$ | $0.816_{\pm0.021}$ | $0.992_{\pm0.004}$ | $0.805_{\pm0.015}$ | $0.549_{\pm0.023}$ | $0.840_{\pm0.010}$ | $0.677_{\pm0.018}$ |
| 15 | $0.813_{\pm0.008}$ | $0.569_{\pm0.030}$ | $0.764_{\pm0.018}$ | $0.993_{\pm0.004}$ | $0.797_{\pm0.014}$ | $0.538_{\pm0.019}$ | $0.824_{\pm0.011}$ | $0.667_{\pm0.016}$ |
| 16 | $0.775_{\pm0.010}$ | $0.432_{\pm0.027}$ | $0.565_{\pm0.029}$ | $0.991_{\pm0.004}$ | $0.758_{\pm0.006}$ | $0.499_{\pm0.019}$ | $0.755_{\pm0.011}$ | $0.628_{\pm0.012}$ |
| 17 | $0.839_{\pm0.013}$ | $0.612_{\pm0.015}$ | $0.799_{\pm0.024}$ | $0.994_{\pm0.003}$ | $0.792_{\pm0.020}$ | $0.533_{\pm0.006}$ | $0.846_{\pm0.009}$ | $0.663_{\pm0.010}$ |
| 18 | $0.789_{\pm0.011}$ | $0.411_{\pm0.018}$ | $0.529_{\pm0.021}$ | $0.991_{\pm0.003}$ | $0.755_{\pm0.009}$ | $0.528_{\pm0.025}$ | $0.750_{\pm0.008}$ | $0.641_{\pm0.016}$ |
| 19 | $0.832_{\pm0.008}$ | $0.586_{\pm0.028}$ | $0.790_{\pm0.029}$ | $0.993_{\pm0.004}$ | $0.791_{\pm0.019}$ | $0.522_{\pm0.012}$ | $0.837_{\pm0.012}$ | $0.657_{\pm0.014}$ |
| 20 | $0.869_{\pm0.013}$ | $0.709_{\pm0.009}$ | $0.925_{\pm0.010}$ | $0.994_{\pm0.002}$ | $0.790_{\pm0.006}$ | $0.514_{\pm0.014}$ | $0.893_{\pm0.007}$ | $0.652_{\pm0.004}$ |
| 21 | $0.877_{\pm0.006}$ | $0.717_{\pm0.015}$ | $0.932_{\pm0.007}$ | $0.990_{\pm0.005}$ | $0.787_{\pm0.009}$ | $0.535_{\pm0.019}$ | $0.897_{\pm0.004}$ | $0.661_{\pm0.009}$ |
| 22 | $0.856_{\pm0.011}$ | $0.688_{\pm0.008}$ | $0.912_{\pm0.004}$ | $0.992_{\pm0.005}$ | $0.829_{\pm0.005}$ | $0.565_{\pm0.013}$ | $0.882_{\pm0.005}$ | $0.697_{\pm0.008}$ |
| 23 | $0.854_{\pm0.006}$ | $0.689_{\pm0.014}$ | $0.914_{\pm0.010}$ | $0.993_{\pm0.003}$ | $0.829_{\pm0.011}$ | $0.569_{\pm0.026}$ | $0.883_{\pm0.004}$ | $0.699_{\pm0.014}$ |
| 24 | $0.869_{\pm0.008}$ | $0.715_{\pm0.012}$ | $0.937_{\pm0.006}$ | $0.991_{\pm0.006}$ | $0.818_{\pm0.014}$ | $0.569_{\pm0.019}$ | $0.896_{\pm0.005}$ | $0.693_{\pm0.015}$ |
| 25 | $0.860_{\pm0.005}$ | $0.709_{\pm0.020}$ | $0.927_{\pm0.008}$ | $0.991_{\pm0.006}$ | $0.829_{\pm0.018}$ | $0.565_{\pm0.023}$ | $0.889_{\pm0.004}$ | $0.697_{\pm0.016}$ |
| 26 | $0.858_{\pm0.012}$ | $0.691_{\pm0.022}$ | $0.912_{\pm0.022}$ | $0.990_{\pm0.004}$ | $0.832_{\pm0.014}$ | $0.584_{\pm0.023}$ | $0.883_{\pm0.010}$ | $0.708_{\pm0.016}$ |
| 27 | $0.870_{\pm0.010}$ | $0.711_{\pm0.008}$ | $0.927_{\pm0.006}$ | $0.993_{\pm0.005}$ | $0.825_{\pm0.013}$ | $0.577_{\pm0.026}$ | $0.894_{\pm0.006}$ | $0.701_{\pm0.019}$ |
| 28 | $0.858_{\pm0.018}$ | $0.681_{\pm0.020}$ | $0.907_{\pm0.016}$ | $0.992_{\pm0.003}$ | $0.840_{\pm0.012}$ | $0.595_{\pm0.032}$ | $0.881_{\pm0.012}$ | $0.717_{\pm0.020}$ |
| 29 | $0.861_{\pm0.015}$ | $0.687_{\pm0.021}$ | $0.902_{\pm0.014}$ | $0.992_{\pm0.004}$ | $0.828_{\pm0.004}$ | $0.587_{\pm0.023}$ | $0.883_{\pm0.010}$ | $0.708_{\pm0.011}$ |
| 30 | $0.871_{\pm0.010}$ | $0.692_{\pm0.013}$ | $0.904_{\pm0.014}$ | $0.994_{\pm0.002}$ | $0.847_{\pm0.018}$ | $0.613_{\pm0.019}$ | $0.888_{\pm0.007}$ | $0.730_{\pm0.016}$ |
| 31 | $0.859_{\pm0.006}$ | $0.685_{\pm0.023}$ | $0.902_{\pm0.014}$ | $0.993_{\pm0.002}$ | $0.845_{\pm0.013}$ | $0.609_{\pm0.013}$ | $0.882_{\pm0.008}$ | $0.727_{\pm0.012}$ |
| 32 | $0.862_{\pm0.004}$ | $0.691_{\pm0.012}$ | $0.899_{\pm0.008}$ | $0.993_{\pm0.003}$ | $0.842_{\pm0.009}$ | $0.601_{\pm0.018}$ | $0.883_{\pm0.004}$ | $0.721_{\pm0.011}$ |
| 33 | $0.857_{\pm0.008}$ | $0.681_{\pm0.016}$ | $0.903_{\pm0.009}$ | $0.992_{\pm0.003}$ | $0.836_{\pm0.009}$ | $0.591_{\pm0.031}$ | $0.881_{\pm0.006}$ | $0.714_{\pm0.019}$ |
| 34 | $0.857_{\pm0.010}$ | $0.684_{\pm0.008}$ | $0.893_{\pm0.009}$ | $0.990_{\pm0.001}$ | $0.843_{\pm0.005}$ | $0.597_{\pm0.015}$ | $0.879_{\pm0.005}$ | $0.720_{\pm0.005}$ |
| 35 | $0.789_{\pm0.012}$ | $0.590_{\pm0.016}$ | $0.815_{\pm0.010}$ | $0.963_{\pm0.013}$ | $0.821_{\pm0.009}$ | $0.562_{\pm0.016}$ | $0.818_{\pm0.010}$ | $0.692_{\pm0.007}$ |

negative (high-repetition) sequences, extract per-layer sequence representations, and train a logistic regression classifier at each layer.

Across both ProtGPT2 and ESM3 (Fig. 15, Fig. 16), we observe a consistent three-stage pattern:

**Early layers.** Probe Accuracy and AUC rise sharply, approaching $\simeq 1.0$ within only a few layers. This indicates that cues predictive of downstream repetition—such as short-range motifs, local copy patterns, or homopolymeric tendencies—are encoded extremely early and become globally accessible with little depth.

**Middle layers.** A shallow U-shaped dip appears: AUC remains high, but Accuracy decreases. This suggests that the repetition signal persists but becomes temporarily entangled with other semantic and structural features, reducing its linear margin while retaining rank-based separability.

**Final layers.** Probe performance recovers near the output, consistent with repetition being sharpened into a logit-aligned attractor state that directly influences next-token prediction.

Table 21: Layer ablation (unconditional) result of ESM3 on Uniref50: repetition and bio-utility metrics per layer (mean $\pm$ std across seeds).

| layer | $H_{norm}$ | Dist-2 | Dist-3 | $R_{hpoly}$ | pLDDT | pTM | R | U |
|---|---|---|---|---|---|---|---|---|
| 0 | $0.459_{\pm 0.017}$ | $0.203_{\pm 0.012}$ | $0.353_{\pm 0.024}$ | $0.561_{\pm 0.029}$ | $0.695_{\pm 0.013}$ | $0.477_{\pm 0.009}$ | $0.433_{\pm 0.021}$ | $0.586_{\pm 0.011}$ |
| 1 | $0.457_{\pm 0.016}$ | $0.204_{\pm 0.013}$ | $0.355_{\pm 0.024}$ | $0.559_{\pm 0.026}$ | $0.700_{\pm 0.017}$ | $0.482_{\pm 0.016}$ | $0.432_{\pm 0.020}$ | $0.591_{\pm 0.016}$ |
| 2 | $0.459_{\pm 0.016}$ | $0.204_{\pm 0.012}$ | $0.354_{\pm 0.025}$ | $0.561_{\pm 0.029}$ | $0.695_{\pm 0.017}$ | $0.475_{\pm 0.018}$ | $0.433_{\pm 0.021}$ | $0.585_{\pm 0.017}$ |
| 3 | $0.454_{\pm 0.010}$ | $0.200_{\pm 0.009}$ | $0.349_{\pm 0.018}$ | $0.557_{\pm 0.023}$ | $0.694_{\pm 0.013}$ | $0.478_{\pm 0.009}$ | $0.429_{\pm 0.014}$ | $0.586_{\pm 0.010}$ |
| 4 | $0.463_{\pm 0.013}$ | $0.205_{\pm 0.011}$ | $0.357_{\pm 0.022}$ | $0.564_{\pm 0.029}$ | $0.696_{\pm 0.015}$ | $0.478_{\pm 0.014}$ | $0.436_{\pm 0.019}$ | $0.587_{\pm 0.014}$ |
| 5 | $0.460_{\pm 0.016}$ | $0.203_{\pm 0.013}$ | $0.354_{\pm 0.027}$ | $0.570_{\pm 0.027}$ | $0.700_{\pm 0.019}$ | $0.481_{\pm 0.019}$ | $0.436_{\pm 0.020}$ | $0.591_{\pm 0.018}$ |
| 6 | $0.459_{\pm 0.019}$ | $0.203_{\pm 0.013}$ | $0.355_{\pm 0.025}$ | $0.567_{\pm 0.035}$ | $0.697_{\pm 0.021}$ | $0.482_{\pm 0.021}$ | $0.435_{\pm 0.024}$ | $0.589_{\pm 0.020}$ |
| 7 | $0.461_{\pm 0.018}$ | $0.204_{\pm 0.013}$ | $0.358_{\pm 0.027}$ | $0.571_{\pm 0.037}$ | $0.702_{\pm 0.018}$ | $0.488_{\pm 0.020}$ | $0.438_{\pm 0.025}$ | $0.595_{\pm 0.019}$ |
| 8 | $0.467_{\pm 0.010}$ | $0.207_{\pm 0.010}$ | $0.360_{\pm 0.018}$ | $0.576_{\pm 0.018}$ | $0.700_{\pm 0.014}$ | $0.486_{\pm 0.021}$ | $0.442_{\pm 0.012}$ | $0.593_{\pm 0.017}$ |
| 9 | $0.463_{\pm 0.008}$ | $0.204_{\pm 0.009}$ | $0.356_{\pm 0.017}$ | $0.576_{\pm 0.024}$ | $0.685_{\pm 0.020}$ | $0.476_{\pm 0.028}$ | $0.440_{\pm 0.013}$ | $0.581_{\pm 0.023}$ |
| 10 | $0.461_{\pm 0.009}$ | $0.204_{\pm 0.009}$ | $0.356_{\pm 0.015}$ | $0.572_{\pm 0.026}$ | $0.687_{\pm 0.017}$ | $0.476_{\pm 0.026}$ | $0.438_{\pm 0.015}$ | $0.582_{\pm 0.021}$ |
| 11 | $0.466_{\pm 0.014}$ | $0.206_{\pm 0.012}$ | $0.360_{\pm 0.020}$ | $0.582_{\pm 0.017}$ | $0.703_{\pm 0.010}$ | $0.484_{\pm 0.019}$ | $0.444_{\pm 0.015}$ | $0.594_{\pm 0.014}$ |
| 12 | $0.464_{\pm 0.012}$ | $0.206_{\pm 0.012}$ | $0.360_{\pm 0.020}$ | $0.572_{\pm 0.022}$ | $0.699_{\pm 0.015}$ | $0.484_{\pm 0.021}$ | $0.440_{\pm 0.016}$ | $0.591_{\pm 0.016}$ |
| 13 | $0.462_{\pm 0.011}$ | $0.206_{\pm 0.009}$ | $0.360_{\pm 0.015}$ | $0.576_{\pm 0.030}$ | $0.704_{\pm 0.013}$ | $0.489_{\pm 0.019}$ | $0.440_{\pm 0.016}$ | $0.597_{\pm 0.016}$ |
| 14 | $0.465_{\pm 0.014}$ | $0.207_{\pm 0.010}$ | $0.361_{\pm 0.021}$ | $0.582_{\pm 0.039}$ | $0.702_{\pm 0.018}$ | $0.488_{\pm 0.024}$ | $0.444_{\pm 0.022}$ | $0.595_{\pm 0.020}$ |
| 15 | $0.463_{\pm 0.009}$ | $0.206_{\pm 0.008}$ | $0.360_{\pm 0.016}$ | $0.575_{\pm 0.023}$ | $0.705_{\pm 0.012}$ | $0.490_{\pm 0.013}$ | $0.440_{\pm 0.014}$ | $0.597_{\pm 0.012}$ |
| 16 | $0.461_{\pm 0.018}$ | $0.206_{\pm 0.014}$ | $0.359_{\pm 0.025}$ | $0.564_{\pm 0.031}$ | $0.703_{\pm 0.020}$ | $0.484_{\pm 0.019}$ | $0.436_{\pm 0.022}$ | $0.593_{\pm 0.018}$ |
| 17 | $0.461_{\pm 0.019}$ | $0.206_{\pm 0.013}$ | $0.359_{\pm 0.025}$ | $0.561_{\pm 0.030}$ | $0.700_{\pm 0.023}$ | $0.483_{\pm 0.025}$ | $0.435_{\pm 0.022}$ | $0.592_{\pm 0.023}$ |
| 18 | $0.451_{\pm 0.012}$ | $0.199_{\pm 0.011}$ | $0.344_{\pm 0.017}$ | $0.546_{\pm 0.017}$ | $0.692_{\pm 0.012}$ | $0.475_{\pm 0.013}$ | $0.423_{\pm 0.014}$ | $0.583_{\pm 0.012}$ |
| 19 | $0.462_{\pm 0.018}$ | $0.203_{\pm 0.015}$ | $0.352_{\pm 0.026}$ | $0.564_{\pm 0.030}$ | $0.695_{\pm 0.017}$ | $0.477_{\pm 0.022}$ | $0.434_{\pm 0.023}$ | $0.586_{\pm 0.019}$ |
| 20 | $0.454_{\pm 0.027}$ | $0.198_{\pm 0.018}$ | $0.343_{\pm 0.028}$ | $0.561_{\pm 0.040}$ | $0.692_{\pm 0.021}$ | $0.471_{\pm 0.013}$ | $0.429_{\pm 0.029}$ | $0.581_{\pm 0.017}$ |
| 21 | $0.449_{\pm 0.021}$ | $0.199_{\pm 0.012}$ | $0.342_{\pm 0.020}$ | $0.537_{\pm 0.032}$ | $0.681_{\pm 0.016}$ | $0.460_{\pm 0.009}$ | $0.419_{\pm 0.022}$ | $0.570_{\pm 0.011}$ |
| 22 | $0.466_{\pm 0.010}$ | $0.208_{\pm 0.009}$ | $0.362_{\pm 0.016}$ | $0.568_{\pm 0.025}$ | $0.696_{\pm 0.013}$ | $0.486_{\pm 0.015}$ | $0.440_{\pm 0.016}$ | $0.591_{\pm 0.013}$ |
| 23 | $0.454_{\pm 0.014}$ | $0.200_{\pm 0.011}$ | $0.346_{\pm 0.017}$ | $0.558_{\pm 0.030}$ | $0.683_{\pm 0.020}$ | $0.468_{\pm 0.021}$ | $0.428_{\pm 0.019}$ | $0.575_{\pm 0.019}$ |
| 24 | $0.457_{\pm 0.025}$ | $0.201_{\pm 0.016}$ | $0.346_{\pm 0.030}$ | $0.580_{\pm 0.046}$ | $0.691_{\pm 0.029}$ | $0.476_{\pm 0.029}$ | $0.437_{\pm 0.031}$ | $0.584_{\pm 0.028}$ |
| 25 | $0.449_{\pm 0.010}$ | $0.194_{\pm 0.010}$ | $0.333_{\pm 0.016}$ | $0.567_{\pm 0.028}$ | $0.670_{\pm 0.009}$ | $0.444_{\pm 0.013}$ | $0.426_{\pm 0.015}$ | $0.557_{\pm 0.010}$ |
| 26 | $0.449_{\pm 0.012}$ | $0.197_{\pm 0.010}$ | $0.338_{\pm 0.018}$ | $0.570_{\pm 0.036}$ | $0.679_{\pm 0.020}$ | $0.455_{\pm 0.013}$ | $0.429_{\pm 0.020}$ | $0.567_{\pm 0.016}$ |
| 27 | $0.448_{\pm 0.009}$ | $0.195_{\pm 0.008}$ | $0.335_{\pm 0.012}$ | $0.584_{\pm 0.022}$ | $0.673_{\pm 0.015}$ | $0.456_{\pm 0.017}$ | $0.432_{\pm 0.011}$ | $0.564_{\pm 0.015}$ |
| 28 | $0.454_{\pm 0.015}$ | $0.199_{\pm 0.013}$ | $0.344_{\pm 0.018}$ | $0.590_{\pm 0.021}$ | $0.681_{\pm 0.012}$ | $0.460_{\pm 0.012}$ | $0.439_{\pm 0.015}$ | $0.570_{\pm 0.012}$ |
| 29 | $0.453_{\pm 0.012}$ | $0.202_{\pm 0.011}$ | $0.351_{\pm 0.022}$ | $0.572_{\pm 0.029}$ | $0.677_{\pm 0.022}$ | $0.461_{\pm 0.023}$ | $0.434_{\pm 0.019}$ | $0.569_{\pm 0.022}$ |
| 30 | $0.463_{\pm 0.013}$ | $0.208_{\pm 0.011}$ | $0.361_{\pm 0.020}$ | $0.590_{\pm 0.023}$ | $0.677_{\pm 0.014}$ | $0.463_{\pm 0.006}$ | $0.446_{\pm 0.017}$ | $0.570_{\pm 0.010}$ |
| 31 | $0.445_{\pm 0.007}$ | $0.193_{\pm 0.007}$ | $0.335_{\pm 0.013}$ | $0.575_{\pm 0.024}$ | $0.675_{\pm 0.009}$ | $0.457_{\pm 0.010}$ | $0.428_{\pm 0.012}$ | $0.566_{\pm 0.009}$ |
| 32 | $0.447_{\pm 0.026}$ | $0.199_{\pm 0.016}$ | $0.348_{\pm 0.030}$ | $0.574_{\pm 0.046}$ | $0.676_{\pm 0.029}$ | $0.472_{\pm 0.019}$ | $0.431_{\pm 0.031}$ | $0.574_{\pm 0.024}$ |
| 33 | $0.467_{\pm 0.011}$ | $0.205_{\pm 0.007}$ | $0.362_{\pm 0.018}$ | $0.595_{\pm 0.030}$ | $0.696_{\pm 0.013}$ | $0.487_{\pm 0.011}$ | $0.448_{\pm 0.017}$ | $0.591_{\pm 0.011}$ |
| 34 | $0.463_{\pm 0.015}$ | $0.203_{\pm 0.011}$ | $0.358_{\pm 0.021}$ | $0.599_{\pm 0.030}$ | $0.691_{\pm 0.014}$ | $0.482_{\pm 0.019}$ | $0.448_{\pm 0.019}$ | $0.586_{\pm 0.015}$ |
| 35 | $0.450_{\pm 0.022}$ | $0.200_{\pm 0.012}$ | $0.351_{\pm 0.024}$ | $0.562_{\pm 0.053}$ | $0.677_{\pm 0.028}$ | $0.475_{\pm 0.024}$ | $0.429_{\pm 0.030}$ | $0.576_{\pm 0.025}$ |
| 36 | $0.479_{\pm 0.011}$ | $0.214_{\pm 0.012}$ | $0.379_{\pm 0.020}$ | $0.602_{\pm 0.030}$ | $0.700_{\pm 0.017}$ | $0.490_{\pm 0.018}$ | $0.459_{\pm 0.017}$ | $0.595_{\pm 0.017}$ |
| 37 | $0.474_{\pm 0.013}$ | $0.215_{\pm 0.010}$ | $0.380_{\pm 0.020}$ | $0.588_{\pm 0.023}$ | $0.693_{\pm 0.018}$ | $0.491_{\pm 0.031}$ | $0.453_{\pm 0.016}$ | $0.592_{\pm 0.024}$ |
| 38 | $0.479_{\pm 0.015}$ | $0.214_{\pm 0.012}$ | $0.376_{\pm 0.023}$ | $0.593_{\pm 0.032}$ | $0.694_{\pm 0.018}$ | $0.487_{\pm 0.020}$ | $0.456_{\pm 0.021}$ | $0.591_{\pm 0.019}$ |
| 39 | $0.469_{\pm 0.013}$ | $0.205_{\pm 0.010}$ | $0.364_{\pm 0.019}$ | $0.594_{\pm 0.032}$ | $0.692_{\pm 0.016}$ | $0.470_{\pm 0.022}$ | $0.449_{\pm 0.020}$ | $0.581_{\pm 0.019}$ |
| 40 | $0.478_{\pm 0.011}$ | $0.211_{\pm 0.012}$ | $0.382_{\pm 0.019}$ | $0.622_{\pm 0.017}$ | $0.683_{\pm 0.018}$ | $0.470_{\pm 0.020}$ | $0.466_{\pm 0.014}$ | $0.576_{\pm 0.018}$ |
| 41 | $0.548_{\pm 0.005}$ | $0.252_{\pm 0.010}$ | $0.468_{\pm 0.014}$ | $0.678_{\pm 0.018}$ | $0.749_{\pm 0.015}$ | $0.555_{\pm 0.023}$ | $0.529_{\pm 0.011}$ | $0.652_{\pm 0.018}$ |
| 42 | $0.598_{\pm 0.018}$ | $0.273_{\pm 0.015}$ | $0.497_{\pm 0.027}$ | $0.724_{\pm 0.039}$ | $0.747_{\pm 0.011}$ | $0.502_{\pm 0.027}$ | $0.569_{\pm 0.025}$ | $0.624_{\pm 0.019}$ |
| 43 | $0.575_{\pm 0.008}$ | $0.257_{\pm 0.008}$ | $0.477_{\pm 0.014}$ | $0.768_{\pm 0.009}$ | $0.707_{\pm 0.008}$ | $0.461_{\pm 0.010}$ | $0.570_{\pm 0.008}$ | $0.584_{\pm 0.008}$ |
| 44 | $0.514_{\pm 0.009}$ | $0.230_{\pm 0.007}$ | $0.416_{\pm 0.013}$ | $0.651_{\pm 0.028}$ | $0.696_{\pm 0.010}$ | $0.446_{\pm 0.010}$ | $0.496_{\pm 0.015}$ | $0.571_{\pm 0.009}$ |
| 45 | $0.537_{\pm 0.007}$ | $0.225_{\pm 0.009}$ | $0.422_{\pm 0.016}$ | $0.785_{\pm 0.012}$ | $0.699_{\pm 0.011}$ | $0.434_{\pm 0.014}$ | $0.549_{\pm 0.007}$ | $0.567_{\pm 0.012}$ |
| 46 | $0.644_{\pm 0.016}$ | $0.300_{\pm 0.014}$ | $0.562_{\pm 0.019}$ | $0.882_{\pm 0.018}$ | $0.774_{\pm 0.014}$ | $0.546_{\pm 0.019}$ | $0.652_{\pm 0.014}$ | $0.660_{\pm 0.016}$ |
| 47 | $0.627_{\pm 0.015}$ | $0.290_{\pm 0.014}$ | $0.543_{\pm 0.019}$ | $0.866_{\pm 0.011}$ | $0.795_{\pm 0.011}$ | $0.554_{\pm 0.010}$ | $0.636_{\pm 0.013}$ | $0.674_{\pm 0.010}$ |

Importantly, this pattern is *stronger in ESM3*: its mid-layer dip is more pronounced and its late-layer sharpening more extreme. This aligns with our main experiments showing that ESM3 exhibits more severe repetition than ProtGPT2.

Table 22: Layer ablation (conditional) result of ESM3 on Uniref50: repetition and bio-utility metrics per layer (mean $\pm$ std across seeds).

| layer | $H_{norm}$ | Dist-2 | Dist-3 | $R_{hpoly}$ | pLDDT | pTM | R | U |
|---|---|---|---|---|---|---|---|---|
| 0 | $0.556_{\pm0.009}$ | $0.343_{\pm0.005}$ | $0.525_{\pm0.010}$ | $0.684_{\pm0.027}$ | $0.815_{\pm0.013}$ | $0.555_{\pm0.009}$ | $0.558_{\pm0.013}$ | $0.685_{\pm0.008}$ |
| 1 | $0.554_{\pm0.007}$ | $0.341_{\pm0.002}$ | $0.520_{\pm0.007}$ | $0.683_{\pm0.032}$ | $0.812_{\pm0.014}$ | $0.549_{\pm0.022}$ | $0.556_{\pm0.013}$ | $0.681_{\pm0.017}$ |
| 2 | $0.553_{\pm0.009}$ | $0.341_{\pm0.003}$ | $0.520_{\pm0.006}$ | $0.677_{\pm0.033}$ | $0.814_{\pm0.014}$ | $0.554_{\pm0.017}$ | $0.554_{\pm0.014}$ | $0.684_{\pm0.014}$ |
| 3 | $0.557_{\pm0.008}$ | $0.343_{\pm0.003}$ | $0.523_{\pm0.007}$ | $0.682_{\pm0.023}$ | $0.809_{\pm0.010}$ | $0.550_{\pm0.025}$ | $0.558_{\pm0.011}$ | $0.680_{\pm0.016}$ |
| 4 | $0.553_{\pm0.010}$ | $0.340_{\pm0.007}$ | $0.519_{\pm0.010}$ | $0.676_{\pm0.022}$ | $0.807_{\pm0.012}$ | $0.543_{\pm0.024}$ | $0.553_{\pm0.012}$ | $0.675_{\pm0.017}$ |
| 5 | $0.555_{\pm0.009}$ | $0.342_{\pm0.005}$ | $0.523_{\pm0.005}$ | $0.680_{\pm0.022}$ | $0.812_{\pm0.011}$ | $0.549_{\pm0.019}$ | $0.556_{\pm0.011}$ | $0.681_{\pm0.012}$ |
| 6 | $0.552_{\pm0.012}$ | $0.337_{\pm0.007}$ | $0.517_{\pm0.010}$ | $0.675_{\pm0.024}$ | $0.811_{\pm0.013}$ | $0.548_{\pm0.019}$ | $0.551_{\pm0.014}$ | $0.679_{\pm0.014}$ |
| 7 | $0.558_{\pm0.014}$ | $0.345_{\pm0.009}$ | $0.529_{\pm0.014}$ | $0.687_{\pm0.033}$ | $0.813_{\pm0.011}$ | $0.556_{\pm0.020}$ | $0.561_{\pm0.019}$ | $0.685_{\pm0.014}$ |
| 8 | $0.558_{\pm0.015}$ | $0.342_{\pm0.009}$ | $0.522_{\pm0.017}$ | $0.686_{\pm0.036}$ | $0.812_{\pm0.018}$ | $0.549_{\pm0.027}$ | $0.559_{\pm0.021}$ | $0.681_{\pm0.022}$ |
| 9 | $0.558_{\pm0.011}$ | $0.343_{\pm0.009}$ | $0.525_{\pm0.016}$ | $0.680_{\pm0.029}$ | $0.812_{\pm0.013}$ | $0.547_{\pm0.024}$ | $0.557_{\pm0.017}$ | $0.680_{\pm0.017}$ |
| 10 | $0.562_{\pm0.006}$ | $0.348_{\pm0.007}$ | $0.532_{\pm0.012}$ | $0.689_{\pm0.018}$ | $0.815_{\pm0.011}$ | $0.554_{\pm0.017}$ | $0.564_{\pm0.010}$ | $0.684_{\pm0.011}$ |
| 11 | $0.563_{\pm0.005}$ | $0.347_{\pm0.005}$ | $0.532_{\pm0.010}$ | $0.693_{\pm0.019}$ | $0.810_{\pm0.005}$ | $0.552_{\pm0.022}$ | $0.565_{\pm0.009}$ | $0.681_{\pm0.012}$ |
| 12 | $0.557_{\pm0.007}$ | $0.342_{\pm0.007}$ | $0.524_{\pm0.013}$ | $0.686_{\pm0.023}$ | $0.811_{\pm0.010}$ | $0.552_{\pm0.020}$ | $0.559_{\pm0.012}$ | $0.682_{\pm0.013}$ |
| 13 | $0.559_{\pm0.004}$ | $0.345_{\pm0.005}$ | $0.530_{\pm0.009}$ | $0.692_{\pm0.023}$ | $0.815_{\pm0.015}$ | $0.557_{\pm0.018}$ | $0.563_{\pm0.010}$ | $0.686_{\pm0.015}$ |
| 14 | $0.561_{\pm0.009}$ | $0.345_{\pm0.007}$ | $0.529_{\pm0.010}$ | $0.691_{\pm0.028}$ | $0.818_{\pm0.010}$ | $0.560_{\pm0.018}$ | $0.563_{\pm0.015}$ | $0.689_{\pm0.012}$ |
| 15 | $0.558_{\pm0.008}$ | $0.342_{\pm0.005}$ | $0.526_{\pm0.011}$ | $0.687_{\pm0.021}$ | $0.811_{\pm0.013}$ | $0.552_{\pm0.024}$ | $0.560_{\pm0.010}$ | $0.681_{\pm0.018}$ |
| 16 | $0.559_{\pm0.014}$ | $0.345_{\pm0.008}$ | $0.530_{\pm0.017}$ | $0.687_{\pm0.031}$ | $0.819_{\pm0.012}$ | $0.560_{\pm0.019}$ | $0.561_{\pm0.019}$ | $0.690_{\pm0.014}$ |
| 17 | $0.553_{\pm0.010}$ | $0.340_{\pm0.008}$ | $0.524_{\pm0.014}$ | $0.683_{\pm0.026}$ | $0.815_{\pm0.017}$ | $0.555_{\pm0.027}$ | $0.556_{\pm0.015}$ | $0.685_{\pm0.021}$ |
| 18 | $0.550_{\pm0.014}$ | $0.341_{\pm0.013}$ | $0.520_{\pm0.017}$ | $0.678_{\pm0.025}$ | $0.815_{\pm0.013}$ | $0.554_{\pm0.015}$ | $0.553_{\pm0.018}$ | $0.685_{\pm0.011}$ |
| 19 | $0.551_{\pm0.013}$ | $0.340_{\pm0.011}$ | $0.520_{\pm0.018}$ | $0.673_{\pm0.030}$ | $0.814_{\pm0.014}$ | $0.550_{\pm0.022}$ | $0.551_{\pm0.019}$ | $0.682_{\pm0.016}$ |
| 20 | $0.550_{\pm0.011}$ | $0.337_{\pm0.010}$ | $0.513_{\pm0.013}$ | $0.676_{\pm0.018}$ | $0.810_{\pm0.011}$ | $0.543_{\pm0.024}$ | $0.550_{\pm0.013}$ | $0.676_{\pm0.016}$ |
| 21 | $0.541_{\pm0.009}$ | $0.335_{\pm0.010}$ | $0.508_{\pm0.012}$ | $0.662_{\pm0.019}$ | $0.808_{\pm0.013}$ | $0.543_{\pm0.020}$ | $0.541_{\pm0.011}$ | $0.676_{\pm0.013}$ |
| 22 | $0.553_{\pm0.014}$ | $0.343_{\pm0.011}$ | $0.522_{\pm0.014}$ | $0.681_{\pm0.020}$ | $0.812_{\pm0.013}$ | $0.547_{\pm0.017}$ | $0.555_{\pm0.014}$ | $0.679_{\pm0.011}$ |
| 23 | $0.552_{\pm0.002}$ | $0.338_{\pm0.009}$ | $0.514_{\pm0.008}$ | $0.687_{\pm0.010}$ | $0.800_{\pm0.020}$ | $0.539_{\pm0.032}$ | $0.555_{\pm0.006}$ | $0.669_{\pm0.026}$ |
| 24 | $0.554_{\pm0.009}$ | $0.339_{\pm0.009}$ | $0.512_{\pm0.009}$ | $0.690_{\pm0.010}$ | $0.802_{\pm0.009}$ | $0.543_{\pm0.023}$ | $0.557_{\pm0.006}$ | $0.672_{\pm0.016}$ |
| 25 | $0.542_{\pm0.010}$ | $0.328_{\pm0.008}$ | $0.497_{\pm0.009}$ | $0.673_{\pm0.015}$ | $0.797_{\pm0.010}$ | $0.529_{\pm0.018}$ | $0.542_{\pm0.010}$ | $0.663_{\pm0.011}$ |
| 26 | $0.543_{\pm0.012}$ | $0.330_{\pm0.007}$ | $0.498_{\pm0.013}$ | $0.680_{\pm0.015}$ | $0.802_{\pm0.002}$ | $0.532_{\pm0.023}$ | $0.546_{\pm0.012}$ | $0.667_{\pm0.013}$ |
| 27 | $0.550_{\pm0.009}$ | $0.336_{\pm0.006}$ | $0.508_{\pm0.005}$ | $0.689_{\pm0.024}$ | $0.804_{\pm0.005}$ | $0.539_{\pm0.019}$ | $0.554_{\pm0.012}$ | $0.672_{\pm0.011}$ |
| 28 | $0.559_{\pm0.012}$ | $0.345_{\pm0.011}$ | $0.523_{\pm0.013}$ | $0.700_{\pm0.022}$ | $0.805_{\pm0.011}$ | $0.540_{\pm0.024}$ | $0.565_{\pm0.013}$ | $0.672_{\pm0.017}$ |
| 29 | $0.561_{\pm0.014}$ | $0.352_{\pm0.008}$ | $0.534_{\pm0.011}$ | $0.699_{\pm0.015}$ | $0.804_{\pm0.011}$ | $0.537_{\pm0.019}$ | $0.568_{\pm0.012}$ | $0.670_{\pm0.011}$ |
| 30 | $0.561_{\pm0.006}$ | $0.351_{\pm0.007}$ | $0.535_{\pm0.007}$ | $0.706_{\pm0.014}$ | $0.810_{\pm0.010}$ | $0.549_{\pm0.023}$ | $0.570_{\pm0.006}$ | $0.680_{\pm0.016}$ |
| 31 | $0.554_{\pm0.011}$ | $0.337_{\pm0.008}$ | $0.517_{\pm0.011}$ | $0.705_{\pm0.009}$ | $0.797_{\pm0.007}$ | $0.536_{\pm0.013}$ | $0.562_{\pm0.009}$ | $0.666_{\pm0.008}$ |
| 32 | $0.546_{\pm0.011}$ | $0.341_{\pm0.009}$ | $0.520_{\pm0.009}$ | $0.680_{\pm0.017}$ | $0.806_{\pm0.012}$ | $0.552_{\pm0.016}$ | $0.552_{\pm0.010}$ | $0.679_{\pm0.013}$ |
| 33 | $0.560_{\pm0.014}$ | $0.347_{\pm0.012}$ | $0.531_{\pm0.013}$ | $0.704_{\pm0.022}$ | $0.806_{\pm0.016}$ | $0.553_{\pm0.016}$ | $0.568_{\pm0.015}$ | $0.680_{\pm0.014}$ |
| 34 | $0.561_{\pm0.008}$ | $0.346_{\pm0.007}$ | $0.530_{\pm0.005}$ | $0.709_{\pm0.012}$ | $0.808_{\pm0.012}$ | $0.547_{\pm0.016}$ | $0.569_{\pm0.008}$ | $0.678_{\pm0.013}$ |
| 35 | $0.552_{\pm0.017}$ | $0.348_{\pm0.012}$ | $0.530_{\pm0.016}$ | $0.681_{\pm0.030}$ | $0.788_{\pm0.021}$ | $0.538_{\pm0.013}$ | $0.557_{\pm0.020}$ | $0.663_{\pm0.014}$ |
| 36 | $0.575_{\pm0.009}$ | $0.358_{\pm0.006}$ | $0.550_{\pm0.004}$ | $0.716_{\pm0.019}$ | $0.806_{\pm0.016}$ | $0.556_{\pm0.023}$ | $0.581_{\pm0.009}$ | $0.681_{\pm0.019}$ |
| 37 | $0.568_{\pm0.010}$ | $0.357_{\pm0.008}$ | $0.552_{\pm0.008}$ | $0.699_{\pm0.019}$ | $0.801_{\pm0.015}$ | $0.548_{\pm0.024}$ | $0.574_{\pm0.012}$ | $0.675_{\pm0.018}$ |
| 38 | $0.577_{\pm0.009}$ | $0.362_{\pm0.006}$ | $0.558_{\pm0.010}$ | $0.709_{\pm0.008}$ | $0.807_{\pm0.014}$ | $0.548_{\pm0.024}$ | $0.582_{\pm0.008}$ | $0.677_{\pm0.018}$ |
| 39 | $0.572_{\pm0.009}$ | $0.357_{\pm0.009}$ | $0.552_{\pm0.011}$ | $0.707_{\pm0.021}$ | $0.799_{\pm0.016}$ | $0.535_{\pm0.022}$ | $0.578_{\pm0.013}$ | $0.667_{\pm0.017}$ |
| 40 | $0.568_{\pm0.010}$ | $0.358_{\pm0.006}$ | $0.555_{\pm0.008}$ | $0.702_{\pm0.017}$ | $0.798_{\pm0.014}$ | $0.535_{\pm0.009}$ | $0.576_{\pm0.010}$ | $0.667_{\pm0.010}$ |
| 41 | $0.580_{\pm0.012}$ | $0.369_{\pm0.009}$ | $0.574_{\pm0.015}$ | $0.709_{\pm0.026}$ | $0.807_{\pm0.012}$ | $0.546_{\pm0.020}$ | $0.587_{\pm0.016}$ | $0.676_{\pm0.016}$ |
| 42 | $0.633_{\pm0.004}$ | $0.397_{\pm0.005}$ | $0.614_{\pm0.005}$ | $0.770_{\pm0.014}$ | $0.823_{\pm0.013}$ | $0.543_{\pm0.015}$ | $0.636_{\pm0.006}$ | $0.683_{\pm0.011}$ |
| 43 | $0.625_{\pm0.012}$ | $0.393_{\pm0.010}$ | $0.618_{\pm0.012}$ | $0.796_{\pm0.017}$ | $0.803_{\pm0.009}$ | $0.528_{\pm0.007}$ | $0.642_{\pm0.012}$ | $0.665_{\pm0.007}$ |
| 44 | $0.577_{\pm0.009}$ | $0.362_{\pm0.009}$ | $0.569_{\pm0.010}$ | $0.717_{\pm0.014}$ | $0.808_{\pm0.010}$ | $0.526_{\pm0.011}$ | $0.587_{\pm0.009}$ | $0.667_{\pm0.010}$ |
| 45 | $0.581_{\pm0.017}$ | $0.357_{\pm0.004}$ | $0.559_{\pm0.013}$ | $0.783_{\pm0.031}$ | $0.811_{\pm0.019}$ | $0.508_{\pm0.012}$ | $0.608_{\pm0.018}$ | $0.660_{\pm0.015}$ |
| 46 | $0.684_{\pm0.004}$ | $0.455_{\pm0.009}$ | $0.706_{\pm0.007}$ | $0.903_{\pm0.008}$ | $0.851_{\pm0.014}$ | $0.591_{\pm0.017}$ | $0.723_{\pm0.005}$ | $0.721_{\pm0.014}$ |
| 47 | $0.666_{\pm0.011}$ | $0.434_{\pm0.003}$ | $0.676_{\pm0.011}$ | $0.887_{\pm0.011}$ | $0.850_{\pm0.012}$ | $0.580_{\pm0.023}$ | $0.703_{\pm0.008}$ | $0.715_{\pm0.016}$ |

## I.2 NEURON-LEVEL EVIDENCE FOR ARCHITECTURE-DEPENDENT REPETITION SUBSPACES

To investigate whether repetition arises from localized malfunctioning units or distributed representational modes, we compute Pearson correlations between each neuron's activation and a continuous repetition score across all layers (Fig. 17, Fig. 18).

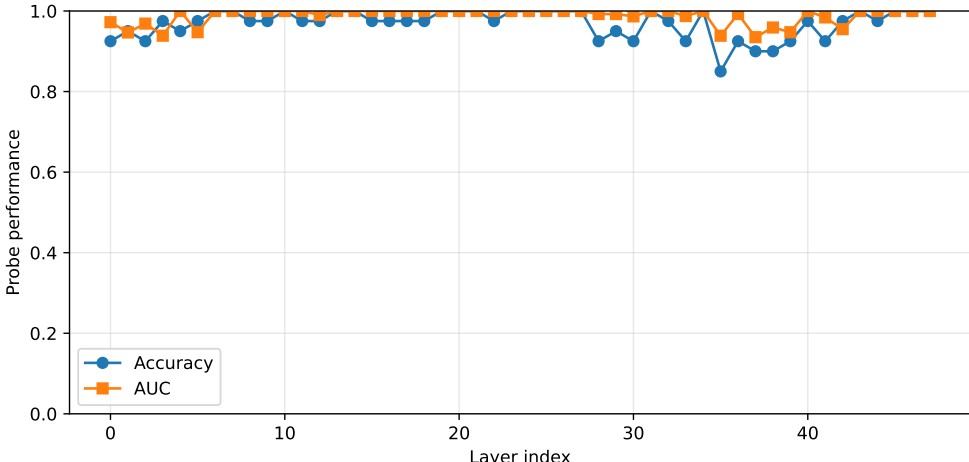

Figure 15: Layer-wise probe performance on ESM3. Accuracy and AUC rise rapidly in early layers, exhibit a mid-layer dip, and sharpen again near the output. The strength of this pattern reflects ESM3's stronger repetition tendency.

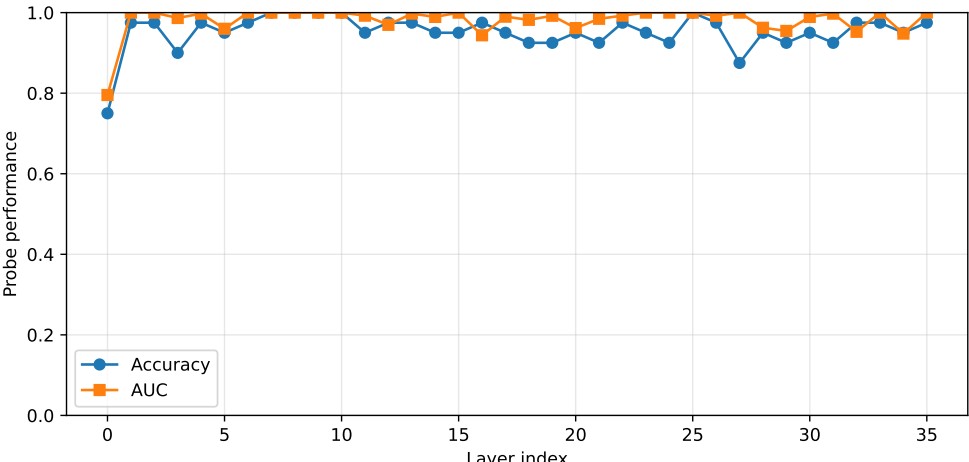

Figure 16: Layer-wise probe performance on ProtGPT2. The same qualitative pattern appears, but with overall weaker magnitude than in ESM3, matching its milder repetition behavior.

Across both models, the correlation distributions are centered near zero and lack isolated high-correlation outliers. This indicates that repetition is not driven by a small set of specialized neurons but reflects a *distributed* representational pattern spread across many dimensions. Repetition therefore aligns more naturally with a direction or subspace in activation space than with single-unit activations.

However, the *strength* of the repetition-related subspace varies strongly across architectures. Prot-GPT2 exhibits narrow, unimodal correlation distributions, whereas ESM3 shows noticeably broader and, in several layers, bimodal patterns. This indicates that ESM3's representations form a more polarized repetition-related subspace—strongly separating neurons that increase versus decrease activation during repetitive degeneration. This architectural difference mirrors our empirical finding that ESM3 struggles significantly more with repetition.

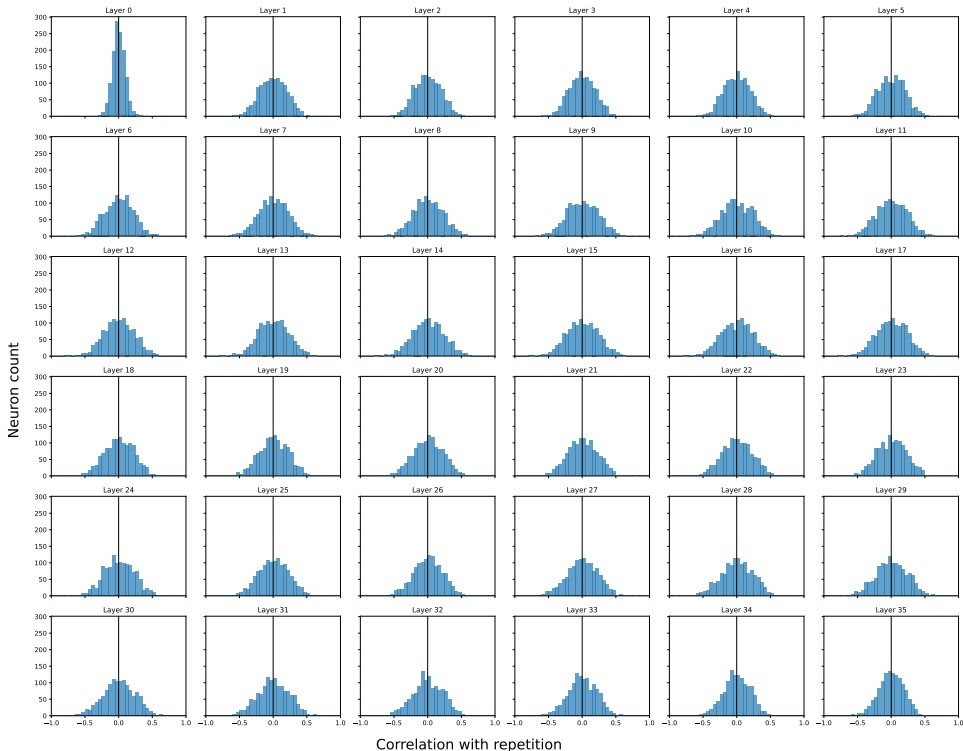

Figure 17: Neuron–repetition correlation distributions for ProtGPT2 across all layers. The distributions are narrow and unimodal, indicating a weaker and more diffuse repetition-related subspace.

### I.3 INSIGHTS AND RECOMMENDATIONS FOR THE PLM COMMUNITY

Our analyses reveal that repetition in protein PLMs should be viewed as a *global and distributed representational phenomenon* rather than a localized failure. Repetition emerges early, becomes most entangled mid-network, and is ultimately sharpened into a logit-aligned attractor state near the output. Architectural choices modulate the strength of this subspace: ESM3 exhibits a broader and more polarized repetition-related subspace than ProtGPT2, explaining its more severe repetitive degeneration.

These results suggest that future PLM development may benefit from diagnostics that track the geometry of repetition-related subspaces across layers, and from interventions that modulate representation trajectories rather than individual neurons or decoding heuristics. Understanding repetition at the level of activation space geometry provides a more principled foundation for mitigating degeneration in next-generation protein language models.

## J FURTHER EVALUATION ON ADDITIONAL PROTEIN LANGUAGE MODELS

We additionally include experiments on two representative backbones beyond those reported in the main paper: **ProGen2** and **DPLM**. These models span distinct architectures (autoregressive vs. diffusion language models), training scales, and inductive biases, thereby broadening the diversity of PLM paradigms assessed in our study.

Consistent with the main experiments in Sec. 5, we evaluate each model under both *unconditional* and *conditional* generation, using the same repetition score $R(x)$, utility score $U(x)$, and structural prediction protocol.

**Overall findings.** Across all addtional evaluated models, UCCS consistently:

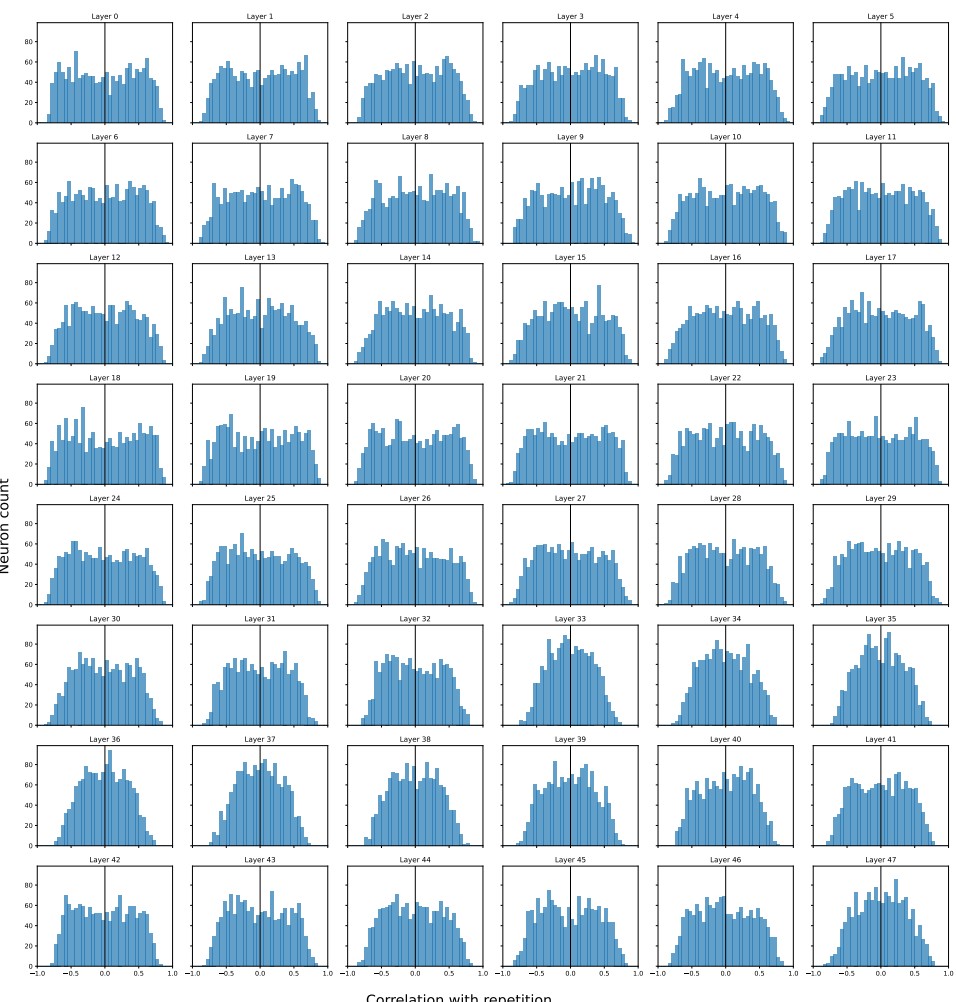

Figure 18: Neuron–repetition correlation distributions for ESM3 across all layers. The wider, and in several layers bimodal, distributions reveal a more polarized and substantially stronger repetition-related subspace, consistent with ESM3's more pronounced repetition tendencies.

1. reduces both motif-level and homopolymer-level repetition;
2. preserves or improves structural plausibility (pLDDT, pTM);
3. outperforms decoding heuristics and mechanism-level baselines.

## J.1 EVALUATION ON PROGEN2

We further validate UCCS on **ProGen2-base** (Madani et al., 2023) (`hugohrban/progen2-base`), an autoregressive PLM available on HuggingFace. Since ProGen2 follows the same generation paradigm as ProtGPT2, we adopt the *identical baselines and decoding settings* used for ProtGPT2, including temperature sampling, top-$p$ sampling, no-repeat-$n$-gram blocking, and repetition penalties.

Across all three datasets, UCCS provides consistent improvements over ProGen2's default sampling behavior. As shown in Table 23, decoding heuristics—including temperature scaling, top-$p$ sampling, and no-repeat-$n$-gram constraints—produce nearly identical repetition scores in the unconditional setting and offer limited control over pathological motifs; stronger constraints (e.g., no-repeat-$n$-gram) may increase $R$ but at the cost of reduced utility. In contrast, UCCS yields mod-

Table 23: ProGen2 results for two tasks with utility constraint: (a) unconditional generation, (b) conditional generation.

| Method | CATH | | UniRef50 | | SCOP | |
|---|---|---|---|---|---|---|
| | $R \uparrow$ | $U \uparrow$ | $R \uparrow$ | $U \uparrow$ | $R \uparrow$ | $U \uparrow$ |
| **(a) Unconditional generation** | | | | | | |
| Original Model | $0.890 \pm 0.002$ | $\checkmark 0.493 \pm 0.010$ | $0.890 \pm 0.002$ | $\checkmark 0.495 \pm 0.015$ | $0.890 \pm 0.002$ | $\checkmark 0.490 \pm 0.009$ |
| Temperature | $0.890 \pm 0.002$ | $\checkmark 0.495 \pm 0.016$ | $0.890 \pm 0.002$ | $0.492 \pm 0.014$ | $0.890 \pm 0.002$ | $\checkmark 0.496 \pm 0.011$ |
| Top-p Sampling | $0.890 \pm 0.002$ | $\checkmark 0.494 \pm 0.013$ | $0.890 \pm 0.002$ | $0.494 \pm 0.009$ | $0.890 \pm 0.002$ | $\checkmark 0.492 \pm 0.011$ |
| No Repeat N-gram | $0.989 \pm 0.001$ | $0.489 \pm 0.011$ | $0.914 \pm 0.001$ | $\checkmark 0.497 \pm 0.013$ | $0.989 \pm 0.001$ | $0.488 \pm 0.011$ |
| Repetition Penalty | $0.917 \pm 0.002$ | $\checkmark 0.495 \pm 0.016$ | $0.917 \pm 0.002$ | $0.490 \pm 0.014$ | $0.917 \pm 0.002$ | $\checkmark 0.491 \pm 0.017$ |
| Neuron Deactivation | $0.893 \pm 0.013$ | $0.490 \pm 0.025$ | $0.911 \pm 0.008$ | $0.486 \pm 0.030$ | $0.876 \pm 0.005$ | $0.478 \pm 0.010$ |
| Probe Steering | $0.896 \pm 0.002$ | $\checkmark 0.501 \pm 0.006$ | $0.894 \pm 0.003$ | $\checkmark 0.503 \pm 0.016$ | $0.894 \pm 0.003$ | $\checkmark 0.503 \pm 0.019$ |
| UCCS | $0.897 \pm 0.009$ | $\checkmark 0.511 \pm 0.014$ | $0.912 \pm 0.004$ | $\checkmark 0.499 \pm 0.014$ | $0.906 \pm 0.002$ | $\checkmark 0.500 \pm 0.007$ |
| **(b) Conditional generation** | | | | | | |
| Original Model | $0.919 \pm 0.003$ | $\checkmark 0.601 \pm 0.015$ | $0.919 \pm 0.005$ | $\checkmark 0.607 \pm 0.014$ | $0.924 \pm 0.002$ | $\checkmark 0.604 \pm 0.014$ |
| Temperature | $0.919 \pm 0.003$ | $\checkmark 0.606 \pm 0.014$ | $0.919 \pm 0.005$ | $\checkmark 0.607 \pm 0.011$ | $0.924 \pm 0.002$ | $\checkmark 0.610 \pm 0.019$ |
| Top-p Sampling | $0.919 \pm 0.003$ | $\checkmark 0.601 \pm 0.010$ | $0.913 \pm 0.005$ | $\checkmark 0.625 \pm 0.012$ | $0.918 \pm 0.004$ | $\checkmark 0.613 \pm 0.014$ |
| No Repeat N-gram | $0.932 \pm 0.001$ | $\checkmark 0.601 \pm 0.010$ | $0.922 \pm 0.004$ | $\checkmark 0.615 \pm 0.015$ | $0.935 \pm 0.002$ | $\checkmark 0.611 \pm 0.017$ |
| Repetition Penalty | $0.944 \pm 0.001$ | $0.582 \pm 0.022$ | $0.940 \pm 0.003$ | $0.570 \pm 0.010$ | $0.944 \pm 0.003$ | $0.589 \pm 0.027$ |
| Neuron Deactivation | $0.912 \pm 0.002$ | $0.599 \pm 0.017$ | $0.913 \pm 0.006$ | $0.598 \pm 0.008$ | $0.916 \pm 0.004$ | $\checkmark 0.605 \pm 0.012$ |
| Probe Steering | $0.920 \pm 0.002$ | $\checkmark 0.607 \pm 0.020$ | $0.921 \pm 0.004$ | $\checkmark 0.618 \pm 0.016$ | $0.925 \pm 0.001$ | $\checkmark 0.621 \pm 0.019$ |
| UCCS | $0.924 \pm 0.003$ | $\checkmark 0.610 \pm 0.005$ | $0.921 \pm 0.005$ | $\checkmark 0.611 \pm 0.013$ | $0.933 \pm 0.003$ | $\checkmark 0.610 \pm 0.015$ |

*Notes.* $R$: repetition score; $U$: biological utility. Cells marked with $\checkmark$ satisfy the utility constraint relative to the Original Model.

est yet reliable gains in repetition reduction while *maintaining or improving* biological utility under both unconditional and conditional generation. Notably, UCCS is the *only* method that simultaneously increases $R$ and satisfies the utility constraint across all evaluation conditions, confirming that its latent repetition direction generalizes robustly to ProGen2 despite its distinct autoregressive architecture and training corpus.

## J.2    EVALUATION ON DPLM

We next evaluate UCCS on the diffusion-based protein language model **DPLM-650M** (Wang et al., 2024) (`airkingbd/dplm_650m`). Because DPLM employs a fundamentally different generation mechanism compared to autoregressive or masked PLMs, we adopt decoding configurations tailored to its diffusion sampling pipeline. Specifically, we sweep four classes of baselines: (i) temperature scaling (0.7, 1.0, 1.3); (ii) sampling strategies {`gumbel_argmax`, `argmax`, `vanilla`}; (iii) disabling the model's internal resampling module; and (iv) resample-ratio sweeps (0.10–0.25). All experiments follow the protocol in Sec. 5, reporting repetition $R(x)$ and biological utility $U(x)$ across CATH, UniRef50, and SCOP.

Results in Table 24 show that DPLM achieves strong structural utility under default decoding, yet still exhibits notable repetition. Decoding heuristics such as temperature scaling and alternative sampling strategies introduce only minor changes to $R$ and generally leave $U$ unchanged. Disabling resampling substantially reduces repetition but induces pronounced degradation in utility, whereas adjusting the resampling ratio increases $R$ while yielding unstable or inconsistent $U$. In contrast, UCCS achieves the best overall trade-off across both unconditional and conditional settings, delivering substantial improvements in repetition reduction (e.g., $R = 0.863$–$0.879$ unconditional; $R = 0.881$–$0.894$ conditional) while *simultaneously improving* structural plausibility relative to the original model. Notably, UCCS is the only method that consistently satisfies the utility constraint across all datasets and tasks, despite the distinctive diffusion-generation dynamics of DPLM. These findings demonstrate that UCCS generalizes robustly beyond autoregressive and masked PLMs, effectively steering diffusion-based models as well.

## K    CASE STUDY: VISUALIZATIONS OF GENERATED PROTEINS

To complement the quantitative results in the main text, we provide qualitative case studies of generated protein structures. Specifically, we visualize generations from both ESM3 and PROTGPT2, evaluated on the UniRef50 dataset under *unconditional* and *conditional* sampling. For interpretabil-

Table 24: DPLM results for two tasks with utility constraint: (a) unconditional generation, (b) conditional generation.

| Method | CATH | | UniRef50 | | SCOP | |
|---|---|---|---|---|---|---|
| | $R \uparrow$ | $U \uparrow$ | $R \uparrow$ | $U \uparrow$ | $R \uparrow$ | $U \uparrow$ |
| **(a) Unconditional generation** | | | | | | |
| Original Model | $0.799 \pm 0.014$ | $\checkmark 0.841 \pm 0.016$ | $0.805 \pm 0.015$ | $\checkmark 0.835 \pm 0.016$ | $0.805 \pm 0.015$ | $\checkmark 0.838 \pm 0.014$ |
| Temperature | $0.806 \pm 0.007$ | $0.837 \pm 0.022$ | $0.805 \pm 0.015$ | $\checkmark 0.840 \pm 0.019$ | $0.805 \pm 0.015$ | $\checkmark 0.840 \pm 0.012$ |
| Gumbel Sampling | $0.806 \pm 0.007$ | $0.833 \pm 0.019$ | $0.805 \pm 0.015$ | $\checkmark 0.838 \pm 0.018$ | $0.805 \pm 0.015$ | $\checkmark 0.840 \pm 0.012$ |
| Disable Resample | $0.734 \pm 0.019$ | $0.820 \pm 0.017$ | $0.734 \pm 0.019$ | $0.827 \pm 0.018$ | $0.734 \pm 0.019$ | $0.820 \pm 0.020$ |
| Different Resample Ratio | $0.887 \pm 0.003$ | $0.832 \pm 0.013$ | $0.849 \pm 0.006$ | $\checkmark 0.853 \pm 0.005$ | $0.885 \pm 0.007$ | $0.831 \pm 0.011$ |
| Neuron Deactivation | $0.696 \pm 0.061$ | $0.704 \pm 0.069$ | $0.678 \pm 0.015$ | $0.695 \pm 0.028$ | $0.701 \pm 0.052$ | $0.713 \pm 0.067$ |
| Probe Steering | $0.811 \pm 0.005$ | $\checkmark 0.843 \pm 0.011$ | $0.815 \pm 0.015$ | $\checkmark 0.847 \pm 0.008$ | $0.815 \pm 0.006$ | $\checkmark 0.842 \pm 0.010$ |
| UCCS | $0.863 \pm 0.013$ | $\checkmark 0.860 \pm 0.008$ | $0.878 \pm 0.004$ | $\checkmark 0.873 \pm 0.010$ | $0.879 \pm 0.006$ | $\checkmark 0.875 \pm 0.008$ |
| **(b) Conditional generation** | | | | | | |
| Original Model | $0.794 \pm 0.017$ | $\checkmark 0.782 \pm 0.012$ | $0.809 \pm 0.009$ | $\checkmark 0.796 \pm 0.011$ | $0.806 \pm 0.007$ | $\checkmark 0.791 \pm 0.007$ |
| Temperature | $0.797 \pm 0.014$ | $\checkmark 0.782 \pm 0.008$ | $0.812 \pm 0.009$ | $\checkmark 0.797 \pm 0.010$ | $0.807 \pm 0.008$ | $\checkmark 0.792 \pm 0.009$ |
| Gumbel Sampling | $0.789 \pm 0.016$ | $0.776 \pm 0.011$ | $0.812 \pm 0.009$ | $\checkmark 0.798 \pm 0.010$ | $0.804 \pm 0.011$ | $\checkmark 0.793 \pm 0.007$ |
| Disable Resample | $0.670 \pm 0.038$ | $0.746 \pm 0.021$ | $0.702 \pm 0.018$ | $0.776 \pm 0.003$ | $0.704 \pm 0.042$ | $0.784 \pm 0.016$ |
| Different Resample Ratio | $0.902 \pm 0.003$ | $\checkmark 0.785 \pm 0.006$ | $0.901 \pm 0.005$ | $\checkmark 0.800 \pm 0.007$ | $0.902 \pm 0.005$ | $\checkmark 0.791 \pm 0.015$ |
| Neuron Deactivation | $0.745 \pm 0.044$ | $0.736 \pm 0.054$ | $0.731 \pm 0.006$ | $0.715 \pm 0.022$ | $0.764 \pm 0.025$ | $0.746 \pm 0.033$ |
| Probe Steering | $0.815 \pm 0.010$ | $\checkmark 0.796 \pm 0.015$ | $0.823 \pm 0.013$ | $\checkmark 0.809 \pm 0.011$ | $0.819 \pm 0.011$ | $\checkmark 0.812 \pm 0.011$ |
| UCCS | $0.881 \pm 0.012$ | $\checkmark 0.821 \pm 0.013$ | $0.893 \pm 0.008$ | $\checkmark 0.845 \pm 0.008$ | $0.894 \pm 0.009$ | $\checkmark 0.840 \pm 0.008$ |

*Notes.* $R$: repetition score; $U$: biological utility. Cells marked with $\checkmark$ satisfy the utility constraint relative to the Original Model.

ity, we select representative sequences with lengths roughly in the ranges of 64, 128, and 256 amino acids (not strictly exact) to illustrate how different intervention methods affect the generation process. These visualizations highlight structural diversity and foldability improvements under UCCS, compared to raw generations and baseline decoding strategies. **Color indicates pLDDT confidence:** blue = high ($> 90$), cyan/green = medium (70–90), orange/red = low ($< 70$).

As shown in Fig 25–28, our UCCS method consistently produces structures that are not only less repetitive but also exhibit higher biological utility compared to the *Original Model*. In particular, generations steered by UCCS display stronger foldability signals, with a higher proportion of blue regions (high pLDDT), while maintaining structural diversity. This demonstrates that our method avoids the degeneration artifacts common in raw generations and achieves a more favorable trade-off than baseline interventions.

## L    THE USE OF LARGE LANGUAGE MODELS (LLMs)

In preparing this manuscript, we made use of a large language model (LLM) as a general-purpose assistive tool. Specifically, the LLM was used for polishing the English presentation of the text (e.g., improving grammar, clarity, and flow), and for generating draft suggestions for sections such as the *Discussion and Conclusion*. All scientific ideas, methods, experiments, and analyses were conceived, designed, and conducted entirely by the authors. The LLM did not contribute to research ideation, experimental design, data analysis, or the generation of scientific claims.

The authors take full responsibility for the entire contents of the paper, including any text generated with LLM assistance. The use of LLMs does not imply authorship, and the LLM is not listed as a co-author.

Table 25: **ESM3 — Unconditional.** Rows are methods; columns are approximate sequence lengths.

| | ∼**64 aa** | ∼**128 aa** | ∼**256 aa** |
|---|---|---|---|
| UCCS | | | |
| Original Model | | | |
| Probe Steering | | | |
| Entropy-based Sampling | | | |
| NeuronDeactivation (TopK=256) | | | |
| TemperatureSampling (T=1.3) | | | |
| TopPSampling (p=0.95) | | | |

Table 26: **ESM3 — Conditional.** Rows are methods; columns are approximate sequence lengths.

| | ~64 aa | ~128 aa | ~256 aa |
|---|---|---|---|
| UCCS | | | |
| Original Model | | | |
| Probe Steering | | | |
| Entropy-based Sampling | | | |
| NeuronDeactivation (TopK=1024) | | | |
| TemperatureSampling (T=1.3) | | | |
| TopPSampling (p=0.95) | | | |

Table 27: **ProtGPT2 — Unconditional.** Rows are methods; columns are approximate sequence lengths.

| | ∼64 aa | ∼128 aa | ∼256 aa |
|---|---|---|---|
| UCCS | | | |
| Original Model | | | |
| Probe Steering | | | |
| NeuronDeactivation (TopK=8) | | | |
| NoRepeatNgram | | | |
| RepetitionPenalty (penalty=1.2) | | | |
| TemperatureSampling (T=1.3) | | | |
| TopPSampling (p=0.98) | | | |

Table 28: **ProtGPT2 — Conditional.** Rows are methods; columns are approximate sequence lengths.

| | ∼**64 aa** | ∼**128 aa** | ∼**256 aa** |
|---|---|---|---|
| UCCS |  |  |  |
| Original Model |  |  |  |
| Probe Steering |  |  |  |
| NeuronDeactivation (TopK=8) |  |  |  |
| NoRepeatNgram |  |  |  |
| RepetitionPenalty (penalty=1.1) |  |  |  |
| TemperatureSampling (T=1.3) |  |  |  |
| TopPSampling (p=0.98) |  |  |  |

