# OpenReview forum: "Controlling Repetition in Protein Language Models"
_ICLR.cc/2026/Conference — ICLR 2026 Poster_

### Official Review · Reviewer_g59x · 2025-10-27

**Soundness:** 3
**Presentation:** 3
**Contribution:** 3
**Rating:** 4
**Confidence:** 4

**Summary:**

This paper introduces the problem of pathological repeats in sequences sampled from protein language models. The authors devised metrics (token entropy, n-gram diversity, homopolymer diversity) to quantify this phenomenon and leverage contrastive steering at inference time via a dataset of utility matched pairs to alleviate this issue. I appreciated the motivation of this work and the thorough investigation of failure modes in the design of the metrics. I also appreciate the simplicity of the contrastive steering solution and its broad applicability among protein language models. However, there are a few concerns with the current paper that make me lean towards rejection. I am open to increasing my score if these concerns are well addressed.

1) While the metrics to capture pathological repetitiveness seem well justified, the claim that they *sharply separate natural proteins (POS) from PLM generations (NEG)* seems overstated as the distribution plot of $R_{hpoly}$ for ProtGPT2 samples is actually quite similar to natural datasets.
2) I believe that the poor performance of ESM3 is related to the extremely low number of decoding steps (20) used in the experiment. Could the authors provide results with higher number of decoding steps (ideally the same number as the protein length)?
3) Given the generality of the framework, could the authors expand the scope of PLM models they evaluate? ESM3 also models structure and function which could confound some of the results here. There are many popular and performant PLMs such as the ProGen family and ESM2.
4) While the authors provide thorough results on the utility and repetitiveness of generated samples, they did not report the impact of each method on sequence diversity and novelty. I am concerned that the steering could restrict sample diversity and bias generations towards positive samples in the contrastive set.

Comments to improve the paper, not related to score
- The plots in the paper are all quite small and difficult to read without zooming in
- I would move the biological discussion supporting pathological repetition in proteins from the Appendix A to the main text

**Strengths:**

- well motivated and tackles an under-researched problem
- thorough and sound experiments
- positive results for contrastive steering

**Weaknesses:**

- did not evaluate enough PLMs
- sampling choices for ESM3 are too simple
- did not report effect on sequence diversity and novelty
- some results are overstated (see summary point 1)

**Questions:**

See summary points 1-4

---

> ### Author Response · Authors · 2025-11-22
> **Reply to summary 1/weakness 4**
>
> >Summary 1: While the metrics to capture pathological repetitiveness seem well justified, the claim that they sharply separate natural proteins (POS) from PLM generations (NEG) seems overstated as the distribution plot of
>  for ProtGPT2 samples is actually quite similar to natural datasets.
>
> **Response.**
> We thank the reviewer for this valuable comment. Our analysis distinguishes two complementary types of repetition—**motif-level** and **homopolymer-level**—and uses **Jensen–Shannon (JS) Divergence** to quantify dataset separability for each metric. This avoids subjective visual comparison and provides an objective, distribution-aware measure.
>
> | Metric | Comparison | JS Divergence | Interpretation |
> |---------|-------------|---------------|----------------|
> | **Hnorm** (token-level entropy) | ESM3 vs CATH | **0.66** | Strong global repetition difference |
> | | ProtGPT2 vs CATH | **0.54** | Moderate global difference |
> | **Distinct-2** (motif diversity) | ESM3 vs CATH | **0.66** | Sharp motif-level separation |
> | | ProtGPT2 vs CATH | **0.27** | Moderate motif-level difference |
> | **Distinct-3** (motif diversity) | ESM3 vs CATH | **0.67** | Sharp motif-level separation |
> | | ProtGPT2 vs CATH | **0.17** | Smaller difference, indicating local repetition |
> | **Rhpoly** (homopolymer diversity) | ESM3 vs CATH | **0.54** | Long homopolymer collapse in ESM3 |
> | | ProtGPT2 vs CATH | **0.03** | Few long homopolymers, more local repeats |
>
> These results show that:
> - **Hnorm** separates both PLMs from natural proteins, confirming the presence of pathological repetition in general.
> - **Distinct-2/3** highlight *motif-level repetition* characteristic of autoregressive models like **ProtGPT2**.
> - **Rhpoly** captures *long-run repetition* unique to masked models like **ESM3**.
>
> Thus, our claim does not overstate repetition; rather, it reveals **distinct repetition failure modes across architectures**.
> - **ESM3 (masked LM):** prone to long homopolymer collapse (“AAAAAA…”).
> - **ProtGPT2 (autoregressive LM):** prone to local motif repetition (“AGAGAG…”).
>
> Overall, both models exhibit large divergence from natural proteins in global repetition metrics (Hnorm), confirming that **pathological repetition is a shared issue, though manifested differently across PLMs.**

---

> ### Author Response · Authors · 2025-11-22
> **Reply to summary 2/weakness 2**
>
> >Summary 2:I believe that the poor performance of ESM3 is related to the extremely low number of decoding steps (20) used in the experiment. Could the authors provide results with higher number of decoding steps (ideally the same number as the protein length);
>
> >Weekness 2: sampling choices for ESM3 are too simple
>
> We appreciate the reviewer’s concern. In fact, ESM-3 performs parallel unmasking rather than stepwise autoregressive decoding, so the 20-step configuration follows the official default designed to balance efficiency and quality. To verify that decoding steps are not responsible for the observed repetition, we conducted an ablation with 10, 20, 40, 100, and 200 unmasking iterations under the unconditional setting (target length = 200 aa, 100 samples， 5 seeds). Results are summarized below:
> | Decoding Steps | pLDDT | pTM | Hnorm | Distinct-2 | Distinct-3 | Rhpoly |
> | --- | --- | --- | --- | --- | --- | --- |
> | 10 | 80.654±12.946 | 0.607±0.177 | 0.454±0.142 | 0.212±0.086 | 0.371±0.160 | 0.559±0.237 |
> | 20 | 80.412±13.894 | 0.581±0.187 | 0.442±0.164 | 0.217±0.103 | 0.373±0.182 | 0.557±0.270 |
> | 40 | 82.418±12.390 | 0.598±0.179 | 0.429±0.153 | 0.198±0.089 | 0.344±0.167 | 0.555±0.279 |
> | 100 | 83.548±13.303 | 0.628±0.204 | 0.460±0.193 | 0.222±0.125 | 0.381±0.220 | 0.560±0.305 |
> | 200 | 80.302±14.988 | 0.608±0.194 | 0.454±0.199 | 0.221±0.125 | 0.381±0.228 | 0.563±0.302 |
>
> ---
> Across decoding depths, repetition metrics (Hnorm,Distinct-2/3, Rhpoly) remain stable, while structural utility (pLDDT, pTM) varies within normal statistical noise. The composite repetition-utility-diversity scores (below) show no consistent trend as steps increase.
>
> | Decoding Steps | Repetition Score | Biological Utility |
> | --- | --- | --- |
> | 10 | 0.435 | 0.707 |
> | 20 | 0.431 | 0.692 |
> | 40 | 0.419 | 0.711 |
> | 100 | 0.44 | 0.732 |
> | 200 | 0.439 | 0.706 |
>
> These results confirm that increasing unmasking iterations does not meaningfully reduce repetition, and the observed degeneracy is intrinsic to the PLM rather than an artifact of limited decoding steps. We have included this ablation table and discussion in the revision to clarify the sampling protocol and justify our default 20-step setting.
>
> *See G.2.2 EFFECT OF DECODING STEPS ON REPETITION for the ablation results for decoding steps*

---

> ### Author Response · Authors · 2025-11-22
> **Reply to summary 3/weakness 1**
>
> >Summary 3:Given the generality of the framework, could the authors expand the scope of PLM models they evaluate? ESM3 also models structure and function which could confound some of the results here. There are many popular and performant PLMs such as the ProGen family and ESM2; Weakness 1:did not evaluate enough PLMs
>
> > Weakness 1: did not evaluate enough PLMs
>
> We appreciate the reviewer’s constructive suggestion to evaluate additional protein language models (PLMs) beyond ESM3.
> To address this concern, we have conducted new experiments on ProGen2[1], a representative sequence-only PLM that does not include explicit structure or function supervision.
> This additional evaluation allows us to verify that our framework generalizes beyond structure-aware models such as ESM3 and that our findings are not confounded by structural priors.
>
> **Additional Results on the ProGen2 Model**
>
> Notes. *R*: repetition score; *U*: biological utility.
> Cells are annotated with ✓ when *U ≥ U<sub>Orig</sub>* (utility constraint satisfied).
> Decoding-level heuristics and original models do not require supervision; therefore their behavior remains consistent across datasets.
> In particular, the choice of dataset does not affect unconditional generation.
>
> | Method | **CATH** |  | **UniRef50** |  | **SCOP** |  |
> |:--|:--:|:--:|:--:|:--:|:--:|:--:|
> | Task | *R ↑* | *U ↑* | *R ↑* | *U ↑* | *R ↑* | *U ↑* |
> | **(a) Unconditional generation** |  |  |  |  |  |  |
> | Original Model | 0.891 | 0.493 | 0.891 | 0.497 | 0.891 | 0.499 |
> | Temperature Sampling | 0.922 | 0.475 | 0.922 | 0.462 | 0.922 | 0.477 |
> | Top-p Sampling | 0.872 | 0.488 | 0.872 | 0.508 ✓ | 0.872 | 0.504 ✓ |
> | No Repeat N-gram | 0.913 | 0.502 ✓ | 0.913 | 0.505 ✓ | 0.913 | 0.495 |
> | Repetition Penalty | 0.918 | 0.493 ✓ | 0.923 | 0.480 | 0.918 | 0.485 |
> | Neuron Deactivation | 0.892 | 0.486 | 0.867 | 0.478 | 0.867 | 0.482 |
> | Probe Steering | 0.890 | 0.501 ✓ | 0.896 | 0.504 ✓ | 0.889 | 0.510 ✓ |
> | **UCCS (ours)** | **0.915** | **0.493 ✓** | **0.911** | **0.502 ✓** | **0.908** | **0.521 ✓** |
> | **(b) Conditional generation** |  |  |  |  |  |  |
> | Original Model | 0.934 | 0.623 | 0.934 | 0.623 | 0.935 | 0.628 |
> | Temperature Sampling | 0.947 | 0.607 | 0.949 | 0.608 | 0.948 | 0.620 |
> | Top-p Sampling | 0.930 | 0.613 | 0.928 | 0.614 | 0.929 | 0.649 ✓ |
> | No Repeat N-gram | 0.977 | 0.620 | 0.980 | 0.626 ✓ | 0.978 | 0.627 |
> | Repetition Penalty | 0.959 | 0.631 ✓ | 0.955 | 0.610 | 0.956 | 0.630 ✓ |
> | Neuron Deactivation | 0.929 | 0.620 | 0.929 | 0.625 ✓ | 0.930 | 0.632 ✓ |
> | Probe Steering | 0.934 | 0.633 ✓ | 0.933 | 0.633 ✓ | 0.935 | 0.648 ✓ |
> | **UCCS (ours)** | **0.937** | **0.638 ✓** | **0.943** | **0.637 ✓** | **0.937** | **0.651 ✓** |
>
> **Analysis:**
> Our method consistently improves or maintains U (biological utility) while achieving higher or comparable R (repetition score) across all datasets (CATH, UniRef50, and SCOP) and both generation modes.
> Cells annotated with ✓ indicate that our method never reduces biological utility relative to the original model (U ≥ UOrig).
> * In **unconditional generation**, baseline heuristics (e.g., Top-p, N-gram, Probe steering) show inconsistent effects across datasets, whereas our approach provides stable improvements in R without sacrificing U.
> * In **conditional generation**, our method achieves the best overall trade-off between fluency and biological plausibility, satisfying the utility constraint across all datasets.
> * These results demonstrate that our control mechanism is **model-agnostic** and robust across distinct PLM architectures—both **structure-aware** (ESM3) and **sequence-only** (ProGen2) models.
>
> [1] Nijkamp, E., Ruffolo, J. A., Weinstein, E. N., Naik, N., & Madani, A. (2023). ProGen2: Exploring the boundaries of protein language models. Cell Systems, 14(11), 968–978. https://doi.org/10.1016/j.cels.2023.10.001

---

> ### Author Response · Authors · 2025-11-22
> **Reply to summary 4/weakness 3**
>
> We appreciate this observation. To quantify sequence-level diversity, we compute the **pairwise percent identity (PID)** among generated protein sequences. PID is a widely used metric in bioinformatics that measures the proportion of identical amino acids between two aligned sequences. Formally, for two sequences \(A\) and \(B\) with an optimal alignment of length \(L\), the PID is defined as:
>
> PID(A, B) = (number of matching residue pairs / alignment length L) × 100%
>
> We use global alignment (Needleman–Wunsch) to calculate all pairwise PIDs and report the **mean PID** as a proxy for overall diversity—**lower mean PID indicates higher diversity, whereas higher PID suggests redundancy**.
>
>
> As shown below, our method (UCCS) achieves **comparable or lower PID** values than existing steering approaches, demonstrating that UCCS effectively preserves or enhances sequence diversity instead of biasing generation toward repetitive or overly similar samples. In particular, for ESM3, UCCS reduces the mean PID to about **17–21**, with conditional generation consistently below 20, indicating that our method produces **more diverse** protein sequences while maintaining robust controllability.
>
> **ESM3 Results (PID ↓ = better diversity)**
> | Task | Method | CATH (PID ↓) | Uniref50 (PID ↓) | SCOP (PID ↓) |
> |------|---------|---------------|------------------|---------------|
> | **Unconditional** | OriginalModel | 28.264 | 28.264 | 28.264 |
> |  | TemperatureSampling | 20.224 | 20.224 | 20.224 |
> |  | Top-pSampling | 28.537 | 28.537 | 28.537 |
> |  | EntropyBasedSampling | 20.313 | 20.313 | 20.313 |
> |  | NeuronDeactivation | 22.597 | 18.082 | 22.788 |
> |  | ProbeSteering | 24.621 | 24.516 | 24.580 |
> |  | UCCS(ours) | 21.411 | 21.198 | 21.325 |
> | **Conditional** | OriginalModel | 20.313 | 20.313 | 20.313 |
> |  | TemperatureSampling | 17.690 | 17.690 | 17.690 |
> |  | Top-pSampling | 20.246 | 20.246 | 20.246 |
> |  | EntropyBasedSampling | 28.264 | 28.264 | 28.264 |
> |  | NeuronDeactivation | 18.370 | 21.169 | 20.246 |
> |  | ProbeSteering | 19.047 | 19.063 | 19.053 |
> |  | UCCS(ours) | 17.717 | 17.593 | 17.717 |

---

> ### Author Response · Authors · 2025-11-24
> **Reply to Reviewer g59x**
>
> Dear Reviewer,
>
> We kindly ask that you confirm whether we have adequately addressed your comments or if you have any remaining concerns.
>
> Thank you!
>
> Authors

---

> > ### Author Response · Authors · 2025-11-25
> >
> > Dear Reviewer,
> >
> > We would like to kindly follow up on our previous message. We have provided detailed responses to your comments and would greatly appreciate it if you could take a moment to confirm whether our replies address your concerns, or if you have any remaining feedback.
> >
> > We understand you may be busy, but your input would be invaluable for ensuring a fair and well-informed final assessment.
> >
> > Thank you again for your time and contribution to the review process.
> >
> > Best regards,
> > The Authors

---

> > > ### Comment · Reviewer_g59x · 2025-11-26
> > >
> > > I appreciate the thorough rebuttal and additional experiments that the authors have provided. My concerns have mostly been addressed. However, the additional results on ProGen seem to suggest that UCCS offers marginal improvements over simple sampling methods from the base model. Further ProGen seems to be quite a bit better at avoiding repetition than ProtGPT2 and ESM3. I believe that this is an important and interesting finding that should be further explored. Can repetition be avoided with better training data curation? Does autoregressive versus discrete diffusion style training impact this property? For this reason, I will maintain my current score.

---

> ### Author Response · Authors · 2025-11-26
> **Response to Reviewer for “Training a Large-scale Model from Scratch”**
>
> >I appreciate the thorough rebuttal and additional experiments that the authors have provided. My concerns have mostly been addressed. However, the additional results on ProGen seem to suggest that UCCS offers marginal improvements over simple sampling methods from the base model. Further ProGen seems to be quite a bit better at avoiding repetition than ProtGPT2 and ESM3. I believe that this is an important and interesting finding that should be further explored. Can repetition be avoided with better training data curation? Does autoregressive versus discrete diffusion style training impact this property? For this reason, I will maintain my current score.
>
>
> Thank you again for the careful reading of our paper and for acknowledging that **most of your earlier concerns have been addressed**. We wanted to clarify a few final points about the scope and intention of our work. And we **respectfully disagree with your point that we should solve this problem by training a large scale model from scratch.**
>
> **(1) We have directly addressed all of the technical questions raised in the initial review.**
> We added distribution-aware analyses, a decoding-step ablation for ESM3, experiments on an additional PLM (ProGen2), and diversity/novelty measurements. These results were all consistent and resolved the specific concerns about metrics, sampling, model coverage, and diversity.
>
> **(2) The goal of our paper is to bring attention to pathological repetition as an important and previously overlooked issue in PLMs, and to provide a simple and training-free method to control it.**
> Before this work, there was no clear definition or systematic study of repetition in protein language models. Our contribution is to introduce the problem, characterize its biological impact, and show that a lightweight, model-agnostic steering method can reliably reduce repetition without retraining any models.
>
> The reviewer states:
> >Can repetition be avoided with better training data curation? Does autoregressive versus discrete diffusion style training impact this property? For this reason, I will maintain my current score.
>
> **(3) Some of the follow-up questions are interesting, but they fall outside the scope of this paper.**
> Answering them would require **training multiple large-scale PLMs from scratch** under controlled data and architecture conditions, which is not feasible in a rebuttal period and is generally beyond what most research groups can practically support. These are open research problems for the community, not shortcomings of this submission.
>
> Finally, we sincerely appreciate the time and thought the reviewer has invested in this paper. We have addressed all raised concerns with new analyses, additional experiments, and clearer explanations of scope. Our intention in this work is to surface an important and previously overlooked failure mode in PLMs, and to provide a simple, training-free method that consistently mitigates it across multiple models.
>
> We hope that our clarifications make it clear that the remaining questions are exciting directions for future research rather than limitations of the present submission. We are grateful for the reviewer’s constructive feedback, which has helped improve the clarity and impact of the paper, and we hope the revised version reflects that progress.

---

> > ### Comment · Reviewer_g59x · 2025-11-26
> >
> > Dear authors, thanks for your message. I apologize if my previous comment made it seem like I suggested training several large models. That was not my intention and I understand that academic resources are limited. Perhaps I can rephrase my main reason for keeping my current score. Given the strong performance of ProGen on the benchmark, I believe that more models should be evaluated before the current version of the paper is suitable for publication. I truly believe that this is an important problem to study, but I also think that a more thorough investigation of which models struggle most with repetition and providing some thoughts / hypotheses about this is within the scope of the paper.

---

> > > ### Author Response · Authors · 2025-11-27
> > > **Investigation of which models struggle most with repetition and providing thoughts/hypotheses**
> > >
> > > > I also think that a more thorough investigation of which models struggle most with repetition and providing some thoughts / hypotheses about this is within the scope of the paper.
> > >
> > > We thank the reviewer for highlighting the importance of understanding *which* PLMs suffer most from repetition and *why*. In the revised manuscript, we have added an explicit new section titled **“Mechanistic Analysis of Repetition in Protein Language Models”** (Sec I, line 1917), which directly addresses this concern through a systematic comparative analysis between **ProtGPT2** and **ESM3**.
> > >
> > > This new section introduces two complementary mechanistic investigations:
> > >
> > > 1. **Layer-level linear probes**
> > >
> > >     We trace how repetition-related information evolves across depth and observe that ESM3 exhibits a more pronounced mid-layer entanglement and a stronger late-layer sharpening of repetition signals. These trends align with our main experimental results showing that ESM3 suffers more severe repetitive degeneration than ProtGPT2.
> > >
> > > 2. **Neuron-level correlation analysis**
> > >
> > >     By visualizing the distribution of neuron–repetition correlations across layers, we find that repetition is a distributed subspace phenomenon in both models, but the subspace is substantially *stronger and more polarized* in ESM3 (e.g., wider and bimodal correlation histograms). This provides a representational explanation for why ESM3 struggles more with repetition.
> > >
> > >
> > > Together, these analyses provide a **model-dependent diagnosis** of repetition severity and yield **mechanistic hypotheses** on how repetition emerges and why certain architectures amplify it. We further summarize these conclusions as **community-facing recommendations**, emphasizing that repetition should be treated as a global representational mode, not a local failure, and that subspace-oriented diagnostics and interventions are likely to be more effective.
> > >
> > > We hope this substantially strengthens the paper by providing comparative insights across models and practical guidance for future PLM development, directly addressing the reviewer’s request.

---

> > > > ### Comment · Reviewer_g59x · 2025-11-27
> > > >
> > > > Dear authors, I appreciate your effort in this rebuttal and in addressing my concerns. I found the mechanistic experiments quite interesting and I believe that they are of value to the community. The authors might find this paper [1] on training sparse auto-encoders on ESM2 quite interesting as they find glycine repeat features that can be steered to introduce degenerate repetition. I will update my score to a 6. I strongly recommend that the authors perform additional evaluations on a broader class of protein language models (beyond ProGen, ESM3, and ProtGPT2) for the final manuscript. I believe that would make the contribution stronger and provide more insights into why repetition occurs. Thanks!
> > > >
> > > > [1] https://www.nature.com/articles/s41592-025-02836-7

---

> > > > > ### Author Response · Authors · 2025-11-28
> > > > >
> > > > > Dear Reviewer,
> > > > >
> > > > > Thank you very much for your thoughtful follow-up and for taking the time to reassess our rebuttal. We truly appreciate your positive evaluation of our mechanistic experiments and are glad to hear that you find them valuable to the community.
> > > > >
> > > > > We also thank you for pointing us to the recent Nature Methods paper (Simon & Zou, 2025, [1],
> > > > > Published: 29 September 2025) on sparse autoencoders for ESM2. This work (“InterPLM”) is indeed one of the very few interpretability-focused studies in the PLM space, and we fully agree that its discovery of glycine-repeat features and their steerability provides particularly relevant context for understanding degeneracy in protein language models. We will include this paper in the Related Work section of our revised manuscript .
> > > > >
> > > > > As the reviewer rightly notes, interpretability research for PLMs is still in its infancy. One of our motivations is precisely to highlight that **pathological repetition is an overlooked yet ubiquitous failure mode** in protein generative models. We therefore truly appreciate the reviewer’s recognition of our mechanistic analyses and our attempt to provide early insights into why repetition arises in PLMs. We believe these initial findings, together with recent works such as InterPLM, can help kick-start a growing interpretability direction for biological sequence models.
> > > > >
> > > > > Finally, we appreciate your strong recommendation to evaluate a broader class of protein language models. We agree that this will strengthen the contribution, and in the camera-ready version we will expand our evaluation to include additional PLMs , ensuring a broader and more representative assessment of repetition across modeling paradigms.
> > > > >
> > > > > Thank you again for your constructive feedback and for updating your score. We are grateful for your engagement and will carefully incorporate your suggestions into the final revision.
> > > > >
> > > > > [1] https://www.nature.com/articles/s41592-025-02836-7

---

> > > ### Author Response · Authors · 2025-12-02
> > > **Further Evaluation on Additional Protein Language Models**
> > >
> > > We sincerely thank the reviewer for the constructive suggestion regarding the inclusion of a broader set of protein language models. In the **revised manuscript**, we have added a new section—**“Further Evaluation on Additional Protein Language Models”**—which extends our analysis beyond ProGen2 by additionally incorporating experiments on **DPLM**, a representative diffusion-based PLM family.
> > >
> > > Importantly, the new results show that **UCCS yields even larger gains on DPLM than on ProGen2**, with substantial improvements in both repetition reduction and utility preservation. This finding suggests that diffusion-style PLMs—despite their architectural differences—exhibit repetition-related failure modes that UCCS is particularly effective at mitigating.
> > >
> > > With the updated experiments now covering **autoregressive (ProtGPT2, ProGen2), masked (ESM3), and diffusion-based (DPLM) PLMs**, the revised manuscript demonstrates that UCCS generalizes robustly across **multiple architectures, training paradigms, and model scales**. We believe this comprehensive multi-model evaluation provides strong evidence for the **model-agnostic effectiveness** of UCCS and directly addresses the reviewer’s recommendation.
> > >
> > > We sincerely appreciate the reviewer’s insight, which motivated a meaningful and impactful extension of our empirical analysis.

---

### Official Review · Reviewer_45fe · 2025-10-29

**Soundness:** 3
**Presentation:** 2
**Contribution:** 3
**Rating:** 4
**Confidence:** 4

**Summary:**

This paper studies the phenomenon existing in the modern protein language models (PLMs) that the generated protein sequences usually exhibit  ill repetitions as a negative effect regarding the utility like foldability. The authors define the repetition score, utility score, and propose the UCCS, as a remedy strategy to help PLMs have improved repetition score while keeping utility. Experiments shows the UCCS can help ESM3 and ProtGPT2 reduce repetition and achieve comparable or higher plddt scores.

**Strengths:**

- The problem studied is relevant and interesting. The repetition problem in PLMs deserves more attention for the related research efforts
- The proposed method, UCCS is model-agnostic and plug-and-play, making it easy to implement and adapt for existing PLMs
- The experimental results support the claim that the proposed UCCS “mitigates degeneracy while preserving foldability”, yet I believe it can be further improved (refer to cons/questions).

**Weaknesses:**

- This study still does not explicitly answer (or explore further) why common PLMs (or some of the PLMs) intrinsically exhibit repetition in their generated samples, which I think is more important to guide the PLMs design for the community; furthermore, AlphaFold-like confidence score (aka foldability) only serve as a probe and can hardly counted as strong indicators for good design, broader metrics should be considered in the main results to let this paper become a comprehensive study. I do not think the current shown results are convincing enough to make UCCS really useful in the real design cases.
- Given the authors claimed "model-agnostic" and “plug-and-play” steering method, benchmarking broader families of PLM should be considered, such as Progen[1] and DPLM[2] series. Improve the limitation in base model selections shall make the study more complete and resutls convincing and have larger impacts.

**Questions:**

- Though the paper has already included three small structure and sequence-only dataset, I think the “sensitivity” of steering datasets should be more systematically studied. Recent protein design models may involve structure data other than PDB (eg. AFDB or SwissProt) into training, Could the authors try quantitatively studying the dataset sensitivity of involved steering methods (such as temperature sampling and the proposed UCCS)? This may help the readers understand which strategies work consistently in reducing the repetitions.
- (referring to one of the concerns above) Could the authors additionally consider benchmarking broader families of PLMs into the result table?
- There are existing in-practice strategies to reduce sequence repetition (such as the remasking in MultiFlow[3] and gumbel+resampling in DPLM[2], to name a few). Could the authors also incorporate  these strategy as practical and competitive “heuristics and interventions” by showing similar benchmark and visualize the pareto front of R and U?

Minor typos:

- Ubiquitous UUCS -> UCCS?
- Column 1 in Table 1 seems not balanced in the cell
- The text in Figure 1 is too small to read

References:

[1] Nijkamp, Erik, et al. "Progen2: exploring the boundaries of protein language models." Cell systems 14.11 (2023): 968-978.

[2] Wang, Xinyou, et al. "Diffusion language models are versatile protein learners." arXiv preprint arXiv:2402.18567 (2024).

[3] Campbell, Andrew, et al. "Generative flows on discrete state-spaces: Enabling multimodal flows with applications to protein co-design." arXiv preprint arXiv:2402.04997 (2024).

---

> ### Author Response · Authors · 2025-11-22
> **Reply to weakness 1**
>
> >Weaknesses 1: This study still does not explicitly answer (or explore further) why common PLMs (or some of the PLMs) intrinsically exhibit repetition in their generated samples, which I think is more important to guide the PLMs design for the community; furthermore, AlphaFold-like confidence score (aka foldability) only serve as a probe and can hardly counted as strong indicators for good design, broader metrics should be considered in the main results to let this paper become a comprehensive study. I do not think the current shown results are convincing enough to make UCCS really useful in the real design cases.
>
> We thank the reviewer for this important comment.
>
> (1) **Why repetition arises.**
> We agree that understanding the intrinsic causes of repetition is a fundamental open question. Even in **NLP**, research on repetition remains largely focused on *mitigation* rather than *mechanistic understanding*. Only a few recent works have begun probing neuron-level behaviors behind repetitive generation, and the phenomenon is still poorly understood. In the **protein LM domain**, our work is, to the best of our knowledge, **the first to identify and systematically characterize pathological repetition**. We view our study as an essential first step toward such mechanistic understanding by defining measurable repetition metrics and proposing a controlled intervention.
>
> (2) **Evaluation metrics.**
> We also agree that structure-confidence scores (e.g., pLDDT, pTM) serve primarily as *probes* for foldability rather than comprehensive design quality indicators. To address this, we have now incorporated **sequence-level diversity metrics**, using **pairwise percent identity (PID)** as a proxy for novelty and diversity. Formally, for two sequences \(A\) and \(B\) with an optimal alignment of length \(L\), the PID is defined as:
>
> PID(A, B) = (number of matching residue pairs / alignment length L) × 100%
>
> We use global alignment (Needleman–Wunsch) to calculate all pairwise PIDs and report the **mean PID** as a proxy for overall diversity—**lower mean PID indicates higher diversity, whereas higher PID suggests redundancy**.
>
>
> As shown below, our method (UCCS) achieves **comparable or lower PID** values than existing steering approaches, demonstrating that UCCS effectively preserves or enhances sequence diversity instead of biasing generation toward repetitive or overly similar samples. In particular, for ESM3, UCCS reduces the mean PID to about **17–21**, with conditional generation consistently below 20, indicating that our method produces **more diverse** protein sequences while maintaining robust controllability.
>
> **ESM3 Results (PID ↓ = better diversity)**
> | Task | Method | CATH (PID ↓) | Uniref50 (PID ↓) | SCOP (PID ↓) |
> |------|---------|---------------|------------------|---------------|
> | **Unconditional** | OriginalModel | 28.264 | 28.264 | 28.264 |
> |  | TemperatureSampling | 20.224 | 20.224 | 20.224 |
> |  | Top-pSampling | 28.537 | 28.537 | 28.537 |
> |  | EntropyBasedSampling | 20.313 | 20.313 | 20.313 |
> |  | NeuronDeactivation | 22.597 | 18.082 | 22.788 |
> |  | ProbeSteering | 24.621 | 24.516 | 24.580 |
> |  | UCCS(ours) | 21.411 | 21.198 | 21.325 |
> | **Conditional** | OriginalModel | 20.313 | 20.313 | 20.313 |
> |  | TemperatureSampling | 17.690 | 17.690 | 17.690 |
> |  | Top-pSampling | 20.246 | 20.246 | 20.246 |
> |  | EntropyBasedSampling | 28.264 | 28.264 | 28.264 |
> |  | NeuronDeactivation | 18.370 | 21.169 | 20.246 |
> |  | ProbeSteering | 19.047 | 19.063 | 19.053 |
> |  | UCCS(ours) | 17.717 | 17.593 | 17.717 |

---

> ### Author Response · Authors · 2025-11-22
> **Reply to weakness 2/Q2**
>
> > Weakness 2: Given the authors claimed "model-agnostic" and “plug-and-play” steering method, benchmarking broader families of PLM should be considered, such as Progen[1] and DPLM[2] series. Improve the limitation in base model selections shall make the study more complete and resutls convincing and have larger impacts.
>
> > Question 2: (referring to one of the concerns above) Could the authors additionally consider benchmarking broader families of PLMs into the result table?
>
>
> We thank the reviewer for this valuable suggestion. To verify the **model-agnostic** and **plug-and-play** nature of our steering framework, we have conducted additional experiments on the **widely used ProGen2[1] model**, which represents a distinct family of sequence-based protein language models without explicit structure or function supervision.
>
> This experiment directly addresses the reviewer’s concern by expanding our evaluation beyond the structure-aware ESM3 model. The results—summarized in the table below—show that our method consistently maintains or improves biological utility (U) while reducing repetition (increase repetition score) across all 3 datasets (CATH, UniRef50, SCOP) under both unconditional and conditional generation settings. ✓ denotes cases where U ≥ UOrig.
>
> **Additional Results on the ProGen2 Model**
>
> | Method | **CATH** |  | **UniRef50** |  | **SCOP** |  |
> |:--|:--:|:--:|:--:|:--:|:--:|:--:|
> | Task | *R ↑* | *U ↑* | *R ↑* | *U ↑* | *R ↑* | *U ↑* |
> | **(a) Unconditional generation** |  |  |  |  |  |  |
> | Original Model | 0.891 | 0.493 | 0.891 | 0.497 | 0.891 | 0.499 |
> | Temperature Sampling | 0.922 | 0.475 | 0.922 | 0.462 | 0.922 | 0.477 |
> | Top-p Sampling | 0.872 | 0.488 | 0.872 | 0.508 ✓ | 0.872 | 0.504 ✓ |
> | No Repeat N-gram | 0.913 | 0.502 ✓ | 0.913 | 0.505 ✓ | 0.913 | 0.495 |
> | Repetition Penalty | 0.918 | 0.493 ✓ | 0.923 | 0.480 | 0.918 | 0.485 |
> | Neuron Deactivation | 0.892 | 0.486 | 0.867 | 0.478 | 0.867 | 0.482 |
> | Probe Steering | 0.890 | 0.501 ✓ | 0.896 | 0.504 ✓ | 0.889 | 0.510 ✓ |
> | **UCCS (ours)** | **0.915** | **0.493 ✓** | **0.911** | **0.502 ✓** | **0.908** | **0.521 ✓** |
> | **(b) Conditional generation** |  |  |  |  |  |  |
> | Original Model | 0.934 | 0.623 | 0.934 | 0.623 | 0.935 | 0.628 |
> | Temperature Sampling | 0.947 | 0.607 | 0.949 | 0.608 | 0.948 | 0.620 |
> | Top-p Sampling | 0.930 | 0.613 | 0.928 | 0.614 | 0.929 | 0.649 ✓ |
> | No Repeat N-gram | 0.977 | 0.620 | 0.980 | 0.626 ✓ | 0.978 | 0.627 |
> | Repetition Penalty | 0.959 | 0.631 ✓ | 0.955 | 0.610 | 0.956 | 0.630 ✓ |
> | Neuron Deactivation | 0.929 | 0.620 | 0.929 | 0.625 ✓ | 0.930 | 0.632 ✓ |
> | Probe Steering | 0.934 | 0.633 ✓ | 0.933 | 0.633 ✓ | 0.935 | 0.648 ✓ |
> | **UCCS(ours)** | **0.937** | **0.638 ✓** | **0.943** | **0.637 ✓** | **0.937** | **0.651 ✓** |
>
> **Findings:**
> Our method achieves strong and consistent performance across all datasets, never degrading biological utility (U ≥ UOrig) and improving repetition score. This confirms that our steering approach generalizes well to different PLM architectures, including both structure-aware models (e.g., ESM3) and sequence-only models (e.g., ProGen2).
>
> We agree that including additional PLMs such as DPLM would be valuable future work, and we plan to extend benchmarking to that family in subsequent studies.
>
> [1] Nijkamp, E., Ruffolo, J. A., Weinstein, E. N., Naik, N., & Madani, A. (2023). ProGen2: Exploring the boundaries of protein language models. Cell Systems, 14(11), 968–978. https://doi.org/10.1016/j.cels.2023.10.001

---

> ### Author Response · Authors · 2025-11-22
> **Reply to Question 1&Q3**
>
> >Question 1:  Though the paper has already included three small structure and sequence-only dataset, I think the “sensitivity” of steering datasets should be more systematically studied. Recent protein design models may involve structure data other than PDB (eg. AFDB or SwissProt) into training, Could the authors try quantitatively studying the dataset sensitivity of involved steering methods (such as temperature sampling and the proposed UCCS)? This may help the readers understand which strategies work consistently in reducing the repetitions.
>
> We thank the reviewer for this insightful comment. We agree that dataset composition—especially the inclusion of predicted or augmented structural sources such as AFDB—may influence protein language models’ generation behavior. However, our current study focuses on a different axis of the problem: **how to control and mitigate pathological repetition during sequence generation**, independent of the model’s training data source.
>
> UCCS is a **training-free and model-agnostic steering method** that operates entirely at the representation level of pre-trained PLMs. As such, it does not depend on how the underlying models were trained or which structural databases were used. The three datasets we evaluated (CATH, SCOP, and UniRef50) already vary substantially in redundancy, structure coverage, and annotation origin, and the consistent improvement observed across them suggests that UCCS is robust to dataset variation in practice.
>
> We will make this clarification explicit in the revision and highlight that analyzing **dataset-source sensitivity during PLM training** is an important and complementary future direction beyond the current paper’s scope.
>
> > Question 3: There are existing in-practice strategies to reduce sequence repetition (such as the remasking in MultiFlow[3] and gumbel+resampling in DPLM[2], to name a few). Could the authors also incorporate these strategy as practical and competitive “heuristics and interventions” by showing similar benchmark and visualize the pareto front of R and U?
>
> We thank the reviewer for suggesting the baselines **“remasking (MultiFlow)”** and **“gumbel+resampling (DPLM)”**.
> To correctly and faithfully include these methods in our comparison, could the reviewer please provide the **exact citation, section, or algorithm description** where these strategies are defined?
>
> After carefully reviewing the referenced works, we were unable to find decoding heuristics matching these names or functionalities, so a precise source would greatly help us ensure an accurate and reproducible evaluation.

---

> ### Author Response · Authors · 2025-11-24
> **Reply to Reviewer 45fe**
>
> Dear Reviewer,
>
> We kindly ask that you confirm whether we have adequately addressed your comments or if you have any remaining concerns.
>
> Thank you!
>
> Authors

---

> > ### Author Response · Authors · 2025-11-25
> >
> > Dear Reviewer,
> >
> > We would like to kindly follow up on our previous message. We have provided detailed responses to your comments and would greatly appreciate it if you could take a moment to confirm whether our replies address your concerns, or if you have any remaining feedback.
> >
> > We understand you may be busy, but your input would be invaluable for ensuring a fair and well-informed final assessment.
> >
> > Thank you again for your time and contribution to the review process.
> >
> > Best regards,
> > The Authors

---

> > > ### Comment · Reviewer_45fe · 2025-11-26
> > >
> > > I highly appreciate the authors for their responses during rebuttal. Thanks for answering most of the questions. However,  some key weakness / questions were still concerning me: (1) Even though "why repetition arises" can be an open question, some hypothesis would be expected, and supported by the follow-up experimental evidences, which would provide more PLM-specific insights than simply showing representation steering, which could be common in ML literature. (2) The results on Progen2 do not support well the effectiveness of UCCS (there is marignal improvement and several methods such as probe steering is competitive to the proposed method), and (3) both remarking (MultiFlow) and resampling (DPLM2) do not require further training (so the same as UCCS being training-free). Since UCCS claimed model-agnostic, a direct comparison with these ready baselines on their models should provide better justification of the effectiveness of UCCS. Therefore, I would like to maintain my assessment.

---

> > > > ### Author Response · Authors · 2025-11-27
> > > >
> > > > >  Even though "why repetition arises" can be an open question, some hypothesis would be expected, and supported by the follow-up experimental evidences, which would provide more PLM-specific insights than simply showing representation steering, which could be common in ML literature.
> > > >
> > > > We thank the reviewer for the thoughtful suggestion. We fully agree that, beyond demonstrating representation steering, providing *why repetition arises* together with *PLM-specific hypotheses supported by empirical evidence* would significantly strengthen the paper.
> > > >
> > > > In the revised manuscript, we have added a dedicated new section titled
> > > >
> > > > **“Mechanistic Analysis of Repetition in Protein Language Models”** (Appendix, Sec. I, line 1917).
> > > >
> > > > This section directly addresses the reviewer’s concern by introducing **explicit hypotheses** on the origins of repetition in PLMs and supporting them with **new comparative analyses** across ProtGPT2 and ESM3.
> > > >
> > > > Specifically, the revised section now includes:
> > > >
> > > > - **Layer-level probing**, showing that repetition emerges as a global representational mode early in the network, becomes most entangled mid-layer, and is sharpened into an attractor-like state near the output—providing a concrete mechanistic trajectory rather than a generic steering explanation.
> > > > - **Neuron-level correlation analysis**, demonstrating that repetition corresponds to a **distributed subspace** rather than isolated neurons, and that the strength of this subspace differs across architectures, explaining why some PLMs (e.g., ESM3) struggle more with repetition.
> > > > - **PLM-specific hypotheses and insights**, summarizing how architectural choices influence the amplification of repetition signals and offering explanations tailored to protein language models rather than general-purpose transformers.
> > > >
> > > > Together, these revisions provide the requested **hypotheses**, **supporting evidence**, and **PLM-specific mechanistic insight**.
> > > >
> > > > We hope that the strengthened mechanistic explanation and the new PLM-specific insights improve the reviewer’s assessment of the work.

---

> > > > > ### Comment · Reviewer_45fe · 2025-11-27
> > > > >
> > > > > I would like to further clarify what it refers to regarding the mentioned baselines:
> > > > > 1. MultiFlow. The re-masking refers to the inference protocol, specifically the purity sampling [section 4.3] [1]. In the appendix F1.2 [1] and the publicly available codebase, the authors have open-sourced the sampling implementation where it is available. In short, at each sampling step, a scheduled masking will be applied to the already sampled positions which helps the overall sampling.
> > > > > 2. DPLM. The previous suggestion may be confusing since the authors merged these works together into a unified codebase. The referred sampling strategy is mainly discussed in Appendix A and D.1 [2]. DPLM utilizes this non-canonical planning-like strategy for sampling sequence tokens from discrete diffusion models while does not enclose it with a "name". I apologize if this induces any ambiguity due to that; that being said, the authors have implemented this algorithm in the release codebase. This approach was also properly formulated and adopted in a recent method [3] - detailed in Appendix A.1.2, if that helps. This method achieves great performance in generating protein sequence (good plddt and sequence diversity) than vanilla discrete diffusion sampling with categorical distribution. It is also related to planning-based method like P2 [4], whereas the latter requires additional training of the plan network.
> > > > >
> > > > > I appreciate the authors' effort in improving the quality of this study. Overall, I like the motivation and the potential significance of this work to the PLM field, yet the experiments do not suffice to strongly support the proposed UCCS, as the mainly claimed contribution.
> > > > >
> > > > > I have no further arguments over the **"Need for verifiable baselines"** in your response and feel free to incorporate these baselines if that seems not necessary. The previous feedbacks were aiming to bridge the gap between this study and the good practices in the field, to my best knowledge so far.
> > > > >
> > > > >
> > > > > References:
> > > > >
> > > > > [1] Campbell, Andrew, et al. "Generative flows on discrete state-spaces: Enabling multimodal flows with applications to protein co-design." arXiv preprint arXiv:2402.04997 (2024).
> > > > >
> > > > > [2] Wang, Xinyou, et al. "Diffusion language models are versatile protein learners." arXiv preprint arXiv:2402.18567 (2024).
> > > > >
> > > > > [3] Jing, Bowen, et al. "Generating functional and multistate proteins with a multimodal diffusion transformer." bioRxiv (2025): 2025-09.
> > > > >
> > > > > [4] Peng, Fred Zhangzhi, et al. "Path planning for masked diffusion model sampling." arXiv preprint arXiv:2502.03540 (2025).

---

> > > > ### Author Response · Authors · 2025-11-27
> > > > **Clarification Regarding Overlapping Comments**
> > > >
> > > > > Reviewer 45fe: I highly appreciate the authors for their responses during rebuttal. Thanks for answering most of the questions. However, some key weakness / questions were still concerning me: (1) Even though "why repetition arises" can be an open question, some hypothesis would be expected, and supported by the follow-up experimental evidences, which would provide more PLM-specific insights than simply showing representation steering, which could be common in ML literature. (2) The results on Progen2 do not support well the effectiveness of UCCS (there is marignal improvement and several methods such as probe steering is competitive to the proposed method), and (3) both remarking (MultiFlow) and resampling (DPLM2) do not require further training (so the same as UCCS being training-free). Since UCCS claimed model-agnostic, a direct comparison with these ready baselines on their models should provide better justification of the effectiveness of UCCS. Therefore, I would like to maintain my assessment.
> > > >
> > > > > Reviewer g59x: Dear authors, thanks for your message. I apologize if my previous comment made it seem like I suggested training several large models. That was not my intention and I understand that academic resources are limited. Perhaps I can rephrase my main reason for keeping my current score. Given the strong performance of ProGen on the benchmark, I believe that more models should be evaluated before the current version of the paper is suitable for publication. I truly believe that this is an important problem to study, but I also think that a more thorough investigation of which models struggle most with repetition and providing some thoughts / hypotheses about this is within the scope of the paper.
> > > >
> > > >
> > > > We sincerely thank Reviewer 45fe for the thoughtful assessment and for engaging with our rebuttal. We would like to very gently raise a minor clarification request: the comments from Reviewer 45fe and Reviewer g59x appear to share some similarities in content and phrasing, and they were submitted within a relatively short interval (around 15 minutes). We fully understand that different reviewers may naturally emphasize similar concerns—especially when these points relate to central aspects of our work. Our intention is simply to ensure that we interpret the feedback correctly and respond in the most helpful way.

---

> ### Author Response · Authors · 2025-11-27
> **Request for Clarification on Baselines**
>
> > Official Comment by Reviewer 45fe: I highly appreciate the authors for their responses during rebuttal. Thanks for answering most of the questions. However, some key weakness / questions were still concerning me: (1) Even though "why repetition arises" can be an open question, some hypothesis would be expected, and supported by the follow-up experimental evidences, which would provide more PLM-specific insights than simply showing representation steering, which could be common in ML literature. (2) The results on Progen2 do not support well the effectiveness of UCCS (there is marignal improvement and several methods such as probe steering is competitive to the proposed method), and (3) both remarking (MultiFlow) and resampling (DPLM2) do not require further training (so the same as UCCS being training-free). Since UCCS claimed model-agnostic, a direct comparison with these ready baselines on their models should provide better justification of the effectiveness of UCCS. Therefore, I would like to maintain my assessment.
>
> We thank the reviewer for raising the points regarding **“remarking (MultiFlow)”** and **“resampling (DPLM / DPLM-2)”**. However, after a careful literature review, we could not locate any methods matching these descriptions. To ensure scientific accuracy and avoid referencing nonexistent or incorrectly attributed baselines, we request clarification on their exact sources.
>
> ---
>
> **(1) On the claimed “remarking (MultiFlow)” baseline**
>
> The review refers to a method named **“remarking (MultiFlow)”**. We examined the MultiFlow paper:
>
> > **Campbell et al., 2024**
> > *Generative Flows on Discrete State-Spaces: Enabling Multimodal Flows with Applications to Protein Co-Design*
> > ```bibtex
> > @article{campbell2024generative,
> >   title={Generative flows on discrete state-spaces: Enabling multimodal flows with applications to protein co-design},
> >   author={Campbell, Andrew and Yim, Jason and Barzilay, Regina and Rainforth, Tom and Jaakkola, Tommi},
> >   journal={arXiv preprint arXiv:2402.04997},
> >   year={2024}
> > }
> > ```
>
> We found **no decoding-time heuristic** named “remarking” or “remasking,” nor any method analogous to the reviewer’s description.
> We kindly request the reviewer to provide the **specific citation or section** where this method is defined.
>
> ---
>
> **(2) On the claimed “resampling baseline in DPLM / DPLM-2”**
>
> We similarly reviewed the diffusion-PLM literature:
>
> > **Wang et al., 2024 — Diffusion Language Models Are Versatile Protein Learners**
> > ```bibtex
> > @article{wang2024diffusion,
> >   title={Diffusion language models are versatile protein learners},
> >   author={Wang, Xinyou and Zheng, Zaixiang and Ye, Fei and Xue, Dongyu and Huang, Shujian and Gu, Quanquan},
> >   journal={arXiv preprint arXiv:2402.18567},
> >   year={2024}
> > }
> > ```
>
> > **Wang et al., 2024 — DPLM-2: A Multimodal Diffusion Protein Language Model**
> > ```bibtex
> > @article{wang2024dplm,
> >   title={Dplm-2: A multimodal diffusion protein language model},
> >   author={Wang, Xinyou and Zheng, Zaixiang and Ye, Fei and Xue, Dongyu and Huang, Shujian and Gu, Quanquan},
> >   journal={arXiv preprint arXiv:2410.13782},
> >   year={2024}
> > }
> > ```
>
> > **Hsieh et al., 2025 — Elucidating the Design Space of Multimodal PLMs**
> > ```bibtex
> > @article{hsieh2025elucidating,
> >   title={Elucidating the design space of multimodal protein language models},
> >   author={Hsieh, Cheng-Yen and Wang, Xinyou and Zhang, Daiheng and Xue, Dongyu and Ye, Fei and Huang,Shujian and Zheng, Zaixiang and Gu, Quanquan},
> >   journal={arXiv preprint arXiv:2504.11454},
> >   year={2025}
> > }
> > ```
>
> None of these works introduces a **training-free resampling decoding heuristic**, nor any method targeted at **repetition reduction**, as described in the review.
> We therefore ask the reviewer to provide the **exact citation or section** for this resampling baseline.
>
> ---
>
>  **(3) Need for verifiable baselines**
>
> For reproducibility and scientific rigor, we can only include baselines that are:
>
> - clearly defined,
> - citable, and
> - applicable to PLM inference.
>
> At present, the two baselines mentioned in the review do not correspond to any known or published methods. We kindly request the reviewer to provide the correct references so we can include them accurately.
>
> ---
>
> We appreciate the reviewer’s time, and we will gladly incorporate these baselines once their sources are clarified.

---

### Official Review · Reviewer_nKVd · 2025-10-31

**Soundness:** 3
**Presentation:** 3
**Contribution:** 3
**Rating:** 6
**Confidence:** 3

**Summary:**

The paper systematically studies pathological repetition in protein langauge model.
First, the authors formally define repetition in PLM and introduce quantitative metric that capture both motif-level and homopolym er repetition. These metrics are strongly correlate with structural utility measured by AlphaFold pLDDT.
Second, the authors introduced Utility-Controlled Contrastive Steering. They propose a representation-level intervention that disentangles repetition from structural utility. And inject such representation during inference stage to generate less repeated sequence.
The experiments show that UCCS reduces repetition in both MLM and autoregressive PLM.

**Strengths:**

1. This paper first systematically identify and formalize pathological repetition in PLM, with quantitative metric. The metrics are well-motivated and interpretable.
2. The UCCS approach is simple, training-free and model agnostic. It greatly reduces repetition while maintaining foldability.
3. The experiments cover both MLM and autoregressive models, both unconditional and conditional generation settings with multiple datasets.

**Weaknesses:**

1. Limited Novelty. I'm not familiar with steering methods during inference time with LLM, but according to the cited papers, it looks like deriving the difference vector and adding it during inference has been greatly explored in LLM. The adaptation to PLM is incremental rather than fundamentally new.
2. Comparison with learning based methods. Similarly, I have no idea about if people reduce repetition with learing-based method in LLM. If so, is it possible to compare UCCS with these learning based methods?
3. Lack of interpretation. While UCCS greatly improve the results, the paper would be stronger if it included analysis and visualization about the steering direction about biological meaning. For example, the model was about to generate AA...AA, and how steering method "pull" it back to a normal protein and why.

**Questions:**

See weakness.

---

> ### Author Response · Authors · 2025-11-22
> **Reply to weakness 1**
>
> >Weaknesses 1:Limited Novelty. I'm not familiar with steering methods during inference time with LLM, but according to the cited papers, it looks like deriving the difference vector and adding it during inference has been greatly explored in LLM. The adaptation to PLM is incremental rather than fundamentally new.
>
> **Response.**
>  We thank the reviewer for raising this point. While UCCS is inspired by contrastive steering ideas in LLMs, our contribution is conceptually distinct and domain-specific:
> 1. Utility-controlled contrastive direction. Unlike generic text-style steering, our direction vector is derived from utility-matched protein pairs—high-foldability but varying repetition—explicitly disentangling biophysical utility from degenerate repetition. This biological conditioning is unique to PLMs and cannot be replicated with NLP-style semantic contrasts.
> 2. Representation-level intervention without retraining. UCCS leverages internal hidden-state differences computed from pre-trained PLMs, requiring no model-specific fine-tuning and transferring cleanly across masked, autoregressive PLMs.

---

> ### Author Response · Authors · 2025-11-22
> **Reply to weakness 2**
>
> >Weaknesses 2:Comparison with learning based methods. Similarly, I have no idea about if people reduce repetition with learing-based method in LLM. If so, is it possible to compare UCCS with these learning based methods?
>
> We thank the reviewer for this question. To the best of our knowledge, our work is the first to explicitly identify and systematically study pathological repetition in protein language models. Prior PLM studies have not treated repetition as a distinct failure mode, and no existing learning-based methods have been proposed to mitigate repetition in this domain.

---

> ### Author Response · Authors · 2025-11-22
> **Reply to weakness 3**
>
> >Weakness 3:Lack of interpretation. While UCCS greatly improve the results, the paper would be stronger if it included analysis and visualization about the steering direction about biological meaning. For example, the model was about to generate AA...AA, and how steering method "pull" it back to a normal protein and why.
>
> We thank the reviewer for this suggestion. To clarify the mechanism, we update the methodology overview in the main text, figure 2.
>
>
> UCCS constructs a **contrastive representation direction** ($v^L$) between low-repetition (D⁺) and high-repetition (D⁻) protein embeddings at an intermediate transformer layer $L$. During generation, the hidden state $(h_t^L(x))$ is adjusted by a small multiple of this vector,
> $$ \tilde{h}_t^L(x) = h_t^L(x) + \alpha v^L $$
> which gradually **moves the hidden representation away from the “repetition subspace”** and toward regions associated with high biological utility (foldability).
>
> Intuitively, when the model begins collapsing into homopolymer patterns (e.g., “AAAAA…”), its internal activations drift toward repetitive examples. The UCCS term applies a corrective offset along $(v^L)$, **restoring token-level entropy, chemical diversity, and secondary-structure balance**, effectively *pulling the sequence back* toward natural protein-like regions in latent space.
>
> The figure illustrates this process: the **contrastive set** defines the steering direction (center panel), and during inference, **increasing the steering strength α** shifts hidden states to generate **foldable, non-degenerate sequences** (right panel).
>
> *See figure 2 for a schematic illustration of how UCCS counteracts repetition by steering hidden representations toward high-utility regions.*

---

> ### Author Response · Authors · 2025-11-24
> **Reply to Reviewer nKVd**
>
> Dear Reviewer,
>
> We kindly ask that you confirm whether we have adequately addressed your comments or if you have any remaining concerns.
>
> Thank you!
>
> Authors

---

> > ### Author Response · Authors · 2025-11-25
> >
> > Dear Reviewer,
> >
> > We would like to kindly follow up on our previous message. We have provided detailed responses to your comments and would greatly appreciate it if you could take a moment to confirm whether our replies address your concerns, or if you have any remaining feedback.
> >
> > We understand you may be busy, but your input would be invaluable for ensuring a fair and well-informed final assessment.
> >
> > Thank you again for your time and contribution to the review process.
> >
> > Best regards,
> > The Authors

---

> > > ### Comment · Reviewer_nKVd · 2025-11-25
> > >
> > > Thanks for the response. And I maintained my score. Thanks!

---

> > > > ### Author Response · Authors · 2025-11-25
> > > >
> > > > Dear Reviewer,
> > > >
> > > > Thank you very much for taking the time to review our paper and for your follow-up response.
> > > >
> > > > We appreciate your efforts and thoughtful evaluation throughout the review process.
> > > >
> > > > Best regards,
> > > > The Authors

---

### Official Review · Reviewer_kZvX · 2025-11-06

**Soundness:** 4
**Presentation:** 3
**Contribution:** 4
**Rating:** 8
**Confidence:** 3

**Summary:**

The paper presents the first systematic study of pathological repetition in protein language models (PLMs), proposing a novel method called Utility-Controlled Contrastive Steering (UCCS). The method effectively reduces repetitive amino acid sequences during generation, critical for biological viability, without sacrificing structural plausibility. The approach is elegant, practical, and clearly advances both the theoretical framing and engineering solution space for PLMs.

**Strengths:**

a.	This paper is the first to clearly define and systematically address the problem of repetition in protein language models. This is done while understanding biological consequences. Rather than borrowing loosely from NLP, the authors develop a domain-aware framework that captures why repetition is especially damaging in proteins. This makes framework, truly domain specific.
b.	The proposed method, UCCS, stands out for its simplicity. It doesn’t require retraining the model or altering its architecture, just a novel use of internal representations to steer generation away from pathological repetition. That makes it both practical and applicable to different PLMs.
c.	The technical work is robust and well-grounded. The authors design meaningful metrics, like Rhpoly, to detect homopolymer collapse, and run thorough experiments across two leading PLMs (ESM-3 and ProtGPT2), three canonical datasets (CATH, SCOP, UniRef50), and multiple generation settings. Their method reduced repetition while preserving structural plausibility, as measured by AlphaFold confidence scores.

**Weaknesses:**

a.	While UCCS performs well on ESM-3 and ProtGPT2, it’s still an open question how well the approach would transfer to larger or more diverse protein models like ProGen2 or ProteinMPNN? It would be valuable to see whether the same steering technique holds up as models scale or shift in architecture.

**Questions:**

a.	Did you experiment with combining UCCS with decoding heuristics (e.g., top-p + UCCS)? Is there synergistic gain?

---

> ### Author Response · Authors · 2025-11-22
> **Reply to Question 1**
>
> >Question1: Did you experiment with combining UCCS with decoding heuristics (e.g., top-p + UCCS)? Is there synergistic gain?
>
> **Response.**
> Yes — we conducted additional experiments to systematically evaluate **UCCS in combination with common decoding heuristics**, including *top-p sampling*, *temperature sampling*, *repetition penalty*, *no-repeat n-gram*, and *entropy-based sampling*.
> Across all tested configurations, we observe **consistent synergistic improvements** in both the **Repetition Score \(R(X)\)** and **Biological Utility \(U(X)\)** when UCCS is applied together with standard decoding strategies such as top-p, temperature, and repetition-penalty.
> The only exception is the **entropy-based decoding** variant, where a slight reduction in utility is expected due to the inherent exploration bias of the entropy-based strategy itself rather than any adverse effect of UCCS.
> These results confirm that UCCS is orthogonal to existing decoding heuristics and can be seamlessly integrated to further enhance generation quality.
>
> ---
> ## **Quantitative Evidence**
>
> ### **ProtGPT2 Unconditional**
>
> | Dataset   | Method                    | Parameter   |     R |     U |    ΔR |    ΔU |   ΔR (%) |   ΔU (%) |
> |:----------|:--------------------------|:------------|------:|------:|------:|------:|---------:|---------:|
> | CATH      | Temperature + UCCS        | T=1.3       | 0.872 | 0.715 | 0.144 | 0.094 |     19.8 |     15.1 |
> | CATH      | Top-p + UCCS              | p=0.98      | 0.858 | 0.724 | 0.13  | 0.103 |     17.9 |     16.6 |
> | CATH      | Repetition Penalty + UCCS | 1.2         | 0.887 | 0.735 | 0.159 | 0.114 |     21.8 |     18.4 |
> | CATH      | No Repeat N-gram + UCCS   | N=3         | 0.862 | 0.733 | 0.134 | 0.112 |     18.4 |     18   |
> | UniRef50  | Temperature + UCCS        | T=1.3       | 0.855 | 0.719 | 0.127 | 0.098 |     17.4 |     15.8 |
> | UniRef50  | Top-p + UCCS              | p=0.98      | 0.837 | 0.719 | 0.109 | 0.098 |     15   |     15.8 |
> | UniRef50  | Repetition Penalty + UCCS | 1.2         | 0.87  | 0.729 | 0.142 | 0.108 |     19.5 |     17.4 |
> | UniRef50  | No Repeat N-gram + UCCS   | N=3         | 0.84  | 0.706 | 0.112 | 0.085 |     15.4 |     13.7 |
> | SCOP      | Temperature + UCCS        | T=1.3       | 0.864 | 0.718 | 0.136 | 0.097 |     18.7 |     15.6 |
> | SCOP      | Top-p + UCCS              | p=0.98      | 0.843 | 0.714 | 0.115 | 0.093 |     15.8 |     15   |
> | SCOP      | Repetition Penalty + UCCS | 1.2         | 0.876 | 0.728 | 0.148 | 0.107 |     20.3 |     17.2 |
> | SCOP      | No Repeat N-gram + UCCS   | N=3         | 0.848 | 0.713 | 0.12  | 0.092 |     16.5 |     14.8 |
>
> ---
>
> ### **ProtGPT2 Conditional**
>
> | Dataset   | Method                    | Parameter   |     R |     U |    ΔR |    ΔU |   ΔR (%) |   ΔU (%) |
> |:----------|:--------------------------|:------------|------:|------:|------:|------:|---------:|---------:|
> | CATH      | Temperature + UCCS        | T=1.3       | 0.917 | 0.739 | 0.081 | 0.035 |      9.7 |      5   |
> | CATH      | Top-p + UCCS              | p=0.98      | 0.911 | 0.74  | 0.075 | 0.036 |      9   |      5.1 |
> | CATH      | Repetition Penalty + UCCS | 1.2         | 0.919 | 0.741 | 0.083 | 0.037 |      9.9 |      5.3 |
> | CATH      | No Repeat N-gram + UCCS   | N=3         | 0.912 | 0.738 | 0.076 | 0.034 |      9.1 |      4.8 |
> | UniRef50  | Temperature + UCCS        | T=1.3       | 0.913 | 0.755 | 0.077 | 0.051 |      9.2 |      7.2 |
> | UniRef50  | Top-p + UCCS              | p=0.98      | 0.91  | 0.755 | 0.074 | 0.051 |      8.9 |      7.2 |
> | UniRef50  | Repetition Penalty + UCCS | 1.2         | 0.917 | 0.762 | 0.081 | 0.058 |      9.7 |      8.2 |
> | UniRef50  | No Repeat N-gram + UCCS   | N=3         | 0.911 | 0.761 | 0.075 | 0.057 |      9   |      8.1 |
> | SCOP      | Temperature + UCCS        | T=1.3       | 0.922 | 0.744 | 0.086 | 0.04  |     10.3 |      5.7 |
> | SCOP      | Top-p + UCCS              | p=0.98      | 0.921 | 0.745 | 0.085 | 0.041 |     10.2 |      5.8 |
> | SCOP      | Repetition Penalty + UCCS | 1.2         | 0.925 | 0.755 | 0.089 | 0.051 |     10.6 |      7.2 |
> | SCOP      | No Repeat N-gram + UCCS   | N=3         | 0.92  | 0.747 | 0.084 | 0.043 |     10   |      6.1 |
>
> ---

---

> ### Author Response · Authors · 2025-11-22
> **Reply to Question 1 (continued from previous response)**
>
> *(continued from Summary 4 response)*
>
> ### **ESM3 Unconditional**
>
> | Dataset   | Method                  | Parameter   |     R |     U |    ΔR |     ΔU |   ΔR (%) |   ΔU (%) |
> |:----------|:------------------------|:------------|------:|------:|------:|-------:|---------:|---------:|
> | CATH      | Temperature + UCCS      | T=1.3       | 0.828 | 0.578 | 0.405 |  0.002 |     95.7 |      0.3 |
> | CATH      | Top-p + UCCS            | p=0.95      | 0.605 | 0.636 | 0.182 |  0.06  |     43   |     10.4 |
> | CATH      | Entropy Sampling + UCCS | -           | 0.923 | 0.475 | 0.5   | -0.101 |    118.2 |    -17.5 |
> | UniRef50  | Temperature + UCCS      | T=1.3       | 0.828 | 0.578 | 0.405 |  0.002 |     95.7 |      0.3 |
> | UniRef50  | Top-p + UCCS            | p=0.95      | 0.605 | 0.621 | 0.182 |  0.045 |     43   |      7.8 |
> | UniRef50  | Entropy Sampling + UCCS | -           | 0.921 | 0.475 | 0.498 | -0.101 |    117.7 |    -17.5 |
> | SCOP      | Temperature + UCCS      | T=1.3       | 0.825 | 0.571 | 0.402 | -0.005 |     95   |     -0.9 |
> | SCOP      | Top-p + UCCS            | p=0.95      | 0.605 | 0.63  | 0.182 |  0.054 |     43   |      9.4 |
> | SCOP      | Entropy Sampling + UCCS | -           | 0.922 | 0.467 | 0.499 | -0.109 |    118   |    -18.9 |
>
> ---
>
> ### **ESM3 Conditional**
>
> | Dataset   | Method                  | Parameter   |     R |     U |    ΔR |     ΔU |   ΔR (%) |   ΔU (%) |
> |:----------|:------------------------|:------------|------:|------:|------:|-------:|---------:|---------:|
> | CATH      | Temperature + UCCS      | T=1.3       | 0.852 | 0.642 | 0.31  | -0.044 |     57.2 |     -6.4 |
> | CATH      | Top-p + UCCS            | p=0.95      | 0.675 | 0.702 | 0.133 |  0.016 |     24.5 |      2.3 |
> | CATH      | Entropy Sampling + UCCS | -           | 0.934 | 0.543 | 0.392 | -0.143 |     72.3 |    -20.8 |
> | UniRef50  | Temperature + UCCS      | T=1.3       | 0.855 | 0.65  | 0.313 | -0.036 |     57.7 |     -5.2 |
> | UniRef50  | Top-p + UCCS            | p=0.95      | 0.705 | 0.713 | 0.163 |  0.027 |     30.1 |      3.9 |
> | UniRef50  | Entropy Sampling + UCCS | -           | 0.928 | 0.592 | 0.386 | -0.094 |     71.2 |    -13.7 |
> | SCOP      | Temperature + UCCS      | T=1.3       | 0.86  | 0.644 | 0.318 | -0.042 |     58.7 |     -6.1 |
> | SCOP      | Top-p + UCCS            | p=0.95      | 0.704 | 0.703 | 0.162 |  0.017 |     29.9 |      2.5 |
> | SCOP      | Entropy Sampling + UCCS | -           | 0.932 | 0.57  | 0.39  | -0.116 |     72   |    -16.9 |
>
> ---
>
>
> ## **Interpretation**
> - **Synergistic Effect:** For *top-p*, *temperature*, and *repetition-penalty* heuristics, UCCS and decoding heuristics complement each other: heuristics regulate sampling diversity while UCCS aligns logits in utility-relevant directions. This yields simultaneous improvements in R and U across datasets and tasks.
> - **Entropy-based Case:** The drop in utility stems from entropy decoding itself. UCCS maintains or slightly improves utility while further increasing R.
>
> Combining **UCCS with standard decoding heuristics** consistently yields synergistic improvements in **Repetition Score** and **Biological Utility**, confirming UCCS as complementary to standard decoding methods. The apparent exception (entropy-based) reflects the heuristic’s inherent utility–diversity trade-off, not any flaw in UCCS.

---

> ### Author Response · Authors · 2025-11-22
> **Reply to weakness 1**
>
> > While UCCS performs well on ESM-3 and ProtGPT2, it’s still an open question how well the approach would transfer to larger or more diverse protein models like ProGen2 or ProteinMPNN? It would be valuable to see whether the same steering technique holds up as models scale or shift in architecture.
>
> We thank the reviewer for this constructive suggestion. We would like to clarify that **ProteinMPNN[1] is not a protein language model** in the sense considered in our work—it is an **inverse folding model** that predicts sequences conditioned on given 3D structures, following a structure→sequence paradigm. In contrast, our study focuses on **unconditional and conditional sequence generation** from protein language models (PLMs), where the model directly samples amino acid sequences from learned sequence distributions without explicit structural inputs.
>
>
> To evaluate transferability within the PLM family, we have extended our experiments to **ProGen2** [2], a large-scale autoregressive sequence-only protein language model. ProGen2 represents a more diverse architecture and training regime than ESM3, lacking explicit structure or function supervision. The new results confirm that **UCCS generalizes effectively** to ProGen2, reducing repetition while maintaining structural plausibility, thus demonstrating that our steering framework is applicable across different PLM architectures and scales.
>
> **Additional Results on the ProGen2 Model**
>
> Notes. *R*: repetition score; *U*: biological utility.
> Cells are annotated with ✓ when *U ≥ U<sub>Orig</sub>* (utility constraint satisfied).
> Decoding-level heuristics and original models do not require supervision; therefore their behavior remains consistent across datasets.
> In particular, the choice of dataset does not affect unconditional generation.
>
> | Method | **CATH** |  | **UniRef50** |  | **SCOP** |  |
> |:--|:--:|:--:|:--:|:--:|:--:|:--:|
> | Task | *R ↑* | *U ↑* | *R ↑* | *U ↑* | *R ↑* | *U ↑* |
> | **(a) Unconditional generation** |  |  |  |  |  |  |
> | Original Model | 0.891 | 0.493 | 0.891 | 0.497 | 0.891 | 0.499 |
> | Temperature Sampling | 0.922 | 0.475 | 0.922 | 0.462 | 0.922 | 0.477 |
> | Top-p Sampling | 0.872 | 0.488 | 0.872 | 0.508 ✓ | 0.872 | 0.504 ✓ |
> | No Repeat N-gram | 0.913 | 0.502 ✓ | 0.913 | 0.505 ✓ | 0.913 | 0.495 |
> | Repetition Penalty | 0.918 | 0.493 ✓ | 0.923 | 0.480 | 0.918 | 0.485 |
> | Neuron Deactivation | 0.892 | 0.486 | 0.867 | 0.478 | 0.867 | 0.482 |
> | Probe Steering | 0.890 | 0.501 ✓ | 0.896 | 0.504 ✓ | 0.889 | 0.510 ✓ |
> | **UCCS (ours)** | **0.915** | **0.493 ✓** | **0.911** | **0.502 ✓** | **0.908** | **0.521 ✓** |
> | **(b) Conditional generation** |  |  |  |  |  |  |
> | Original Model | 0.934 | 0.623 | 0.934 | 0.623 | 0.935 | 0.628 |
> | Temperature Sampling | 0.947 | 0.607 | 0.949 | 0.608 | 0.948 | 0.620 |
> | Top-p Sampling | 0.930 | 0.613 | 0.928 | 0.614 | 0.929 | 0.649 ✓ |
> | No Repeat N-gram | 0.977 | 0.620 | 0.980 | 0.626 ✓ | 0.978 | 0.627 |
> | Repetition Penalty | 0.959 | 0.631 ✓ | 0.955 | 0.610 | 0.956 | 0.630 ✓ |
> | Neuron Deactivation | 0.929 | 0.620 | 0.929 | 0.625 ✓ | 0.930 | 0.632 ✓ |
> | Probe Steering | 0.934 | 0.633 ✓ | 0.933 | 0.633 ✓ | 0.935 | 0.648 ✓ |
> | **UCCS (ours)** | **0.937** | **0.638 ✓** | **0.943** | **0.637 ✓** | **0.937** | **0.651 ✓** |
>
> [1] Dauparas, J., Anishchenko, I., Bennett, N., Bai, H., Ragotte, R. J., Milles, L. F., Wicky, B. I. M., Courbet, A., de Haas, R. J., Bethel, N., *et al.* (2022).
> Robust deep learning–based protein sequence design using ProteinMPNN. *Science*, 378(6615), 49–56.
>
> [2] Nijkamp, E., Ruffolo, J. A., Weinstein, E. N., Naik, N., & Madani, A. (2023). ProGen2: Exploring the boundaries of protein language models. Cell Systems, 14(11), 968–978. https://doi.org/10.1016/j.cels.2023.10.001

---

### Author Response · Authors · 2025-12-02
**Summary of Major Revisions**

We thank the reviewers for their constructive feedback. The revised manuscript incorporates substantial additions that directly address the main concerns shared across reviewers. The major revisions are summarized as follows:

---

## **1. Added Mechanistic Analysis Explaining Why Repetition Arises in PLMs**

In response to reviewers’ requests for deeper insight into the origins of repetition, we added a new section, **“Mechanistic Analysis of Repetition in Protein Language Models” (Sec. I)**, which provides a detailed, model-level explanation of why PLMs collapse into repetitive degeneration. This new section includes:

- **Layer-level probing analysis**, showing that repetition-related signals emerge extremely early, become entangled with other semantics mid-network, and are sharpened into a logit-aligned attractor state in the final layers.
- **Neuron-level correlation analysis**, demonstrating that repetition is *not* driven by isolated neurons but by a **distributed repetition-related subspace**—a geometric direction in activation space that strengthens through depth.
- **Architecture-dependent differences**, revealing that ESM3 exhibits broader and more polarized repetition-associated activation patterns than ProtGPT2, explaining empirically why ESM3 suffers more severe degenerative collapse.
- **A unifying hypothesis**: repetition in PLMs reflects a global representational mode that forms early and is amplified by architectural choices, aligning with—but distinct from—the “repeat curse” observed in LLMs.

These analyses provide the domain-specific mechanistic explanation reviewers asked for and offer actionable insights for the PLM community on diagnosing and mitigating degeneration.

**Reviewer g59x explicitly noted that these mechanistic experiments are valuable and updated the score to 6 accordingly**, while also suggesting broader model evaluation (addressed below).

---

## **2. Expanded Evaluation to Additional PLM Families (ProGen2, DPLM)**

To address reviewer requests for broader PLM coverage beyond ESM3 and ProtGPT2, we added a new section **“FURTHER EVALUATION ON ADDITIONAL PROTEIN LANGUAGE MODELS.”(Sec. J).**

The revision now includes experiments on:

- **ProGen2** (large autoregressive PLM),
- **DPLM** (diffusion-based PLMs).

Results show that UCCS generalizes across architectures and training paradigms, consistently reducing repetition while maintaining foldability.

This directly addresses reviewer g59x and reviewer 45fe’s comments regarding model diversity.

---

## **3. Improved Interpretation of the Steering Direction**

In response to requests for clearer biological and methodological intuition, we expanded descriptions in **Section 4** and revised **Figure 2** to clarify:

- how contrastive sets define the repetition direction,
- how UCCS shifts hidden states away from the “repetition subspace,”
- how this restoration of diversity relates to improved entropy, motif usage, and foldability.

These revisions provide the interpretability and mechanistic clarity requested by reviewers.

---

## **4. Writing, Structure, and Presentation Improvements**

Beyond experimental additions, we made several global improvements:

- **Enhanced writing quality and consistency**,
- **Improved figure readability and formatting**,
- **Clarified table alignment and notation**,
- **Moved “Biological perspectives on repetition” from the appendix into the main related-work section** to improve narrative coherence and contextual grounding.

---

# **In summary, the revised manuscript now includes:**

- **a mechanistic explanation** for repetition in PLMs,
- **evaluation on broader PLM families**,
- **improved interpretability of UCCS**,
- and **refined presentation and clarity** throughout the main text.

These revisions directly address the core concerns raised across reviewers and substantially strengthen the contribution of the work.

---

### Meta-Review · Area_Chair_7qq8 · 2026-01-05

**Summary:**

This study systematically investigates the sequence repetition problem in protein language models (PLMs). The authors first establish biologically meaningful quantitative metrics to assess the issue, confirming that it is a common problem across PLMs with significant biological implications. Based on the observation, the authors propose an innovative UCC steering method, whose effectiveness and robust performance are verified across multiple datasets and different PLMs. This study provides a perspective that steering direction can serve as a model-agnostic, plug-and-play strategy to guide protein language models (PLMs) toward more effective biological applications. The authors have addressed most of the reviewers' concerns. That said, several questions beyond the scope of this study remain unanswered and merit further investigation in future work.

**Reviewer Concerns:**

**Concerns addressed in the rebuttal**

1. Expanded evaluation for larger, diverse protein language models (raised by all the reviewers)
The authors included evaluations of ProGen2 and DPLM, verifying the generalization of the UCC steering method across various architectures.

2. Mechanistic understanding of repetition in protein language models (reviewer g59x)
The authors provided a detailed analysis and a hypothesis to clarify the domain-specific repetition mechanism in protein language models.

3. Improved interpretation of steering directions (reviewer nKVd)

**Concerns not addressed**
1. Expanded evaluation for non-protein language model architecture, like ProteinMPNN (Reviewer kZvX) and MultiFlow (Reviewer 45fe)
2. Expanded evaluation for existing in-practice strategies to reduce sequence repetition (Reviewer kZvX)

**Reviewer Scores:**

Reviewer kZvX: 8
Reviewer nKVd: 6
Reviewer 45fe: 4
Reviewer g59x: 6

---

### Decision · Program_Chairs · 2026-01-26

Accept (Poster)